# Symmetry fractionalization, mixed-anomalies and dualities in quantum spin models with generalized symmetries

Heidar Moradi[1*], Ömer M. Aksoy[2†], Jens H. Bardarson[3] and Apoorv Tiwari[3‡]

**1** School of Physics and Astronomy, University of Kent, Canterbury CT2 7NZ, United Kingdom
**2** Condensed Matter Theory Group, Paul Scherrer Institute,
CH-5232 Villigen PSI, Switzerland
**3** Department of Physics, KTH Royal Institute of Technology, Stockholm, 106 91 Sweden

⋆ haider.moradi@gmail.com , † omermertaksoy@gmail.com , ‡ t.apoorv@gmail.com

## Abstract

We investigate the gauging of higher-form finite Abelian symmetries and their sub-groups in quantum spin models in spatial dimensions $d = 2$ and 3. Doing so, we naturally uncover gauged models with dual higher-group symmetries and potential mixed 't Hooft anomalies. We demonstrate that the mixed anomalies manifest as the symmetry fractionalization of higher-form symmetries participating in the mixed anomaly. Gauging is realized as an isomorphism or duality between the bond algebras that generate the space of quantum spin models with the dual generalized symmetry structures. We explore the mapping of gapped phases under such gauging related dualities for 0-form and 1-form symmetries in spatial dimension $d = 2$ and 3. In $d = 2$, these include several non-trivial dualities between short-range entangled gapped phases with 0-form symmetries and 0-form symmetry enriched Higgs and (twisted) deconfined phases of the gauged theory with possible symmetry fractionalizations. Such dualities also imply strong constraints on several unconventional, i.e., deconfined or topological transitions. In $d = 3$, among others, we find, dualities between topological orders via gauging of 1-form symmetries. Hamiltonians self-dual under gauging of 1-form symmetries host emergent non-invertible symmetries, realizing higher-categorical generalizations of the Tambara-Yamagami fusion category.

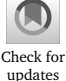

# 1  Introduction

Global symmetries play a fundamental role in understanding many aspects of quantum physics. The existence of symmetries aids in the organization of states and operators into representations of the global symmetry and imposes non-perturbative constraints on the dynamics and low-energy phases and phase transitions realized in a quantum system. Applications of symmetry are at the heart of much of modern physics. For instance, global symmetries constrain the particle content of the Standard Model of particle physics and provide the basis for Landau's classification scheme of phases of matter.

In the past decade, there has been a paradigm shift in the understanding of global symmetries in quantum field theory, based on the insight that any topological sub-sector of operators in a quantum field theory embodies a symmetry [1]. This has led to vast generalizations beyond the conventional notion of symmetry, which relied on the existence of global symmetry operators defined on all of space, or more generally on co-dimension-1 hypersurfaces in

spacetime, which satisfy group composition rules and commute with the Hamiltonian. The operators *charged* under such conventional symmetries are point-like or zero-dimensional and transform in representations of the global symmetry. Such symmetries have been generalized in two broad directions, corresponding to identifying two classes of topological operators as symmetries. Namely, allowing symmetry operators (i) to be defined on sub-manifolds of spacetime that have co-dimension higher than one has led to so-called higher-group symmetries [1–15], and (ii) to satisfy more general composition rules than those of a group has lead to non-invertible symmetries [16–52]. Within the domain of higher-group symmetries, a $p$-form symmetry is generated by a codimension-$(p+1)$ dimensional operator and operators charged under such symmetries have dimension greater than or equal to $p$ [1,47,49].[1] Non-invertible symmetries, instead, as the name suggests, involve symmetry operators that do not have any inverse. Composition rules for non-invertible symmetry operators correspond to an algebra rather than a group. All these generalizations, encompassing higher-group and non-invertible symmetries, collectively fall under the umbrella of global categorical symmetries. This name owes itself to the significant role played by higher fusion categories [53,54] in describing such symmetries and their charges. Just as group theory and group representation theory provide the mathematical language for conventional symmetries, fusion categories organize the composition rules of topological operators across all (co)-dimensions. Moreover, fusion categories also capture intricate topological information, including quantum anomalies and other refined features such as symmetry fractionalization patterns.

In a short span of time, global categorical symmetries have already made numerous important contributions in advancing our understanding of fundamental problems in physics. Key accomplishments include resolving the phase diagram of pure non-Abelian gauge theories [4], elucidating the phase diagram of adjoint quantum chromodynamics in 1+1 dimensions [55] and expanding Landau's paradigm to incorporate topologically ordered phases of matter [56,57]. Global categorical symmetries also played a central role in recent constructions that organize the symmetry aspects of a quantum system into a topological order in one dimension higher. These go under the name of symmetry topological field theories [20,58–60] in the high-energy physics community and topological holography [57] or holographic or categorical symmetry [61–64] in the condensed-matter community. Such constructions are an efficient way to unravel large webs of dualities, related to topologically manipulating the symmetry aspects of the system while leaving the local physics unchanged. Constructions applicable to quantum lattice models have also been used to unify Landau and beyond Landau physics [57] in $d = 1$ and in the domain of finite Abelian symmetries.

Gauging a global symmetry is a well-understood method to manipulate the symmetry structure of a system in a controlled yet nontrivial way. It provides insights into subtle aspects of global categorical symmetries and facilitates an efficient search for quantum theories exhibiting diverse symmetries. Gauging involves transforming a theory with a global symmetry into a theory with a local symmetry or redundancy, achieved by coupling the theory to a background gauge field and summing over its realizations. The resulting gauged theory possesses global symmetries that extend beyond conventional 0-form global symmetries and can be deduced in full generality, allowing for the construction of models with potentially novel symmetries [18, 27, 30, 44, 45, 65, 66]. As an example gauging a $p$-form finite Abelian group $\mathsf{G}$ in $d + 1$ dimensions, delivers a theory which has a $(d - p - 1)$-form symmetry corresponding to the Pontrjagin dual $\mathsf{G}^\vee$, the group of homomorphisms from $\mathsf{G}$ to $U(1)$.[2] When the total $p$-form group is a central extension of $\mathsf{K}$ by $\mathsf{N}$ determined by an extenstion class $\kappa$, gauging

---

[1]Here *codimension* is defined relative to the spacetime dimension. In $d + 1$ spacetime dimensions, a $p$-form symmetry is a topological defect defined on $(d + 1) - (p + 1) = d - p$ dimensional submanifolds.

[2]More precisely, this is the invertible sub-category inside the category of $d$-representations of a $p$-form group $\mathsf{G}$. We will however only describe the invertible component. The rest of the symmetry category can be obtained by considering all possible condensation defects.

N ⊂ G produces a gauged theory with a $p$-form symmetry K, a $(d-p-1)$-form symmetry $N^\vee$ and a mixed anomaly between them, which depends on $\kappa$ [65]. Similarly, theories with non-invertible symmetries can be obtained by gauging subgroups that act via outer automorphisms on [27], or have a mixed anomaly with the remaining symmetry structure [24]. Another avenue for non-invertible symmetries is gauging invertible symmetries on sub-manifolds of spacetime [26,35,67]. The symmetry defects thus obtained are referred to as condensation defects. To summarize, many of the constructions of models with exotic global categorical symmetries employ some kind of generalized gauging procedure. Gauging provides a systematic playground to start with a theory that has a familiar symmetry structure and 'discover' quantum systems with novel symmetry structures. In this early stage in the study of global categorical symmetries this contributes to a systematic understanding of these novel symmetry structures.

Yet another reason to study gauging of finite global symmetries is that such gaugings are realized as dualities in quantum systems. For example, the well-known Kramers-Wannier and Jordan-Wigner dualities are essentially gaugings of the $\mathbb{Z}_2$ internal and $\mathbb{Z}_2$ fermion-parity symmetry in $1d$ lattice models [68]. Dualities can be used to provide profound non-perturbative insights into quantum systems and are therefore very valuable. Furthermore, gauging related dualities in dimensions higher than $d=1$ map short-range entangled states to long-range entangled or topologically ordered states. For instance gauging a $\mathbb{Z}_n$ symmetric paramagnet in $d=2$ gives the $\mathbb{Z}_n$ topological order [69]. Recently, it has been appreciated that gauging can be implemented in quantum circuits via measurements [70–72] and since it is desirable for quantum computation platforms to prepare such states [73], understanding such dualities is a pre-requisite.

While much of the impetus driving the understanding of global categorical symmetries comes from quantum field theoretic studies, our work takes a distinct approach by examining various aspects of global categorical symmetries in the lattice setting. We mostly restrict ourselves to higher-group symmetries with possible 't Hooft anomalies. A theory has an 't Hooft anomaly with respect to a global symmetry if the partition function of the theory coupled to background symmetry gauge fields is not gauge invariant. Instead the partition function transforms under background gauge transformations by a $U(1)$ phase which cannot be absorbed by any local counter-terms, but can however be absorbed by an invertible topological field theory in one higher dimension [74,75]. A related consequence is that such a symmetry cannot be promoted to a gauge symmetry. However, certain so called mixed 't Hooft anomalies involve more than one symmetry group such that when restricted to any single symmetry group, the anomaly is nullified or trivialized. In such cases, it is possible to gauge any single symmetry group but not the full symmetry structure. The anomalies we encounter in this paper are mixed 't Hooft anomalies involving higher groups.

We study spin systems defined on a $d$-spatial dimensional lattice with each $p$-dimensional cell (i.e., vertices for $p=0$, edges for $p=1$ etc.) equipped with a finite dimensional Hilbert space, typically the group algebra of a finite Abelian group G. In more conventional condensed matter physics language, these are nothing but spin degrees of freedom (for example, single species of standard spin-$\frac{1}{2}$ d.o.f. for $G=\mathbb{Z}_2$).[3] Within such a setup, a $p$-form symmetry corresponding to the group G is generated by operators defined on any closed $(d-p)$-dimensional sub-lattice (see Fig. 1). We organize the space of $p$-form symmetric quantum Hamiltonians as an algebra $\mathfrak{B}_p$ of operators that commute with the $p$-form symmetry. Such a *bond algebra* has already been useful in understanding dualities and systematizing the space of quantum systems with fixed global symmetries [76–79]. Gauging the $p$-form symmetry amounts to making

---

[3]In this paper we employ lattice gauge theory and simplicial calculus language, as it makes many aspects of the construction more transparent. But note that underlying everything is nothing but spin models.

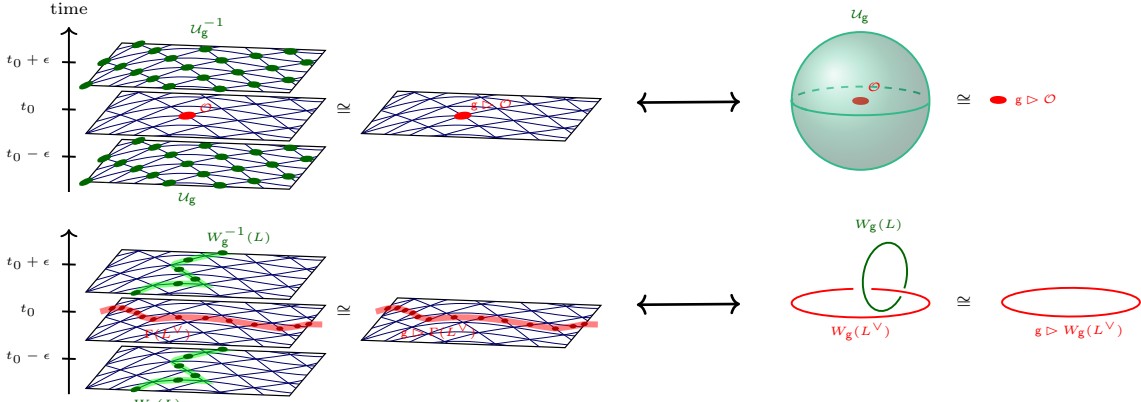

Figure 1: The action of symmetry operators via conjugation can be understood as topological linking in spacetime. 0-form symmetry operators act on all of space therefore, conjugating a local operator with the symmetry operators amounts to linking a point with a co-dimension-1 sphere in the spacetime picture. Similarly 1-form symmetries are generated by co-dimension-1 operators in space. These act on lines by conjugation. Intersection on a time slice becomes linking in the spacetime picture.

the symmetry local.[4] This is done by introducing $G$-valued gauge degrees of freedom on the $(p + 1)$-cells and demanding local gauge invariance by requiring that a collection of Gauss operators act as the identity on the gauged system. Doing so, one obtains a dual bond algebra $\mathfrak{B}^{\vee}_{d-p-1}$, isomorphic to $\mathfrak{B}_p$. We analyze the symmetries of $\mathfrak{B}^{\vee}_{d-p-1}$ and recover the dual $(d - p - 1)$-form $G^{\vee}$ symmetries one expects upon gauging a finite Abelian 0-form symmetry. We carry out an analogous procedure for gauging subgroups of $p$-form finite Abelian groups. Therefore one finds the following gauging-related isomorphism of bond-algebras

$$\mathfrak{B}_p \xrightleftharpoons[\text{gauging } G^{\vee} (d-p-1)\text{-form sub-symmetry}]{\text{gauging } G \ p\text{-form sub-symmetry}} \mathfrak{B}^{\vee}_{d-p-1}. \tag{1}$$

In the case of gauging finite subgroups, this allows us to pinpoint lattice manifestations of mixed quantum anomalies. Quantum anomalies in lattice systems are much less understood [80–83] as compared with their field theoretic counterparts. In particular there has been an effort towards understanding mixed anomalies involving crystalline symmetries on the lattice due to their expected connection with Lieb-Schultz-Mattis constraints [83–101]. In this regard, we hope that our work will shed light on how to diagnose mixed anomalies in lattice models. Specifically, we find that a mixed anomaly between a $p$-form symmetry $K$ and a $(d-p-1)$-form symmetry $N^{\vee}$ manifests as the fractionalization involving the two symmetries $K$ and $N^{\vee}$ that participate in the anomaly. See [45], for a higher-categorical discussion of such anomalies. Symmetry fractionalization is well-studied in topological orders [7, 102, 103], with fractional quantum Hall (FQH) systems providing the paradigmatic examples where anyons display $U(1)$ symmetry fractionalization by carrying a fractional $U(1)$ charge. Recently the role of symmetry fractionalization in understanding quantum anomalies has also been emphasized [45, 104,

---

[4]The meaning of *making a symmetry local* requires clarification for higher-form symmetries. For conventional 0-form symmetries, a symmetry is parameterized by a 0-form $\lambda_0$: $\phi \rightarrow \phi + \lambda_0$ such that $d\lambda_0 = 0$ (i.e. $\lambda_0$ is constant). Making it local means we want it to be invariant even when $d\lambda_0 \neq 0$. This is done by introducing a 1-form gauge field $a$ with transformation $a \rightarrow a + d\lambda_0$ and constructing minimal couplings to $\phi$ (covariant derivatives in the continuum). For more general $p$-form symmetries, the symmetry is parameterized by $p$-forms $\lambda_p$ such that $d\lambda_p = 0$, i.e. co-cycles. Making it *local* means it should be invariant under any co-chain, i.e. even when $d\lambda_p \neq 0$. This requires the introduction of a $(p + 1)$-form gauge field $a_{p+1}$ with $a_{p+1} \rightarrow a_{p+1} + d\lambda_p$.

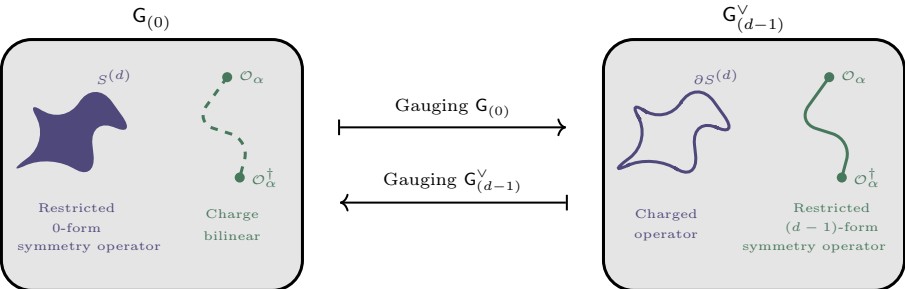

Figure 2: The gauging-related duality realized as an isomorphism of bond algebras, furnishes a mapping of order parameters. The figure depicts a mapping of operators between a bond algebra $\mathfrak{B}$ with 0-form symmetry $G_{(0)}$ and a gauged bond algebra $\mathfrak{B}^\vee$ with a $(d-1)$-form symmetry $G^\vee$. A 0-form symmetry operator restricted to an open ball-like region $S^{(d)}$ maps to an operator charged under the dual $(d-1)$-form symmetry at the boundary of $S^{(d-1)}$. A 0-form charge bilinear located at points $x_i$ and $x_f$ maps to $(d-1)$-form symmetry operator, i.e., a non-unique line connecting the $x_i$ and $x_f$.

105] however a lattice study remains missing. In this work, we detail such a relation between symmetry fractionalization patterns and mixed anomalies in lattice spin models.

The fact that gauging is realized as an isomorphism of bond algebras implies a duality between the pre-gauged system $\mathfrak{T}$ and gauged system $\mathfrak{T}^\vee$. Strictly speaking, such dualities are invertible only when one considers all the symmetry sectors of $\mathfrak{T}$ and $\mathfrak{T}^\vee$ [106], where a symmetry sector is specified by symmetry twisted boundary conditions and symmetry eigenspaces. More precisely, the duality implies that the spectrum of a Hamiltonian in a certain symmetry sector and its dual gauged Hamiltonian in a dual symmetry sector are the same. Another consequence is the equality of correlation functions

$$\langle \mathcal{O}_1(x_1, t_1) \cdots \mathcal{O}_n(x_n, t_n) \rangle_\Phi = \langle \mathcal{O}_1^\vee(x_1, t_1) \cdots \mathcal{O}_n^\vee(x_n, t_n) \rangle_{\Phi^\vee}, \tag{2}$$

where $\Phi$ collectively denotes the symmetry sector and twisted boundary condition labels of theory $\mathfrak{T}$ and $\mathcal{O}_j$ are operators in the bond algebra $\mathfrak{B}$. $\Phi^\vee$ and $\mathcal{O}_j^\vee$ are the images of $\Phi$ and $\mathcal{O}_j$ under the gauging map. This in turn implies that the phase diagrams of $\mathfrak{T}$ and $\mathfrak{T}^\vee$ are isomorphic. Therefore such dualities can be used to read off many non-perturbative constraints on the phase diagrams of the systems being investigated. Gapped phases on either side of the duality, as well as universality classes of phases transitions, can be mapped. Knowing the order parameters of a certain ordered phase in $\mathfrak{T}$ can be used to immediately furnish the order parameters for the dual phase in $\mathfrak{T}^\vee$ (see Fig. 2). See also [57] for a detailed holographic perspective in $2 + 1/1 + 1$ dimensions.

Recently, dualities in spin models related to partial gauging of finite Abelian symmetry have been studied [107–109]. In [107], the mapping of transitions under such dualities in $1d$ was emphasized and it was pointed out that deconfined quantum critical [110–113] transitions realized in the model after partial gauging are dual to conventional Landau transitions in the model before partial gauging. Therefore such dualities provide a promising avenue to bootstrap the understanding of conventional transitions to explore unconventional transitions. In this work, we harness this methodology to explore several unconventional transitions in $d = 2$ and 3 dimensional models with mixed anomalies involving higher group symmetries. For instance, in $d = 2$ the Landau symmetry-breaking transition map models with $\mathbb{Z}_n$ 0-form symmetry map to anyon condensation transitions [114, 115] between topological orders after partial-gauging. Similarly, transitions between symmetry protected topological phases (SPTs) map to transitions between distinct symmetry enriched topological orders after partial gaug-

ing. In $d = 3$, among others, we find dualities between topological orders via gauging of 1-form symmetries. Although in this paper we confine ourselves to studying dualities from (partial) gauging of (higher) symmetries, other types of dualities exist related to automorphism group or cohomology group of global symmetries [57], for example by stacking SPT phases. Gauging dualities map $p$-form symmetries to $(d-p-1)$-form symmetries and thus there are usually no self-dualities. Few exceptions are in even spacetime dimensions (0-form symmetries in $1+1$ dimensions or 1-form symmetries in 3+1 dimensions). However, by combining gauging with these other dualities, such as SPT stacking, it is possible to find dualities between phases of the same type of symmetry in any dimension. Self-dual points of such dualities will give rise to exotic non-invertible symmetries. For example, a duality between $2+1$ dimensional toric code and double semion model can be constructed by gauging a 1-form symmetry, stacking with a 0-form SPT phase and then gauge the 0-form symmetry. There will exist a phase-transition between these topological orders that is self-dual under this mapping.

**Summary of results:**    In this work, we study the gauging of finite Abelian higher-form symmetries and their subgroups in quantum spin models. Along the way we clarify various notions related to mixed anomalies and symmetry fractionalization patterns, as well as detail how gaugings of finite generalized symmetries are realized as dualities between classes of quantum spin models with certain dual global symmetries. We discuss the mapping of gapped phases under such gauging-related dualities and also discuss more general consequences for the structure of the phase diagrams of the dual quantum systems. Below is a summary of the main results:

1. We describe the gauging of higher-form finite Abelian symmetries and sub-symmetries on the lattice.

2. We study the mapping of the energy spectrum under gauging dualities. In particular of symmetry sectors, i.e., symmetry eigen-sectors and symmetry twisted boundary conditions, under dualities related to partial gauging of higher-form symmetries.

3. We clarify how mixed-anomalies involving higher-form global symmetries manifest on the lattice. For instance, we investigate the higher-group with a $\mathbb{Z}_2$ $p$-form and $\mathbb{Z}_2$ $(d-p-1)$-form symmetry and a mixed anomaly given by

$$S_{\text{anom}} = \mathrm{i}\pi \int A_{p+1} \cup \text{Bock}(A_{d-p}), \tag{3}$$

where $A_{p+1}$ and $A_{d-p}$ are the background gauge field for the $p$-form and $(d-p-1)$-form symmetry. The anomaly manifests as a symmetry fractionalization pattern such that the $\mathbb{Z}_2$ $p$-form (respectively $(d - p - 1)$-form) symmetry fractionalizes to $\mathbb{Z}_4$ depending on the symmetry twisted boundary condition of the $(d - p - 1)$-form (respectively $p$-form) symmetry. More precisely

$$\begin{aligned}
\mathcal{U}_p^2(\Sigma^{(d-p)}) = \mathcal{T}_{d-p-1}(\Sigma^{(d-p)}) &= \exp\left\{\mathrm{i}\pi \oint_{\Sigma^{(d-p)}} A_{d-p}\right\}, \\
\mathcal{U}_{d-p-1}^2(\Sigma^{(p+1)}) = \mathcal{T}_p(\Sigma^{(p+1)}) &= \exp\left\{\mathrm{i}\pi \oint_{\Sigma^{(p+1)}} A_{p+1}\right\},
\end{aligned} \tag{4}$$

where $\mathcal{U}_q$ and $\mathcal{T}_q$ are the operators that implement the $q$-form symmetry and measure the $q$-form symmetry twisted boundary conditions respectively.

4. We study how gapped phases dualize in $d = 2$ and 3 dimensions under (partial) gauging of global 0-form and 1-form symmetries and point out symmetry fractionalization patterns that distinguish certain gapped phases. For example, this leads to interesting concrete spin models with anyonic excitations that carry a fractional charge of a global symmetry, reminiscent of the FQHE.

5. We describe how the order parameters of all gapped phases map under gauging and partial gauging related dualities (see Fig. 2). These can be used to compute non-trivial phase-diagrams and study phase-transitions in higher-dimensions, similar to [57] in 1+1 dimensions.

6. Describe a $\mathbb{Z}_n$ 1-form generalization of Kramer's Wannier duality in $d = 3$. Amongst many things, this enables us to show a certain duality between $\mathbb{Z}_k$ and $\mathbb{Z}_{n/k}$ topological orders in $d = 3$. Furthermore it also allows us to construct spin models in 3+1 dimensions with non-invertible symmetries, for example at phase-transitions between certain topological ordered phases.

7. Along the way we connect various field-theoretic aspects of gauge models, parallel transport as well as notions from differential and simplicial geometry to the context of quantum spin models.

**Organization of the paper**

The paper is organized as follows. In Sec. 2, we describe dualities obtained by gauging finite Abelian 0-form (sub-)symmetries as bond algebra isomorphisms. Section 3 describes such dualities from a quantum field theory point of view. In Secs. 4 and 5, we explore how the dualities act on the phase diagrams of two and three dimensional spin models, respectively. Section 6 focuses on gauging finite Abelian 1-form global symmetries and the corresponding dualities in two- and three-dimensional space. Section 7 concludes.

**Notation and conventions**

Here we briefly summarize the notation and conventions adopted in this paper.

- We denote by $d$ the spatial dimensions while spacetime dimension is denoted by $(d+1)$.

- We denote by $M$ the $(d+1)$-dimensional spacetime manifold and often assume that $M = M_d \times S^1$, where $M_d$ is a $d$-dimensional spatial manifold. We use a triangulation of $M_d$ denoted by $M_{d,\triangle}$. In Sec. 6, we use a square or cubic lattice, but with slight abuse of notation, we continue to denote it as $M_{d,\triangle}$.

- We denote by Greek letters $\Sigma^{(p)}$ and $\gamma$ non-contractible $p$ and 1-cycles on $M$. $S^{(p)}$ and $L$ are used for general $p$-chains and 1-chains on $M$ respectively.

- On the triangulation $M_{d,\triangle}$, we denote by $\mathsf{e} \in M_{d,\triangle}$ and $\mathsf{p} \in M_{d,\triangle}$ the oriented edges and plaquettes, respectively. We denote by $\mathsf{o}(\mathsf{e},\mathsf{p}) = \pm 1$ the relative orientation between the edge $\mathsf{e}$ and plaquette $\mathsf{p}$ such that $\mathsf{e} \subset \mathsf{p}$, i.e., we assign $\mathsf{o}(\mathsf{e},\mathsf{p}) = +1$ or $\mathsf{o}(\mathsf{e},\mathsf{p}) = -1$ if orientation of edge $\mathsf{e}$ aligns or anti-aligns with that of plaquette $\mathsf{p}$, respectively. Similarly, given any 1-chain $L$, we denote by $\mathsf{o}(\mathsf{e},L)$ and $\mathsf{o}(\mathsf{p},S^{(p)})$ the relative orientations between 1-chain $L$ and the edge $\mathsf{e} \subset L$ and between $p$-chain $S^{(p)}$ and the plaquette $\mathsf{p} \subset S^{(p)}$, respectively.

- We denote by $\mathsf{G}_{(p)}$ a $p$-form symmetry group $\mathsf{G}$.

- We denote by $G^{\epsilon}_{(0,d-1)} = \left[ K_{(0)}, N_{(d-1)} \right]^{\epsilon}$ a $d$-group with 0-form symmetry $K_{(0)}$, $(d-1)$-form symmetry $N_{(d-1)}$ and a mixed anomaly which depends on $\epsilon$. Other higher groups are denoted in a similar fashion.

- We denote by $A^{(H)}_p$ the $p$-form background gauge field associated with group $H$. We denote by lowercase $a^{(H)}_p$ the dynamical $p$-form gauge fields associated with group $H$. We drop the superscript $H$ when the group corresponding to $p$-form gauge fields $A_p$ $a_p$ is clear from the context.

- $\mathcal{B}_G(\mathcal{V})$ denotes the bond algebra of $G$ symmetric operators on the Hilbert space $\mathcal{V}$.

- We make extensive use of simplicial calculus notation to discuss spin models. For a quick review of simplicial calculus, we refer the reader to Appendix E of Ref. [57]. For more details, see the standard texts [116, 117] on algebraic topology.

For readers that are interested in spin models but unfamiliar with algebraic topology, we will briefly define the minimal set of objects and their relation to spin model language. A triangulation $M_{d,\triangle}$ is a decomposition of a manifold $M$ into $n$-simplices $[v_0, \ldots, v_n]$. A 0-simplex $v$ is a vertex, 1-simplex $e = [v_0, v_1]$ is an edge, a 2-simplex $p = [v_0, v_1, v_2]$ is a plaquette and so on. The ordering of the vertices in $[v_0, \ldots, v_n]$ defines an orientation. We denote by $C^n(M, G)$, the set of $G$-valued $n$-cochains. In words, $\phi \in C^n(M, G)$ is a spin configuration (a map) that assigns to each simplex $[v_0, \ldots, v_n]$ a $G$-calued spin. For example, a 0-cochain $\phi \in C^0(M, G)$ is a spin configuration of $G$-valued spins, i.e., an assignment of a value $\phi_v \in G$ to each vertex $v$. For $G = \mathbb{Z}_2$, $\phi$ is just a spin configuration of a quantum spin-1/2 model and whereby $|\phi\rangle = |\phi_{v_1}, \phi_{v_2}, \ldots\rangle$ denotes the spin-1/2 basis.[5] Similarly, a 1-cochain $a \in C^1(M, G)$ is a spin configuration living on each edge $a_e = a_{[v_0,v_1]}$, a 2-cochain $b \in C^2(M, G)$ is a spin configuration on each plaquette $b_p = b_{[v_0,v_1,v_2]}$ and so on.

Next we need the so-called coboundary map $d : C^n(M, G) \rightarrow C^{n+1}(M, G)$. If $\phi$ is a 0-simplex, then $d\phi$ is a 1-simplex and on edges it evaluates to $d\phi([v_0, v_1]) = \phi_{v_1} - \phi_{v_0}$. For example for $G = \mathbb{Z}_2$, this measures whether two neighbouring spins point the same direction or not. Similarly for 1-simplices it is defined as $da([v_0, v_1, v_2]) = a_{[v_1,v_2]} - a_{[v_0,v_2]} + a_{[v_0,v_1]}$.

Finally we need the cup product, which from a $p$-cochain $a_p$ and a $q$-cochain $b_q$ define s a $p + q$ cochain $c_{p+q} = a_p \cup b_q$. This acts on a $p + q$ simplex as $a_p \cup b_q([v_0, \ldots, v_{p+q}]) = a_p([v_0, \ldots, v_p]) b_q([v_p, \ldots, v_{p+q}])$.

Cochains, coboundary maps and cup products are the discrete versions of differential forms, exterior derivatives and wedge products from differential geometry, respectively, and they satisfy similar properties.

## 2 Gauging as bond algebra isomorphisms

### 2.1 Gauging Abelian finite symmetry

In this section we review the gauging of finite 0-form symmetries $G$ in quantum spin models. To simplify our presentation, we focus on the case where $G = \mathbb{Z}_n$. However, the concepts and arguments can be readily extended to encompass any finite Abelian group.

---

[5]For clock model type spins we need $G = \mathbb{Z}_n$, for k-layers of spins $G = \mathbb{Z}_2 \times \cdots \times \mathbb{Z}_2$ and so on.

Consider a $d$ dimensional quantum spin model defined on the triangulation of an oriented manifold $M_d$ denoted as $M_{d,\triangle}$. Let each vertex v of $M_{d,\triangle}$ be endowed with a local $n$ dimensional complex Hilbert space $\mathcal{V}_\mathsf{v} \cong \mathbb{C}^n$. The total Hilbert space is a tensor product

$$\mathcal{V} = \bigotimes_\mathsf{v} \mathcal{V}_\mathsf{v}. \tag{5}$$

There is an action of $\mathbb{Z}_n$ clock and shift operators $\{X_\mathsf{v}, Z_\mathsf{v}\}_\mathsf{v}$ on $\mathcal{V}$ such that

$$Z_\mathsf{v} X_{\mathsf{v}'} = \omega_n^{\delta_{\mathsf{v}\mathsf{v}'}} X_{\mathsf{v}'} Z_\mathsf{v}, \qquad Z_\mathsf{v}^n = X_\mathsf{v}^n = 1, \qquad \forall\, \mathsf{v}, \mathsf{v}', \tag{6}$$

where $\omega_n := \exp\{2\pi \mathrm{i}/n\}$. We are interested in the space of Hamiltonians which are symmetric with respect to the $\mathbb{Z}_n$ symmetry generated by

$$\mathcal{U} = \prod_\mathsf{v} X_\mathsf{v}. \tag{7}$$

The space of linear local operators at each vertex v is spanned by $\mathcal{O}_\mathsf{v}^{(\mathsf{h},\alpha)} = X_\mathsf{v}^\mathsf{h} Z_\mathsf{v}^{-\alpha}$, where $\mathsf{h}, \alpha \in \{0, \dots, n-1\}$. Here $\alpha \in \mathrm{Rep}(\mathbb{Z}_n) \cong \mathbb{Z}_n$ labels the representation the operator $\mathcal{O}_\mathsf{v}^{(\mathsf{h},\alpha)}$ transforms in under a global symmetry transformation

$$\mathcal{U}^\mathsf{g} \mathcal{O}_\mathsf{v}^{(\mathsf{h},\alpha)} \mathcal{U}^{-\mathsf{g}} = \mathsf{R}_\alpha(\mathsf{g}) \mathcal{O}_\mathsf{v}^{(\mathsf{h},\alpha)}, \tag{8}$$

where $\mathsf{R}_\alpha(\mathsf{g}) = \omega_n^{\alpha \mathsf{g}}$. This decomposes the space of linear operators acting on $\mathcal{V}_\mathsf{v}$ into irreducible representations of $\mathbb{Z}_n$. We will use the terminology that operators with non-trivial $\alpha$ are *charged* under $\mathbb{Z}_n$. Now, the space of $\mathbb{Z}_n$ symmetric Hamiltonians is isomorphic to the bond algebra

$$\mathsf{B}_{\mathbb{Z}_{n,(0)}}(\mathcal{V}) = \left\langle X_\mathsf{v}, \; Z_{\mathsf{s}(\mathsf{e})} Z_{\mathsf{t}(\mathsf{e})}^\dagger \;\middle|\; \forall\, \mathsf{v}, \mathsf{e} \right\rangle, \tag{9}$$

where $\mathsf{s}(\mathsf{e})$ and $\mathsf{t}(\mathsf{e})$ are the source and target vertex of the oriented edge e

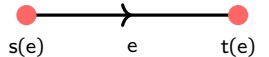

Said differently any $\mathbb{Z}_n$ symmetric Hamiltonian on $\mathcal{V}$ can be expressed as a sum of products of the generators $X_\mathsf{v}$, $Z_{\mathsf{s}(\mathsf{e})} Z_{\mathsf{t}(\mathsf{e})}^\dagger$ and is therefore an element of the algebra $\mathfrak{B}_{\mathbb{Z}_{n,(0)}}(\mathcal{V})$. Note that the generators only act on a single vertex or edge, but products and sums of these generate any other operator that commutes with (7). As written in (9), the bond algebra has no restrictions related to locality. However when constructing physical Hamiltonians, one typically imposes locality-related constraints such that the Hamiltonian is a sum of operators that each have support within some open ball-like region in $M_{d,\triangle}$. We will not attempt to formalize such constraints. Instead, we will put them in by hand when studying models in later sections.

### 2.1.1 Twisted boundary conditions: Gauge connections and parallel transport

Gauging, and dualities in general, act non-trivially on symmetry-twisted boundary conditions and symmetry sectors [57, 68]. Therefore it is insightful to define symmetry-twisted bond-algebras to keep track of how various symmetry sectors map under gauging-related dualities. Implementing a symmetry-twisted boundary-condition $\mathsf{g} \in \mathbb{Z}_n$ along a non-contractible cycle $\gamma$[6] means that any charged local operator $\mathcal{O}_\mathsf{v}^{(\mathsf{h},\alpha)}$ transforms as

$$\mathcal{O}_\mathsf{v}^{(\mathsf{h},\alpha)} \longrightarrow \mathcal{U}^\mathsf{g} \mathcal{O}_\mathsf{v}^{(\mathsf{h},\alpha)} \mathcal{U}^{-\mathsf{g}} = \omega_n^{\alpha \mathsf{g}} \mathcal{O}_\mathsf{v}^{(\mathsf{h},\alpha)}, \tag{10}$$

---

[6]We denote the non-contractible 1-cycles by $\gamma$ and more general paths or cycles by $L$.

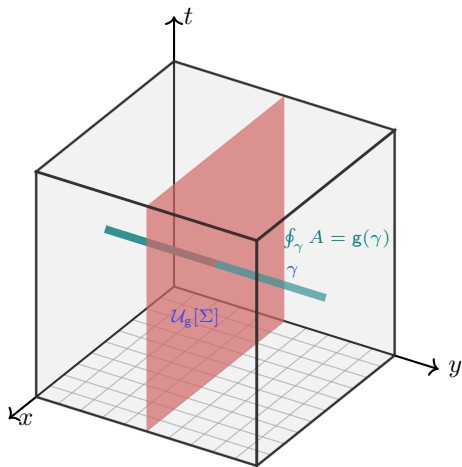

Figure 3: Symmetry-twisted boundary conditions with $g \in G$ are implemented by inserting a symmetry defect $\mathcal{U}_g[\Sigma]$ on a non-contractible cycle along time. Above illustrates a spacetime $M = S^1 \times T^2 = T^3$ (opposite sides of the cube are identified) and the symmetry defect wraps the $x-t$ cycle. Any bond-operator crossing the defect, will be transformed by g. An equivalent way to achieve this is to couple the symmetry G to a background connection $A$, with a holonomy $\oint_\gamma A = g(\gamma)$ along the cycle $\gamma$ dual to $\Sigma$. Any operator parallel transported along $\gamma$ will experience a symmetry-twist.

when the operator is transported along $\gamma$. From the space-time point of view such symmetry-twisted boundary conditions correspond to inserting a symmetry defect the extends along the time-direction, such that operators which cross the symmetry defect transform via the symmetry action (see figure 3). For example in the transverse field Ising model, anti-periodic boundary conditions are implemented this way by flipping the sign of the bond that crosses the symmetry defect: $\sigma_1^z \sigma_L^z \to \sigma_1^z \mathcal{U} \sigma_L^z \mathcal{U}^{-1} = -\sigma_1^z \sigma_L^z$, where $\mathcal{U} = \prod_i \sigma_i^x$ is the $\mathbb{Z}_2$ symmetry.

An alternative point of view, that is more in the spirit of this paper, is to couple the $\mathbb{Z}_n$ symmetric theory to a background gauge field (also called a connection). For translation symmetric theories on tori, this leads to symmetry-twisted translation operators with the property $T_g^{N_\gamma} = \mathcal{U}_g$, where $N_\gamma$ is the number of sites along the $\gamma$ cycle. This naturally implements (10), and corresponds to a group extension of the translation group with $\mathbb{Z}_n$ leading to fractionalized momenta. See appendix D of [57] for more details. Here we are interested in general triangulations of general manifolds, where the notion of translation symmetry might not be present. Consider a background gauge field $A \in Z^1(M_{d,\triangle}, \mathbb{Z}_n)$ (a $\mathbb{Z}_n$ connection) with non-trivial holonomy along non-trivial 1-cycles. Practically this means that we assign an element $A_e \in \mathbb{Z}_n$ on each edge such that $dA = 0$ and

$$\oint_\gamma A = g(\gamma) \in \mathbb{Z}_n \,, \tag{11}$$

where $g(\gamma)$ is the holonomy around $\gamma$ sometimes also referred to, somewhat misleadingly as the flux through the loop $\gamma$. The holonomy only depends on the homology class of $\gamma$, in particular $g(\gamma) = 0$ when $\gamma$ is contractible. Therefore, the background gauge-field assigns a $g \in \mathbb{Z}_n$ for each non-contractible cycle $[\gamma]$, corresponding to the symmetry-twisted boundary condition on that cycle. We can use the $\mathbb{Z}_n$ connection to define a form of parallel transport

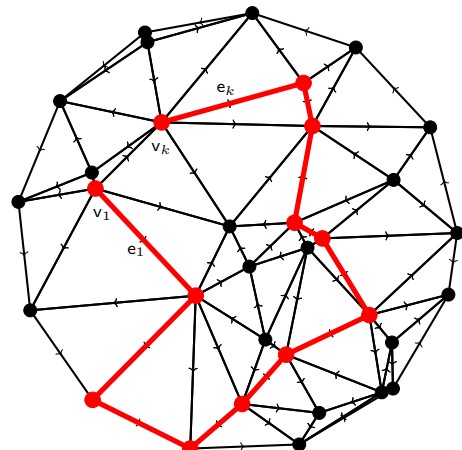

Figure 4: Parallel transport along a curve $L$ from $v_1$ to $v_k$ on $M_{d,\triangle}$ of the $\mathbb{Z}_n$ connection $a$.

along any curve $L$. First on each edge e consider the $T_e$ operator

$$
T_e \mathcal{O}_v T_e^{-1} = \begin{cases} \mathcal{O}_{t(e)}, & \text{if } v = s(e), \\ \mathcal{O}_{s(e)}, & \text{if } v = t(e), \\ \mathcal{O}_v, & \text{if } v \neq s(e), t(e), \end{cases} \tag{12}
$$

which permutes operators and states between vertices connected to the edge e. Here $\mathcal{O}_v$ is any local operator acting the vertex v. The $T_e$ operator can be constructed explicitly as

$$
T_e = \sum_{p,q=0}^{n-1} \omega_n^{pq} X_{s(e)}^p Z_{s(e)}^q X_{t(e)}^{-p} Z_{t(e)}^{-q}, \tag{13}
$$

see appendix B for details. For any curve $L = \{e_1, \ldots, e_k\}$ from vertex $v_1$ to $v_k$ (see Fig. 4), the parallel transport operator is defined as

$$
T_g[L] = \overleftarrow{\prod_{e \in L}} T_e \left[ X_{s(e)}^{o(e,L)} \right]^{A_e} = T_{e_k} \left[ X_{v_k}^{o(e_k,L)} \right]^{A_{e_k}} \cdots T_{e_1} \left[ X_{v_1}^{o(e_1,L)} \right]^{A_{e_1}}, \tag{14}
$$

where the arrow in $\overleftarrow{\prod}_{e \in L}$ indicates the direction of the product, and $o(e, L) = +1$ if the orientation of the edge e aligns with $L$ and $o(e, L) = -1$ if not. Note that we have labeled $T_g[L]$ using the holonomies g in (11), instead of the background gauge field $A$ since the former is the gauge-invariant content of $A$. One can readily check that for the local operator $\mathcal{O}_v^{(h,\alpha)} = X_v^h Z_v^\alpha$ transforming in the $\alpha$ representation of $\mathbb{Z}_n$ we have

$$
T_g[L] \mathcal{O}_{v_1}^{(h,\alpha)} T_g[L]^{-1} = \omega_n^{\alpha \int_L A} \mathcal{O}_{v_k}^{(h,\alpha)}, \tag{15}
$$

where $\int_L A = \sum_{e \in L} o(e, L) A_e$. The parallel transport of the charged operator accrues a $U(1)$ phase corresponding to the Wilson line along $L$ with charge $\alpha$. In particular, for closed loops we get

$$
T_g[L] \mathcal{O}_v^{(h,\alpha)} T_g[L]^{-1} = \omega_n^{\alpha \oint_L A} \mathcal{O}_v^{(h,\alpha)} = \omega_n^{\alpha g(L)} \mathcal{O}_v^{(h,\alpha)}, \tag{16}
$$

which is the holonomy associated to the background connection $A$, (11). With this, given any Hamiltonian constructed with the above bond-algebra we can define the symmetry twisted

Hamiltonian through the substitution of bond operators[7]

$$Z_{\mathsf{s(e)}}Z_{\mathsf{t(e)}}^{\dagger} = Z_{\mathsf{s(e)}}T_0[\mathsf{e}]Z_{\mathsf{s(e)}}^{\dagger}T_0[\mathsf{e}]^{-1} \longrightarrow Z_{\mathsf{s(e)}}T_{\mathsf{g}}[\mathsf{e}]Z_{\mathsf{s(e)}}^{\dagger}T_{\mathsf{g}}[\mathsf{e}]^{-1}. \tag{17}$$

Essentially, defining bond operators on an edge $\mathsf{e}$ using parallel transport from its source to its target vertex. The $X_{\mathsf{e}}$ operators are unaffected by this as they are not charged. Note that the product of bond-operators along a curve from $\mathsf{v}_1$ to $\mathsf{v}_k$ becomes

$$\prod_{\mathsf{e}\in L}\Big[Z_{\mathsf{s(e)}}Z_{\mathsf{t(e)}}^{\dagger}\Big]^{\mathsf{o(e},L)} = Z_{\mathsf{v}_1}Z_{\mathsf{v}_k}^{\dagger} \quad \longrightarrow \quad \prod_{\mathsf{e}\in L}Z_{\mathsf{s(e)}}^{\mathsf{o(e},L)}\,T_{\mathsf{g}}[\mathsf{e}]Z_{\mathsf{s(e)}}^{-\mathsf{o(e},L)}\,T_{\mathsf{g}}[\mathsf{e}]^{-1} = Z_{\mathsf{v}_1}\omega_n^{\int_L A}Z_{\mathsf{v}_k}^{\dagger}, \tag{18}$$

where $\omega_n^{\int_L A}$ is the Wilson line between charged operators which is the expected minimal coupling for a background gauge field. For non-contractible loops along homology cycles $\gamma \in H_1(M_{d,\triangle},\mathbb{Z}_n)$ we find

$$\prod_{\mathsf{e}\in\gamma}\Big[Z_{\mathsf{s(e)}}Z_{\mathsf{t(e)}}^{\dagger}\Big]^{\mathsf{o(e},\gamma)} \longrightarrow \omega_n^{\oint_\gamma A} = \omega_n^{\mathsf{g}(\gamma)}, \tag{19}$$

which correspond to the twisted boundary condition along that cycle. Operators that cross the symmetry operator insertion along the time direction, transform accordingly. With a slight abuse of notation, we will define the symmetry-twisted bond-algebra as

$$\mathsf{B}_{\mathbb{Z}_{n,(0)}}(\mathcal{V};\mathsf{g}) = \Big\langle X_{\mathsf{v}},\, Z_{\mathsf{s(e)}}Z_{\mathsf{t(e)}}^{\dagger} \,\Big|\, \prod_{\mathsf{e}\in\gamma}\Big[Z_{\mathsf{s(e)}}Z_{\mathsf{t(e)}}^{\dagger}\Big]^{\mathsf{o(e},\gamma)} \overset{!}{=} \omega_n^{\mathsf{g}(\gamma)} \quad \forall\, \mathsf{v},\mathsf{e}\Big\rangle. \tag{20}$$

A common way to implement this in spin-chain models is if there are $L$ sites along a cycle $\gamma$, define $Z_{L+1}^{\dagger} \equiv \omega_n^{\mathsf{g}(\gamma)}Z_1^{\dagger}$.[8] It is convenient to define the operator

$$\mathcal{T} = \prod_{\mathsf{e}\in\gamma}\Big[Z_{\mathsf{s(e)}}Z_{\mathsf{t(e)}}^{\dagger}\Big]^{\mathsf{o(e},\gamma)}, \tag{21}$$

as a way of *'measuring'* the twisted boundary condition, or equivalently the holonomy of the background connection $A$. Since all the operators in $\mathfrak{B}_{\mathbb{Z}_{n,(0)}}(\mathcal{V})$ (by definition) commute with $\mathcal{U}$, we can simultaneously block-diagonalize $\mathfrak{B}_{\mathbb{Z}_{n,(0)}}(\mathcal{V})$ into eigensectors of $\mathcal{U}$ labelled by $\alpha \in \mathrm{Rep}(\mathbb{Z}_{n,(0)})$. Doing so, the bond algebra decomposes as

$$\begin{aligned} \mathfrak{B}_{\mathbb{Z}_{n,(0)}}(\mathcal{V};\mathsf{g}) &= \bigoplus_\alpha \mathfrak{B}_{\mathbb{Z}_n}(\mathcal{V};(\alpha,\mathsf{g})), \\ \mathfrak{B}_{\mathbb{Z}_{n,(0)}}(\mathcal{V};(\alpha,\mathsf{g})) &= \Big\langle X_{\mathsf{v}},\, Z_{\mathsf{s(e)}}Z_{\mathsf{t(e)}}^{\dagger} \,\Big|\, \mathcal{U} \overset{!}{=} \omega_n^\alpha,\, \mathcal{T} \overset{!}{=} \omega_n^{\mathsf{g}(\gamma)} \quad \forall\, \mathsf{v},\mathsf{e}\Big\rangle. \end{aligned} \tag{22}$$

The notation $\mathcal{U} \overset{!}{=} \omega_n^\alpha$ means that we are restricting to the corresponding eigenspace of $\mathcal{U}$.

### 2.1.2 Gauging $\mathbb{Z}_n$ symmetry and the dual bond-algebra

Next, consider gauging the global $\mathbb{Z}_n$ symmetry which effectively amounts to turning the background gauge-field into a dynamical quantum field and make the theory invariant under local

---

[7]Note that in the absence of translation symmetry, the definition of twisted boundary condition is relative to another Hamiltonian. Given the Hamiltonian $H$, we can twist the boundary condition to obtain $H_{\mathsf{g}}$ by inserting symmetry operators in the time-direction or equivalently coupling to a background gauge field/connection. We can only talk about $H_{\mathsf{g}}$ as twisted, relative to $H$.

[8]Using parallel transport and the background gauge field $A$ is a more precise way to do this, we will however abuse the notation somewhat for simplicity.

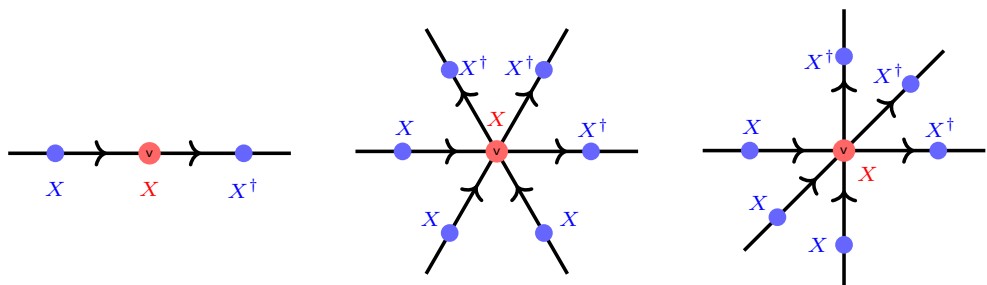

Figure 5: The figure depicts the operators that are used to define the Gauss operators in (a) d=1, (b) d=2 and (c) d=3 dimensions.

$\mathbb{Z}_n$ transformations. In order to do so, we first introduce gauge degrees of freedom ($\mathbb{Z}_n$ spins) on each link e, living in the Hilbert space $\mathcal{V}_e \cong \mathbb{C}^n$. The edge Hilbert space also admits an action of clock and shift operators $X_e, Z_e$ analogous to (6). We thus obtain the extended Hilbert space

$$\mathcal{V}_{\text{ext}} = \bigotimes_e \mathcal{V}_e \bigotimes_v \mathcal{V}_v = \text{Span}_{\mathbb{C}} \left\{ |a, \phi\rangle \mid a \in C^1(M_{d,\triangle}, \mathbb{Z}_n), \ \phi \in C^0(M_{d,\triangle}, \mathbb{Z}_n) \right\}, \tag{23}$$

where $C^p(M_{d,\triangle}, \mathbb{Z}_n)$ denotes the set of $\mathbb{Z}_n$-valued $p$-cochains on $M_{d,\triangle}$, i.e., an assignment of $\mathbb{Z}_n$ elements to the $p$-cells of $M_{d,\triangle}$. The clock and shift operators act on the basis states as

$$\begin{aligned}
Z_v |a, \phi\rangle &= \omega_n^{\phi_v} |a, \phi\rangle, & X_v |a, \phi\rangle &= |a, \phi + \delta^{(v)}\rangle, \\
Z_e |a, \phi\rangle &= \omega_n^{a_e} |a, \phi\rangle, & X_e |a, \phi\rangle &= |a + \delta^{(e)}, \phi\rangle,
\end{aligned} \tag{24}$$

where $\delta^{(v)}, \delta^{(e)}$ are $\mathbb{Z}_n$-valued 0 and 1-cochains such that

$$\left[\delta^{(v)}\right]_{v'} = \delta_{v,v'}, \qquad \left[\delta^{(e)}\right]_{e'} = \delta_{e,e'}. \tag{25}$$

The original spin degrees of freedoms on each vertex $\phi_v$ can be thought of as a matter field while the newly introduced spins on each edge $a_e$ can be thought of as a $\mathbb{Z}_n$ gauge field. The physical Hilbert space $\mathcal{V}_{\text{phys}} \subset \mathcal{V}_{\text{ext}}$ is defined as the eigenvalue +1 subspace of the collection of Gauss operators (see Fig. 5)

$$\mathcal{G}_v = X_v \prod_{e|s(e)=v} X_e^\dagger \prod_{e|t(e)=v} X_e =: X_v A_v^\dagger, \tag{26}$$

here e|s(e) and e|t(e) mean all edges e such that v is their source and target, respectively. The Gauss operator is defined such that the global $\mathbb{Z}_n$ transformation of charged operators

$$Z_v \longrightarrow \mathcal{U} Z_v \mathcal{U}^\dagger = \omega_n^{-1} Z_v, \tag{27}$$

become local gauge transformations

$$Z_v \longrightarrow \mathcal{G}[\lambda] Z_v \mathcal{G}[\lambda]^\dagger = \omega_n^{-\lambda_v} Z_v, \qquad Z_e \longrightarrow \mathcal{G}[\lambda] Z_e \mathcal{G}[\lambda]^\dagger = \omega_n^{\lambda_{s(e)} - \lambda_{t(e)}} Z_e = \omega_n^{-d\lambda_e} Z_e. \tag{28}$$

Here we have defined a general combination of Gauss operators parameterized by a $\mathbb{Z}_n$-valued 0-cochain $\lambda$ as $\mathcal{G}[\lambda] := \prod_v (\mathcal{G}_v)^{\lambda_v}$ that implements a $\mathbb{Z}_n$ gauge transformation. On the states, the Gauss operators act as

$$\mathcal{G}[\lambda]|a, \phi\rangle = |a + d\lambda, \phi + \lambda\rangle. \tag{29}$$

In this representation, the operators $X_e$ and $Z_e$ are the $\mathbb{Z}_n$ electric and gauge field respectively. Since the operator $Z_v$ is charged under the global symmetry which is being gauged, one needs to minimally couple the $Z_{s(e)}^\dagger Z_{t(e)}$ to the gauge field via the replacement

$$Z_{s(e)} Z_{t(e)}^\dagger \longrightarrow Z_{s(e)} Z_e Z_{t(e)}^\dagger, \tag{30}$$

which is the quantum version of (18). One can readily see that this minimally coupled operator is invariant under local gauge-transformations (28). Therefore the bond algebra after gauging is[9]

$$B_{\mathbb{Z}_{n,(d-1)}}(\mathcal{V}_{ext}) \simeq B_{\mathbb{Z}_n}/\mathbb{Z}_n = \left\langle X_v,\, Z_{s(e)} Z_e Z_{t(e)}^\dagger \,\Big|\, \mathcal{G}_v \overset{!}{=} 1,\, \prod_{e \in L}\left[Z_{s(e)} Z_e Z_{t(e)}^\dagger\right]^{o(e,L)} \overset{!}{=} 1 \,\forall\, e,v \right\rangle. \tag{31}$$

Note that there is an additional constraint $\prod_{e \in L}\left[Z_{s(e)} Z_e Z_{t(e)}^\dagger\right]^{o(e,L)} \overset{!}{=} 1$ for any cycle $L$ (contractible or not) on the lattice. This follows from the fact that this operator is the image of the operator $\prod_{e \in L}\left[Z_{s(e)} Z_{t(e)}^\dagger\right]^{o(e,L)} = 1$ in the pre-gauged bond algebra (9). Since gauging is an isomorphism of bond algebras, it maps the identity operator in $\mathcal{V}$ to the identity operator in $\mathcal{V}_{ext}$. See Appendix A for an alternative formulation where this appears more naturally (see Eq. (A.10)). A consequence of this is that in the physical Hilbert space[10]

$$da = 0, \tag{32}$$

and therefore $a \in Z^1(M_{d,\triangle}, \mathbb{Z}_n)$ corresponds to a bonafide $\mathbb{Z}_n$ gauge field. When mapping the symmetry twisted bond algebra (22) under the gauging-related bond algebra isomorphism, one obtains

$$\mathfrak{B}_{\mathbb{Z}_{n,(d-1)}}(\mathcal{V}_{ext}\,;(\alpha,g))$$
$$= \left\langle X_v,\, Z_{s(e)} Z_e Z_{t(e)}^\dagger \,\Big|\, \mathcal{G}_v \overset{!}{=} 1,\, \prod_{e \in L}\left[Z_{s(e)} Z_e Z_{t(e)}^\dagger\right]^{o(e,L)} \overset{!}{=} \omega_n^{g(L)},\, \mathcal{U} \overset{!}{=} \omega^\alpha \quad \forall\, e,v \right\rangle. \tag{33}$$

The constraints $\prod_{e \in L}\left[Z_{s(e)} Z_e Z_{t(e)}^\dagger\right]^{o(e,L)} \overset{!}{=} \omega_n^{g(L)}$, restrict the extended Hilbert space to a single gauge class of $\mathbb{Z}_n$ gauge fields labelled by $g \in H^1(M_{d,\triangle}, \mathbb{Z}_n)$, which satisfies

$$da = 0, \qquad \oint_L a = g(L). \tag{34}$$

Note that for contractible loops we have $g(L) = 0$. It is always more convenient to work in a basis in which the Gauss constraint has been solved. To do so, we perform a unitary transformation $U$ such that the Gauss operator $U\mathcal{G}_v U^\dagger$ only acts on the vertex Hilbert space $\mathcal{V}_v$ [118]. More precisely, we require a unitary $U$ such that

$$U\mathcal{G}_v U^\dagger = X_v. \tag{35}$$

In the basis (23), the unitary transformed Gauss operator is

$$U\mathcal{G}[\lambda]U^\dagger = \sum_{a,\phi} |a, \phi + \lambda\rangle\langle a, \phi|. \tag{36}$$

---

[9]We use the notation $\mathfrak{B}_{\mathbb{Z}_{n,(d-1)}}(\mathcal{V}_{ext})$ for the gauged bond algebra $\mathfrak{B}_{\mathbb{Z}_n}(\mathcal{V}_{ext})/\mathbb{Z}_n$, since after gauging it is the bond algebra of a $(d-1)$-form symmetry $\mathbb{Z}_{n,(d-1)}$ as we will shortly see.

[10]Note that $\prod_{e \in L} Z_e |a, \phi\rangle = \omega^{\oint_L a}|a, \phi\rangle = \omega^{\int_{S_L} da}|a, \phi\rangle$ by Stoke's theorem where $S_L$ is a surface such that $\partial S_L = L$. In particular for a loop $L_p$ around a plaquette $p$ we have $\oint_{L_p} a = (da)_p$.

Such a unitary can be conveniently expressed in terms of controlled-X operators as

$$U = \prod_{\mathsf{v}} \left[ \prod_{\mathsf{e}|\mathsf{t}(\mathsf{e})=\mathsf{v}} (CX^\dagger)_{\mathsf{v},\mathsf{e}} \prod_{\mathsf{e}|\mathsf{s}(\mathsf{e})=\mathsf{v}} (CX)_{\mathsf{v},\mathsf{e}} \right]. \tag{37}$$

Here $(CX)_{\mathsf{v},\mathsf{e}}$ and $(CX^\dagger)_{\mathsf{v},\mathsf{e}}$ act on the edge $\mathsf{e}$ and vertex $\mathsf{v}$ such that

$$\begin{aligned}
(CX)_{\mathsf{v},\mathsf{e}}|a_\mathsf{e}, \phi_\mathsf{v}\rangle &= |a_\mathsf{e} + \phi_\mathsf{v}, \phi_\mathsf{v}\rangle, \\
(CX^\dagger)_{\mathsf{v},\mathsf{e}}|a_\mathsf{e}, \phi_\mathsf{v}\rangle &= |a_\mathsf{e} - \phi_\mathsf{v}, \phi_\mathsf{v}\rangle,
\end{aligned} \tag{38}$$

The various operators transform under this unitary transformation as

$$\begin{aligned}
UZ_\mathsf{v}U^\dagger &= Z_\mathsf{v}, & UX_\mathsf{e}U^\dagger &= X_\mathsf{e}, \\
UX_\mathsf{v}U^\dagger &= X_\mathsf{v}A_\mathsf{v}, & UZ_\mathsf{e}U^\dagger &= Z^\dagger_{\mathsf{s}(\mathsf{e})}Z_\mathsf{e}Z_{\mathsf{t}(\mathsf{e})}.
\end{aligned} \tag{39}$$

One then obtains a bond algebra unitarily equivalent to (31) as

$$\widetilde{\mathfrak{B}}_{\mathbb{Z}_{n,(d-1)}}(\mathcal{V}_{\text{edge}}) = \left\langle A_\mathsf{v}, \, Z_\mathsf{e} \; \Big| \; \prod_{\mathsf{e}\in L} Z_\mathsf{e}^{\mathsf{o}(\mathsf{e},L)} \overset{!}{=} 1 \; \forall \, \mathsf{e}, \mathsf{v} \right\rangle, \tag{40}$$

where $A_\mathsf{v}$ was defined in (26). In writing this expression, we have removed the vertex degrees of freedom by implementing the unitary transformed Gauss constraint $X_\mathsf{v} \overset{!}{=} 1$. Therefore the bond algebra is an algebra of operators on the edge Hilbert space $\mathcal{V}_{\text{edge}} = \otimes_\mathsf{e} \mathcal{V}_\mathsf{e}$. The symmetry twisted bond algebra (33) has the following form in this basis

$$\widetilde{\mathsf{B}}_{\mathbb{Z}_{n,(d-1)}}(\mathcal{V}_{\text{edge}}; (\alpha, \mathsf{g})) = \left\langle A_\mathsf{v}, \, Z_\mathsf{e} \; \Big| \; \prod_{\mathsf{e}\in\gamma} Z_\mathsf{e}^{\mathsf{o}(\mathsf{e},\gamma)} \overset{!}{=} \omega_n^{\mathsf{g}(\gamma)}, \prod_\mathsf{v} A_\mathsf{v} \overset{!}{=} \omega_n^\alpha \; \forall \, \mathsf{e}, \mathsf{v} \right\rangle. \tag{41}$$

This bond algebra after gauging is symmetric with respect to a $(d-1)$-form symmetry that is dual to the 0-form symmetry (7) and is generated by closed (Wilson) loops on $L$ [see Eq. (A.15) in Appendix A]

$$W_L = \prod_{\mathsf{e}\in L} Z_\mathsf{e}^{\mathsf{o}(\mathsf{e},L)}. \tag{42}$$

Note that the symmetry twisted boundary conditions for a $(d-1)$-form symmetry are defined with respect to a non-contractible $d$-cycle, i.e. all of space. Therefore, we obtain that in the bond-algebra $\widetilde{\mathfrak{B}}_{\mathbb{Z}_{n,(d-1)}}$, the role of $\alpha$ and $\mathsf{g}$ are swapped, i.e., $\mathsf{g}$ label the symmetry eigensectors while $\alpha$ label the symmetry twisted boundary condition. In Sec. 3, we will describe the mapping of symmetry twisted sectors after gauging in a more general setting.

### 2.1.3 Dual higher-symmetries and twist defects

Gauge theories are atypical in condensed matter, however they can emerge as low energy descriptions of condensed matter models. It is often convenient to drop the constraint $\prod_L Z_\mathsf{e}^{\mathsf{o}(\mathsf{e},L)} = 1$ for contractible loops $L$, and consider a larger Hilbert space $\hat{\mathcal{V}}_{\text{edge}}$ with the bond algebra

$$\hat{\mathsf{B}}_{\mathbb{Z}_{n,(d-1)}}(\mathcal{V}_{\text{edge}}) = \left\langle A_\mathsf{v}, \, Z_\mathsf{e} \; \Big| \; \forall \, \mathsf{e}, \mathsf{v} \right\rangle. \tag{43}$$

This bond algebra has a subalgebra

$$\mathfrak{B}_{\left[\mathbb{Z}_{n,(d-1)}, \mathbb{Z}_{n,(1)}\right]}(\mathcal{V}_{\text{edge}}) = \left\langle A_\mathfrak{v}, \, B_\mathfrak{p} \; \Big| \; \forall \, \mathfrak{p}, \mathfrak{v} \right\rangle \subset \hat{\mathfrak{B}}_{\mathbb{Z}_{n,(d-1)}}(\mathcal{V}_{\text{edge}}), \tag{44}$$

which is the bond algebra of models with a $\mathbb{Z}_{n,(1)}$ in addition to the $\mathbb{Z}_{n,(d-1)}$ $(d-1)$ symmetry. Here the plaquette operator is defined as

$$B_\mathsf{p} = \prod_{\mathsf{e}\in\mathsf{p}} Z_\mathsf{e}^{\mathsf{o}(\mathsf{e},\mathsf{p})}, \tag{45}$$

which is the smallest contractible loop $B_\mathsf{p} = W_{\partial\mathsf{p}}$ around a plaquette p. The $(d-1)$-form symmetry is generated by lines (42), and the 1-form symmetry is generated by closed $(d-1)$-dimensional manifolds $S^{(d-2),\vee}$ in the dual lattice

$$\Gamma\left(S^{(d-1),\vee}\right) = \prod_{\mathsf{e}} X_\mathsf{e}^{\mathrm{Int}\left(\mathsf{e},\, S^{(d-1),\vee}\right)}, \tag{46}$$

where $\mathrm{Int}\left(\mathsf{e},\, S^{(d-1),\vee}\right)$ denotes the intersection number of the surface $S^{(d-1),\vee}$ and the edge e. $\mathrm{Int}\left(\mathsf{e},\, S^{(d-1)^\vee}\right) = 0$ when the edge and surface do not intersect and +1 or -1 if the edge is oriented along or against the outward normal of the surface (see Fig. 14). Compare these to (A.17) and (A.18). The vertex operators $A_\mathsf{v}$ are the smallest contractible sphere $S_\mathsf{v}^{(d-1),\vee}$ in the dual lattice around the vertex v, i.e., $A_\mathsf{v} = \Gamma(S_\mathsf{v}^{(d-1),\vee})$. The generators of the bond algebra $B_\mathsf{p}$ and $A_\mathsf{v}$ all commute with each other and in fact $\mathfrak{B}_{\left[\mathbb{Z}_{n,(d-1)},\mathbb{Z}_{n,(1)}\right]}(\mathcal{V}_{\mathrm{edge}})$ is the commutant algebra of $\hat{\mathfrak{B}}_{\mathbb{Z}_{n,(d-1)}}(\mathcal{V}_{\mathrm{edge}})$. The simplest Hamiltonian to write within this bond algebra is

$$H = -\sum_\mathsf{v} A_\mathsf{v} - \sum_\mathsf{p} B_\mathsf{p} + \mathrm{H.c.}, \tag{47}$$

which is nothing but the $\mathbb{Z}_n$ toric code in $d$ spatial dimensions. This model spontaneously breaks the $(d-1)$ and 1-form symmetries and is topologically ordered. Ground-states of this model satisfy $B_\mathsf{p} \overset{!}{=} 1$, which is the constraint for the gauge-invariant Hilbert-space. The gauge theory therefore emerges dynamically at low energies. If we add a term $\lambda \sum_e Z_e$, for small $\lambda$, the 1-form symmetry is explicitly broken. But the theory is still in the same topological phase, as the 1-form symmetry emerges at low energy and is spontaneously broken. This is a general property of topologically ordered phases where at low-energy it is described by a topological quantum field theory (TQFT), which has higher-form symmetries that are spontaneously broken.

We saw that the constraint $B_\mathsf{p} = \prod_{\mathsf{e}\in\mathsf{p}} Z_\mathsf{e}^{\mathsf{o}(\mathsf{e},\mathsf{p})} \overset{!}{=} 1$ was necessary for the mapping between (22) and (41) to be invertible. However, there is nothing inconsistent with the full unconstrained bond algebra (43). It is natural to wonder whether the duality holds on this larger algebra. In order to see how that works, let us decompose the Hilbert space into simultaneous eigenspaces of all plaquette operators $B_\mathsf{p}$

$$\hat{\mathcal{V}}_{\mathrm{edge}} = \bigoplus_{\Phi\in C^2(M_{d,\triangle},\mathbb{Z}_n)} \hat{\mathcal{V}}_{\mathrm{edge}}^\Phi, \tag{48}$$

where each 2-cochain $\Phi = \{\phi_\mathsf{p}\}$ is an assignment of $\phi_\mathsf{p} \in \mathbb{Z}_n$ values on each plaquette. All states in $\hat{\mathcal{V}}_{\mathrm{edge}}^\Phi$ are eigenstates of the plaquette operators with the eigenvalues $B_p|\psi\rangle_\Phi = \omega_n^{\phi_\mathsf{p}}|\psi\rangle_\Phi$. In particular, the space with all $\phi_\mathsf{p} = 0$ (denoted as $\Phi = 0$), satisfies the previous constraint $B_\mathsf{p} = 1$ for all p. We can similarly decompose the bond-algebra

$$\hat{\mathfrak{B}}_{\mathbb{Z}_{n,(d-1)}}(\mathcal{V}_{\mathrm{edge}}) = \bigoplus_{\Phi\in C^2(M_{d,\triangle},\mathbb{Z}_n)} \hat{\mathfrak{B}}_{\mathbb{Z}_{n,(d-1)}}(\hat{\mathcal{V}}_{\mathrm{edge}}^\Phi). \tag{49}$$

We have already seen that $\hat{\mathfrak{B}}_{\mathbb{Z}_{n,(d-1)}}(\hat{\mathcal{V}}_{\mathrm{edge}}^{\Phi=0})$ is dual to the bond algebra (22). Gauging gives rise

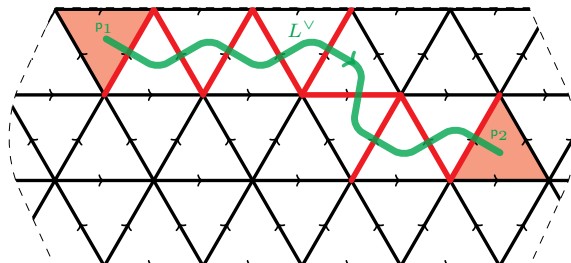

Figure 6: The bond algebra for the $(d-1)$-form symmetry $\hat{\mathfrak{B}}_{\mathbb{Z}_{n,(d-1)}}(\hat{\mathcal{V}}^{\Phi}_{\text{edge}})$, has the constraint $B_{\mathsf{p}} = \omega_n^{\phi_{\mathsf{p}}}$. This is dual to the bond algebra of 0-form symmetries $\mathsf{B}^{\Phi}_{\mathbb{Z}_{n,(0)}}(\mathcal{V})$ which is coupled to a background gauge-field $a \in C^1(M_{d,\triangle}, \mathbb{Z}_n)$ such that $(\mathrm{d}a)_{\mathsf{p}} = \phi_{\mathsf{p}}$. The picture shows an example where $\phi_{\mathsf{p}} = \alpha(\delta_{\mathsf{p},\mathsf{p}_2} - \delta_{\mathsf{p},\mathsf{p}_1})$. This leads to $a$ that is equal to zero on all edges, except the red edges. This is equivalent to the insertion of a 0-form symmetry operator in spacetime that crosses the each time slice along the green line, creating extrinsic twist defects at $\mathsf{p}_1$ and $\mathsf{p}_2$. Since the symmetry operator is topological, the green line can be deformed without changing anything. This follows from the fact that $\mathrm{d}a = \Phi$ is invariant under $a + \mathrm{d}\lambda$.

to the following bond algebra duality

$$\hat{\mathsf{B}}_{\mathbb{Z}_{n,(d-1)}}(\hat{\mathcal{V}}^{\Phi}_{\text{edge}}) \cong \mathsf{B}^{\Phi}_{\mathbb{Z}_{n,(0)}}(\mathcal{V}) = \left\langle X_{\mathsf{v}}, \tilde{Z}_{\mathsf{s}(\mathsf{e})}\tilde{Z}^{\dagger}_{\mathsf{t}(\mathsf{e})} \,\middle|\, \prod_{\mathsf{e}\in p}\left[\tilde{Z}_{\mathsf{s}(\mathsf{e})}\tilde{Z}^{\dagger}_{\mathsf{t}(\mathsf{e})}\right]^{\mathsf{o}(\mathsf{e},\mathsf{p})} \stackrel{!}{=} \omega_n^{\phi_{\mathsf{p}}}, \, \forall \, \mathsf{v}, \mathsf{e} \right\rangle. \quad (50)$$

In order to understand this better, let us consider $\Phi$ such that $\phi_{\mathsf{p}} = \alpha(\delta_{\mathsf{p},\mathsf{p}_2} - \delta_{\mathsf{p},\mathsf{p}_1})$ or equivalently $B_{\mathsf{p}} = 1$ for all $\mathsf{p} \neq \mathsf{p}_1, \mathsf{p}_2$, while $B_{\mathsf{p}_1} = \omega_n^{-\alpha}$ and $B_{\mathsf{p}_2} = \omega_n^{\alpha}$. In the toric code, this corresponds to the subspace with plaquette-like anyonic excitations on $\mathsf{p}_1$ and $\mathsf{p}_2$. In order to define $\tilde{Z}_{\mathsf{s}(\mathsf{e})}\tilde{Z}^{\dagger}_{\mathsf{t}(\mathsf{e})}$, we need to consider a line $L^{\vee}$ on the dual lattice as in figure 6. We then have[11]

$$\tilde{Z}_{\mathsf{s}(\mathsf{e})}\tilde{Z}^{\dagger}_{\mathsf{t}(\mathsf{e})} = \begin{cases} \omega_n^{\alpha} Z_{\mathsf{s}(\mathsf{e})}Z^{\dagger}_{\mathsf{t}(\mathsf{e})}, & \mathsf{e} \in L, \\ Z_{\mathsf{s}(\mathsf{e})}Z^{\dagger}_{\mathsf{t}(\mathsf{e})}, & \mathsf{e} \notin L, \end{cases} \quad (51)$$

where all the red bonds in figure 6 have a phase $\omega_n^{\alpha}$. This guarantees the correct mapping $B_{\mathsf{p}} = \prod_{\mathsf{e}\in\mathsf{p}} Z_{\mathsf{e}}^{\mathsf{o}(\mathsf{e},\mathsf{p})} \stackrel{!}{=} \omega_n^{\phi_{\mathsf{p}}} \longrightarrow \prod_{\mathsf{e}\in p}\left[\tilde{Z}_{\mathsf{s}(\mathsf{e})}\tilde{Z}^{\dagger}_{\mathsf{t}(\mathsf{e})}\right]^{\mathsf{o}(\mathsf{e},\mathsf{p})} \stackrel{!}{=} \omega_n^{\phi_{\mathsf{p}}}$. This can be understood as the insertion of an open 0-form symmetry surface $\mathcal{U}^{\alpha}[S^{(d)}]$ along time such that it crosses the Hilbert space time-slice along the green curve in figure 6. Equation (51) can be understood as every bond operator $Z_{\mathsf{t}(\mathsf{s})}Z_{\mathsf{t}(\mathsf{e})}$ that crosses the green line, get transformed by $\mathcal{U}^{\alpha}[S^{(d)}]$. This creates two twist defects on the plaquettes $\mathsf{p}_1$ and $\mathsf{p}_2$. Any Hamiltonian constructed from the bond algebra $\mathfrak{B}^{\Phi}_{\mathbb{Z}_{n,(0)}}(\mathcal{V})$, will have extrinsic twist defects on plaquettes where $\Phi$ is not zero. This means that an invertible duality exists for the full uncontrained $(d-1)$-form symmetric bond algebra (43), but the dual algebra is a direct sum of 0-form symmetric bond algebras with all possible twist defect

$$\hat{\mathfrak{B}}_{\mathbb{Z}_{n,(d-1)}}(\mathcal{V}_{\text{edge}}) \cong \bigoplus_{\Phi \in C^2(M_{d,\triangle}, \mathbb{Z}_n)} \mathfrak{B}^{\Phi}_{\mathbb{Z}_{n,(0)}}(\mathcal{V}). \quad (52)$$

This also makes sense as the dimension of the Hilbert space of spins on edges $\mathcal{V}_{\text{edge}}$ is larger than the spins on vertices $\mathcal{V}$.

---

[11]For general $\Phi \in C^2(M_{d,\triangle}, \mathbb{Z}_n)$, we couple to a background gauge field $A$: $\tilde{Z}_{\mathsf{s}(\mathsf{e})}\tilde{Z}^{\dagger}_{\mathsf{t}(\mathsf{e})} = Z_{\mathsf{s}(\mathsf{e})}\omega_n^{A_{\mathsf{e}}}Z^{\dagger}_{\mathsf{t}(\mathsf{e})}$ such that $(\mathrm{d}A)_{\mathsf{p}} = \phi_p$. Thus the presence of twist defects violate the 1-cocycle condition at the locations of the defects.

The bond algebra (22) and (41), their duality to each other and their gapped phases can be directly and systematically derived using a topological order in one higher dimensions using a construction we call Topological Holography [57]. This is closely related to concepts in high-energy physics and string theory, such as symmetry TFTs [20,58–60] or holographic symmetry [61–64]. We will however not go pursue that approach in this paper.

Finally, it is worth mentioning that the full symmetry structure of $\widetilde{\mathfrak{B}}_{\mathbb{Z}_{n,(d-1)}}$ is a higher representation category $d\mathrm{Rep}(\mathbb{Z}_n)$ whose invertible subcategory is generated by $W_L$ for $L \in Z_1(M_{d,\triangle}, \mathbb{Z})$ [29, 30, 44, 45, 66]. The remaining higher dimensional symmetry operators are obtained via condensations of lines corresponding to subgroups of $\mathrm{Rep}(\mathbb{Z}_n) \cong \mathbb{Z}_n$ on sub-manifolds of dimension greater that 1 [26, 46]. In passing, we also comment that gauging non-Abelian (sub) symmetries lead to more interesting symmetry categories which are necessarily non-invertible. In general, in d+1 spatial dimensions, gauging a non-Abelian G 0-form symmetry produces a dual quantum system with a $d\mathrm{Rep}(G)$ symmetry. This contains a non-invertible 1-form subsymmetry $\mathrm{Rep}(G)$ corresponding to topological Wilson lines obtained after gauging G. As in the case for Abelian G, all other symmetry generators in $d\mathrm{Rep}(G)$ are obtainable as condensation defects of the $\mathrm{Rep}(G)$ lines [44, 45].

## 2.2 Gauging finite Abelian sub-symmetry

In this section, we describe the gauging of a subgroup of $\mathbb{Z}_n$. In particular, if $\mathbb{Z}_q \subset \mathbb{Z}_n$, then there is a short exact sequence

$$1 \longrightarrow \mathbb{Z}_q \longrightarrow \mathbb{Z}_n = \mathbb{Z}_{pq} \longrightarrow \mathbb{Z}_{n/q} = \mathbb{Z}_p \longrightarrow 1 . \tag{53}$$

Such a sequence is determined by an extension class $[\epsilon] \in H^2(\mathbb{Z}_p, \mathbb{Z}_q) = \mathbb{Z}_{\gcd(p,q)}$. This means we can think of $\mathbb{Z}_{pq}$ as $\mathbb{Z}_q \times \mathbb{Z}_p$ as a set but with a $\epsilon$-twisted product

$$(a, b) \times (a', b') = (a + a' + \epsilon(b, b'), b + b'), \qquad a, a' \in \mathbb{Z}_q, b, b' \in \mathbb{Z}_p . \tag{54}$$

We will often use the following notation for the $\epsilon$-twisted products: $\mathbb{Z}_{pq} \simeq \mathbb{Z}_q \times_\epsilon \mathbb{Z}_p$. Note that while $(a, 0)$ corresponds to a subgroup $\mathbb{Z}_q \subset \mathbb{Z}_n$, $(0, b)$ is not a $\mathbb{Z}_p$ subgroup. We will see that gauging $\mathbb{Z}_q \subset \mathbb{Z}_n$ furnishes a theory with a $\mathbb{Z}_{n/q}$ global 0-form symmetry, a $(d-1)$-form $\mathbb{Z}_q$ symmetry and a mixed anomaly between them that is determined by $\epsilon$ [65]. Written schematically

$$\mathbb{Z}_{n,(0)} \xrightarrow{\text{gauging } \mathbb{Z}_{q,(0)}} \mathsf{G}^\epsilon_{(0,d-1)} = \left[ \mathbb{Z}_{n/q,(0)}, \mathbb{Z}_{q,(d-1)} \right]^\epsilon , \tag{55}$$

where $\mathsf{G}^\epsilon_{(0,d-1)} = \left[ \mathbb{Z}_{n/q,(0)}, \mathbb{Z}_{q,(d-1)} \right]^\epsilon$ is a (higher) $d$-group consisting of 0-form and $(d-1)$-form symmetries with a mixed anomaly determined by $\epsilon$. It couples to background 1-form and $d$-form gauge fields. In general, anomalies impose strong constraints on the low energy physics realized in a quantum system. For instance, any gapped non-degenerate state can only preserve a sub-symmetry that trivializes the anomaly. We will explore such aspects of anomalies in later sections while studying phase diagrams of spin models with mixed anomalies.

Henceforth, we will use the simplest case with a non-trivial extension class which occurs when $p = q = 2$ to pinpoint lattice manifestations of the mixed anomaly. Although this is the simplest case that exemplifies these features, the lessons learnt can be generalized to any finite Abelian group. Let us again consider the triangulation $M_{d,\triangle}$ endowed with the tensor product of local vector spaces $\mathcal{V}_v = \mathbb{C}^4 \cong \mathbb{C}[\mathbb{Z}_4]$ assigned to each vertex v. The operator algebra acting on $\mathcal{V} = \otimes_v \mathcal{V}_v$ is generated by the operators $X_v$ and $Z_v$ that satisfy the relations

$$Z_v X_{v'} = i X_{v'} Z_v , \qquad Z_v^4 = X_v^4 = 1 . \tag{56}$$

We are interested in the space of Hamiltonians which are symmetric with respect to the $\mathbb{Z}_4$ symmetry generated by $\mathcal{U} = \prod_v X_v$, which is contained within the bond algebra

$$\mathfrak{B}_{\mathbb{Z}_{4,(0)}}(\mathcal{V}) = \left\langle X_v, \, Z_{s(e)}Z_{t(e)}^\dagger \, \middle| \, \forall \, v, e \right\rangle. \tag{57}$$

Similar to (22) in the case of gauging $\mathbb{Z}_n$, we define the bond algebra in a definite symmetry sector as

$$B_{\mathbb{Z}_{4,(0)}}(\mathcal{V};(\alpha, g)) = \left\langle X_v, \, Z_{s(e)}Z_{t(e)}^\dagger \, \middle| \, \mathcal{U} \overset{!}{=} i^\alpha, \, \prod_{e \in \gamma}\left[Z_{s(e)}Z_{t(e)}^\dagger\right]^{o(e,\gamma)} \overset{!}{=} i^{g(\gamma)} \quad \forall \, v, e \right\rangle, \tag{58}$$

where $\alpha$ and g are the symmetry eigen-sector and symmetry twisted boundary condition labels respectively. In order to gauge the $\mathbb{Z}_2$ subgroup of the global symmetry, we introduce $\mathbb{Z}_2$ gauge degrees of freedom on the edges such that the extended Hilbert space is

$$\mathcal{V}_{\text{ext}} = \bigotimes_e \mathbb{C}_e^2 \bigotimes_v \mathbb{C}_v^4. \tag{59}$$

There is an action of Pauli operators $\sigma_e^\mu$ ($\mu = 0, x, y, z$) on the edge Hilbert space $\mathcal{V}_e$. We define a basis $\{|a, \phi\rangle\}$ that spans $\mathcal{V}_{\text{ext}}$, where $\phi \in C^0(M_{d,\triangle}, \mathbb{Z}_4)$ and $a \in C^1(M_{d,\triangle}, \mathbb{Z}_2)$ such that

$$\begin{aligned}
Z_v|a, \phi\rangle &= i^{\phi_v}|a, \phi\rangle, & X_v|a, \phi\rangle &= |a, \phi + \delta^{(v)}\rangle, \\
\sigma_e^z|a, \phi\rangle &= (-1)^{a_e}|a, \phi\rangle, & \sigma_e^x|a, \phi\rangle &= |a + \delta^{(e)}, \phi\rangle,
\end{aligned} \tag{60}$$

where $\delta^{(v)} \in C^0(M_{d,\triangle}, \mathbb{Z}_4)$ and $\delta^{(e)} \in C^1(M_{d,\triangle}, \mathbb{Z}_2)$ are defined in (25) and the addition is implicitly modulo 4 or modulo 2 depending on the group the cochains involved are valued in. The physical Hilbert space is the gauge-invariant subspace of $\mathcal{V}_{\text{ext}}$. The notion of gauge invariance follows from considering the local representative of the $\mathbb{Z}_2$ symmetry being gauged (generated by $\mathcal{U}^2$) and appending it with link operators, i.e.,

$$\mathcal{G}_v = X_v^2 \prod_{e \supset v} \sigma_e^x =: X_v^2 A_v, \tag{61}$$

where $\prod_{e \supset v}$ denotes the product over edges which are connected to the vertex v. A general Gauss operator $\mathcal{G}[\lambda] = \prod_v \mathcal{G}_v^{\lambda_v}$ with $\lambda \in C^0(M_{d,\triangle}, \mathbb{Z}_2)$ acts on the above mentioned basis spanning $\mathcal{V}_{\text{ext}}$ as a $\mathbb{Z}_2$ gauge transformation

$$\mathcal{G}[\lambda]|a, \phi\rangle = |a - d\lambda, \phi + 2\lambda\rangle, \tag{62}$$

or on the level of operators

$$Z_v \longrightarrow \mathcal{G}[\lambda]Z_v\mathcal{G}[\lambda]^\dagger = (-1)^{\lambda_v}Z_v, \qquad \sigma_e^z \longrightarrow \mathcal{G}[\lambda]\sigma_e^z\mathcal{G}[\lambda]^\dagger = (-1)^{d\lambda_e}Z_e. \tag{63}$$

To gauge the bond algebra, (57), we lift $\mathfrak{B}_{\mathbb{Z}_4}(\mathcal{V})$ to the enlarged $\mathcal{V}_{\text{ext}}$, impose the Gauss constraint and consider gauge-invariant versions of each of the operators. Doing so, we find

$$\begin{aligned}
\mathfrak{B}_{G_{(0,d-1)}^\epsilon}(\mathcal{V}_{\text{ext}}) &\simeq \mathfrak{B}_{\mathbb{Z}_4}/\mathbb{Z}_2 \\
&= \left\langle X_v, \, Z_{s(e)}\sigma_e^z Z_{t(e)}^\dagger \, \middle| \, \mathcal{G}_v \overset{!}{=} 1, \, \prod_{e \in L}\left[Z_{s(e)}\sigma_e^z Z_{t(e)}^\dagger\right]^{o(e,L)} \overset{!}{=} 1, \, \forall \, v, e \right\rangle,
\end{aligned} \tag{64}$$

where $L$ are contractible loops on the direct lattice and we have defined the (higher) $d$-group

$$G_{(0,d-1)} = \left[\mathbb{Z}_{2,(0)}, \, \mathbb{Z}_{2,(d-1)}\right]^\epsilon. \tag{65}$$

The constraint on $\prod_{e \in L}\left[Z_{s(e)}\sigma_e^z Z_{t(e)}^\dagger\right]^{o(e,L)}$ descends from the fact that this operator is the image of $\prod_{e \in L}\left[Z_{s(e)}Z_{t(e)}^\dagger\right]^{o(e,L)} = 1$ under the bond algebra isomorphism. We could impose additional constraints

$$\prod_v X_v \overset{!}{=} i^\alpha, \qquad \prod_{e \in L}\left[Z_{s(e)}\sigma_e^z Z_{t(e)}^\dagger\right]^{o(e,L)} \overset{!}{=} i^{g(L)}, \tag{66}$$

which allow us to track how the symmetry sectors map under partial gauging. To understand this we need to ask, what these sectors mean in terms of the symmetry structure of the partially gauged bond algebra.

**Symmetries of the partially gauged bond algebra:** There are two types of operators that commute with the entire algebra (64). Firstly, since we have gauged $\mathbb{Z}_2 \subset \mathbb{Z}_4$, we expect there to still be a residual $\mathbb{Z}_2$ 0-form symmetry, which is generated by an operator which acts on all of space $M_{d,\triangle}$ simultaneously via the operator

$$\mathcal{U} = \prod_v X_v. \tag{67}$$

At first glance, this may look like a $\mathbb{Z}_4$ symmetry generator as $X_v^4 = 1$. However since the $\mathbb{Z}_2$ subgroup has been gauged, $\mathcal{U}$ actually generates a $\mathbb{Z}_2$ symmetry. Although, as is evident from (66), $\mathcal{U}^2$ is not always 1 as one would expect for a usual $\mathbb{Z}_2$ symmetry. This peculiarity is rooted in the fact that the $\mathbb{Z}_2$ 0-form symmetry participates in a mixed anomaly with the second kind of symmetry, which is a $\mathbb{Z}_2$ $(d-1)$-form symmetry generated by the following operator defined on a non-contractible 1-cycle $\gamma$

$$W_\gamma = \prod_{e \in \gamma} w_e^{o(e,\gamma)}, \qquad w_e = Z_{s(e)}\sigma_e^z Z_{t(e)}^\dagger. \tag{68}$$

Notice that the local representative of the line operator cannot be the naive choice $\sigma_e^z$ since it is not gauge invariant. Furthermore $W_L$, for a contractible 1-cycle $L$, acts as the identity on the constrained (flux-free) Hilbert space. Just like $\mathcal{U}$, it is unusual to think of the line operators $W_\gamma$ as $(d-1)$-form $\mathbb{Z}_2$ generators since they do not square to the identity depending on g in (66).

In order to clarify the mixed anomaly, we need to define operators that measure the symmetry twisted boundary conditions for the 0-form and $(d-1)$-form symmetries. As discussed in Sec. 2.1, symmetry twisted boundary conditions with respect to the 0-form symmetry (STBC$^{(0)}$) are related to the holonomy $g(\gamma)$ of a 1-form $\mathbb{Z}_2$ gauge field around any non-contractible 1-cycle $\gamma$. Likewise the symmetry twisted boundary conditions with respect to the $d-1$-form symmetry (STBC$^{(d-1)}$) correspond to a holonomy $\alpha$ of a $d$-form $\mathbb{Z}_2$ background gauge field around the fundamental $d$-cycle, i.e. around all of space. These holonomies are measured by the operators $\mathcal{T}_\gamma^{(0)}$ and $\mathcal{T}^{(d-1)}$

$$\mathcal{T}_\gamma^{(0)} = \prod_{e \in \gamma} Z_{s(e)}^2 Z_{t(e)}^2 = (-1)^{g(\gamma)}, \qquad \mathcal{T}^{(d-1)} = \prod_v A_v = (-1)^\alpha. \tag{69}$$

A manifestation of the mixed-anomaly is that the 0-form $\mathbb{Z}_2$ symmetry is fractionalized to $\mathbb{Z}_4$ in the symmetry twisted sector of the $(d-1)$-form symmetry and conversely, the $(d-1)$-form symmetry is fractionalized to a $\mathbb{Z}_4$ symmetry in the symmetry twisted sector of the 0-form symmetry (see Fig. 7)

$$\mathcal{U}^2 = \mathcal{T}^{(d-1)}, \qquad W_\gamma^2 = \mathcal{T}_\gamma^{(0)}. \tag{70}$$

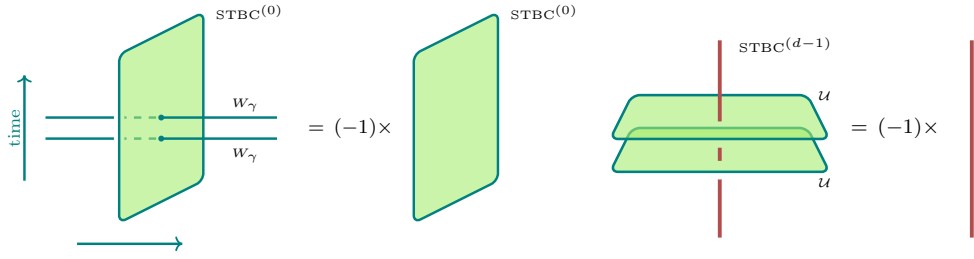

Figure 7: Figure (a) depicts the equation $W_\gamma^2 = \mathcal{T}_\gamma^{(0)}$ in terms of symmetry defects. The symmetry twisted boundary conditions for $\mathcal{T}_\gamma^{(0)}$ is implemented by a codimension-1 operator (in green) that extends in the $d-1$ homology dual to $\gamma$. Figure (b) depicts the $\mathcal{U}^2 = \mathcal{T}^{(d-1)}$ in terms of symmetry defects.

Another related consequence of the mixed anomaly is the symmetry fractionalization on the local representative of a $(d-1)$-form symmetry generator. More precisely, the operator $W_L$ defined on an open line $L$, takes the form

$$W_L = Z_{\mathsf{s}(L)} \left[ \prod_{\mathsf{e} \in L} \sigma_{\mathsf{e}}^z \right] Z_{\mathsf{t}(L)}^\dagger. \tag{71}$$

Notice, that the end-points of the string are now appended with operators that are charged under $\mathcal{U}$ and in fact carry a fractional charge, i.e., the $\mathcal{U}^2$ eigenvalue is $-1$ for such an operator (see Fig. 9). This is the phenomena of symmetry fractionalization [7, 102–105] and is related to a mixed anomaly between the 0-form symmetry and $(d-1)$-form symmetry as we will describe in more detail in the next section.

**Solving the Gauss constraint:** Let us now try to find a unitary that disentangles the edge degrees of freedom from the Gauss operator such that the Gauss constraint may be solved. Specifically, we seek an operator $U$ such that

$$U \, \mathcal{G}_{\mathsf{v}} \, U^\dagger = X_{\mathsf{v}}^2. \tag{72}$$

We make an Ansatz that the action is a generalized controlled operation, i.e., it acts as

$$U = \sum_{a,\phi} |a + f(\phi), \phi\rangle\langle a, \phi|. \tag{73}$$

Inserting the Ansatz (73) into (72), one finds the constraint $f(\phi) + f(\phi + 2\delta^{(\mathsf{v})}) = \mathrm{d}\delta^{(\mathsf{v})}$, which can be solved by $f = \mathrm{d}\lfloor \phi/2 \rfloor$, where $\lfloor \cdot \rfloor$ denoted the floor function. Therefore the unitary is

$$U = \sum_{a,\phi} |a + \mathrm{d}\lfloor \phi/2 \rfloor, \phi\rangle\langle a, \phi|. \tag{74}$$

Using (74), the action on all the remaining operators in the bond algebra can be computed. For instance $\sigma_{\mathsf{e}}^x$ and $Z_{\mathsf{v}}$ remain invariant under the action of $U$. Meanwhile $X_{\mathsf{v}}$ and $\sigma_{\mathsf{e}}^z$ transform in a more involved way

$$
\begin{aligned}
U X_{\mathsf{v}} U^\dagger &= \sum_{a,\phi} \left| a + \mathrm{d}(\lfloor \phi/2 \rfloor + \lfloor (\phi + \delta^{(\mathsf{v})})/2 \rfloor), \phi + \delta^{(\mathsf{v})} \right\rangle\langle a, \phi| \\
&= X_{\mathsf{v}} \left[ P_{\mathsf{v}}^{(+)} + A_{\mathsf{v}} P_{\mathsf{v}}^{(-)} \right], \\
U Z_{\mathsf{s}(\mathsf{e})} \sigma_{\mathsf{e}}^z Z_{\mathsf{t}(\mathsf{e})}^\dagger U^\dagger &= \sum_{a,\phi} \exp\left\{ i\pi (a + \mathrm{d}\lfloor \phi/2 \rfloor)_{\mathsf{e}} + \frac{i\pi(\mathrm{d}\phi)_{\mathsf{e}}}{2} \right\} |a, \phi\rangle\langle a, \phi| \\
&= \frac{1}{2} \left( 1 - iZ_{\mathsf{s}(\mathsf{e})}^2 \right) \sigma_{\mathsf{e}}^z \left( 1 + iZ_{\mathsf{t}(\mathsf{e})}^2 \right),
\end{aligned}
\tag{75}
$$

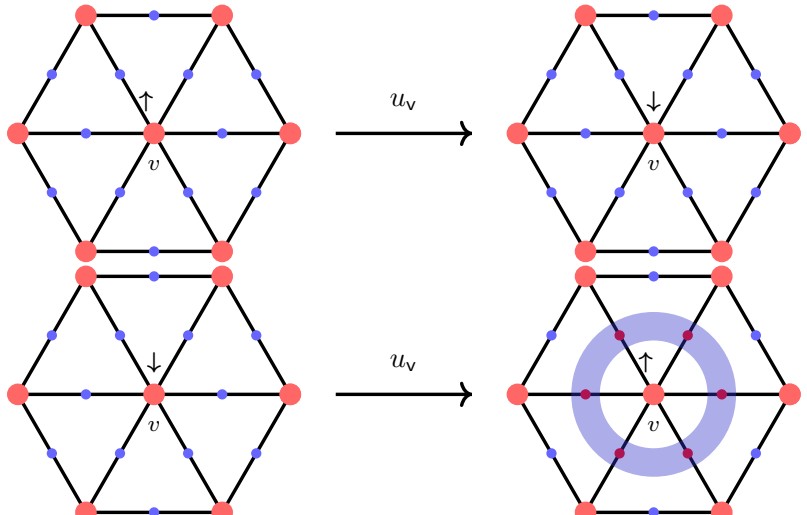

Figure 8: The operator $u_v$ acts in two steps. In the first step, it measures the spin at vertex $v$ and implements a gauge transformation (via $A_v$) on the neighboring edges depending on the measurement outcome. In the second step, it flips the spin at $v$.

where we have defined $P_v^{(\pm)} = (1 \pm Z_v^2)/2$. We are now in a position to write down the unitary transformed bond algebra

$$\widetilde{\mathfrak{B}}_{\mathsf{G}_{(0,d-1)}}(\mathcal{V}_{\mathrm{ext}}) = \left\langle X_v \left[ P_v^{(+)} + A_v P_v^{(-)} \right], \, \frac{1 - iZ_{s(e)}^2}{\sqrt{2}} \sigma_e^z \frac{1 + iZ_{t(e)}^2}{\sqrt{2}} \, \middle| \, X_v^2 \overset{!}{=} 1, \, \prod_{e \in L} \sigma_e^z \overset{!}{=} 1, \, \forall v, e \right\rangle. \quad (76)$$

Since the constraint has now been localized on the vertices, it can be readily solved. We define a restricted basis on the vertex Hilbert space $\mathcal{V}_v^{\mathrm{rest.}} \subset \mathcal{V}_v$ spanned by

$$
\begin{aligned}
|\uparrow\rangle &= \frac{1}{\sqrt{2}} \left\{ |\phi_v = 0\rangle + |\phi_v = 2\rangle \right\}, \\
|\downarrow\rangle &= \frac{1}{\sqrt{2}} \left\{ |\phi_v = 1\rangle + |\phi_v = 3\rangle \right\},
\end{aligned}
\quad (77)
$$

for which $X_v^2 = 1$. The operators $X_v$ and $Z_v^2$ acting on $\mathcal{V}_v$ can be restricted to $\mathcal{V}_v^{\mathrm{rest.}}$ since these operators commute with $X_v^2$ and therefore leave the space spanned by 77 invariant. In this basis,

$$X_v \Big|_{\mathcal{V}_v^{\mathrm{rest.}}} \sim \sigma_v^x, \qquad Z_v^2 \Big|_{\mathcal{V}_v^{\mathrm{rest.}}} \sim \sigma_v^z. \quad (78)$$

Since only combinations of $X_v$ and $Z_v^2$ appear in the bond algebra in (76), we can solve the Gauss constraint and directly work in the restricted Hilbert space

$$\widetilde{\mathfrak{B}}_{\mathsf{G}_{(0,d-1)}}(\mathcal{V}^{\mathrm{rest.}}) = \left\langle \sigma_v^x \left[ P_v^{(+)} + A_v P_v^{(-)} \right], \, \frac{1 - i\sigma_{s(e)}^z}{\sqrt{2}} \sigma_e^z \frac{1 + i\sigma_{t(e)}^z}{\sqrt{2}} \, \middle| \, \prod_{e \in L} \sigma_e^z \overset{!}{=} 1 \, \forall v, e \right\rangle, \quad (79)$$

where $\mathcal{V}^{\mathrm{rest.}} = \bigotimes_v \mathcal{V}_v^{\mathrm{rest.}} \bigotimes_e \mathcal{V}_v \subset \mathcal{V}_{\mathrm{ext.}}$ and $P_v^{(\pm)} := (1 \pm \sigma_v^z)/2$ on $\mathcal{V}^{\mathrm{res.}}$.

**Symmetry and mixed anomaly for the transformed bond algebra:** Since the bond algebras (64) and (79) are isomorphic, they have identical symmetry structures. In the frame of $\widetilde{\mathfrak{B}}_{\mathsf{G}_{(0,d-1)}}$, the $\mathbb{Z}_2$ 0-form symmetry is generated by

$$\mathcal{U} = \prod_v u_v, \qquad u_v = \sigma_v^x \left[ P_v^{(+)} + A_v P_v^{(-)} \right], \quad (80)$$

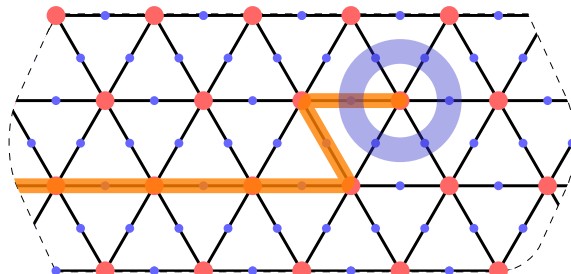

Figure 9: The end-point of a 1-form symmetry $W(L)$ traps an operator which carries a fractional charge under the 0-form symmetry. This symmetry fractionalization is a diagnostic of a mixed anomaly between the 0-form and 1-form symmetry.

while the $(d-1)$-form symmetry generated by the (closed) line operator

$$W_\gamma = \prod_{e \in \gamma} w_e^{o(e,\gamma)}, \qquad w_e = \frac{1}{2}\left(1 - i\sigma^z_{s(e)}\right)\sigma^z_e\left(1 + i\sigma^z_{t(e)}\right). \tag{81}$$

These two symmetries have a mixed anomaly, which manifests as

$$\begin{aligned} \mathcal{U}^2 &= \prod_v A_v = \mathcal{T}^{(d-1)}, \\ W_\gamma^2 &= \prod_{e \in \gamma} \sigma^z_e = \mathcal{T}^{(0)}_\gamma, \end{aligned} \tag{82}$$

and reproduces (70). In the next sections, we will see that this anomaly has important consequences for the phase realized in the partially gauged model.

# 3 Gauging as topological dualities

In this section, we describe a general procedure [65] to gauge a finite global symmetry and its sub-groups from a space-time point of view deriving relations between partition functions and energy spectra of dual theories. We will pay attention to the global symmetry of the gauged theory thus obtained and to the mapping of the symmetry twisted sectors between the gauged and original theory. As we described in Sec. 2, when gauging a finite subgroup of the full symmetry group, the dual or gauged theory has a symmetry structure with a mixed anomaly. In such cases, the mapping of symmetry sectors can be subtle. We detail how this works for the simplest case of gauging $\mathbb{Z}_2 \subset \mathbb{Z}_4$, which contains the main new result of this section. Although, we describe this specific simplest case, our analysis and approach generalize to other finite Abelian groups.

## 3.1 Gauging finite Abelian symmetry

We begin by describing the gauging of a finite Abelian symmetry group in a $d+1$ dimensional quantum system $\mathfrak{T}$, before addressing its subgroups. Unlike the previous analysis working on the level of Hilbert spaces of spin models, here we take a space-time point of view and work on the level of partition functions.

### 3.1.1 Gauging, ungauging and dual symmetries

Let us consider a quantum system, denoted by $\mathfrak{T}$ in $d+1$ spacetime dimensions and symmetric under a finite Abelian group $G_{(0)}$. Such a theory can be defined in the presence of

a background $G_{(0)}$ gauge field $A_1$, which can equivalently be understood as a network of codimension-1 (in spacetime) symmetry defects. We denote the partition function of $\mathfrak{T}$ coupled to a $A_1$ background by $\mathcal{Z}_{\mathfrak{T}}[A_1]$. If the theory $\mathfrak{T}$ does not have any 't Hooft anomaly with respect to the group $G_{(0)}$, then $G_{(0)}$ can be gauged in $\mathfrak{T}$ to obtain a new theory $\mathfrak{T}^{\vee}$. Gauging the 0-form symmetry amounts to summing over background gauge fields/symmetry defect networks [18, 26, 119, 120]. The partition function of the gauged theory has the form

$$\mathcal{Z}_{\mathfrak{T}^{\vee}} = \frac{1}{|G|^{b_0(M)}} \sum_{a_1} \mathcal{Z}_{\mathfrak{T}}[a_1], \tag{83}$$

where we have assumed that $M$ is path connected and the sum is over gauge classes of G-bundles, i.e., $a_1 \in H^1(M, G)$. Here $b_n(M)$ is the $n$'th Betti number of $M$. The theory $\mathfrak{T}^{\vee}$ has a $(d-1)$ form symmetry $G^{\vee} = \text{hom}(G, \mathbb{R}/2\pi\mathbb{Z}) \cong G$ [18, 65, 121], denoted by $G^{\vee}_{(d-1)}$. The symmetry operator corresponding to an element $g^{\vee} \in G^{\vee}$ defined on a 1-cycle $\gamma$ is

$$W_{g^{\vee}}(\gamma) = \exp\left\{ i g^{\vee} \oint_{\gamma} a_1 \right\}. \tag{84}$$

We can couple $\mathfrak{T}^{\vee}$ to a background $d$-form gauge field $A_d^{\vee}$, which corresponds to inserting a network of line-like symmetry operators. The partition function of $\mathfrak{T}^{\vee}$ in the presence of a $A_d^{\vee}$ is given by

$$\mathcal{Z}_{\mathfrak{T}^{\vee}}[A_d^{\vee}] = \frac{1}{|G|^{b_0(M)}} \sum_{a_1} \mathcal{Z}_{\mathfrak{T}}[a_1] \exp\left\{ i \int_M a_1 \cup A_d^{\vee} \right\}. \tag{85}$$

Gauge invariance of the partition function under background gauge transformations of $A_d^{\vee}$ is guaranteed by the fact that $da_1 = 0$. The $G^{\vee}_{(d-1)}$ global symmetry of the gauged theory $\mathfrak{T}^{\vee}$ can itself be gauged to deliver a theory $\mathfrak{T}^{\vee\vee}$ which again has a symmetry $G_{(0)}$

$$\mathcal{Z}_{\mathfrak{T}^{\vee\vee}}[A_1] = \frac{1}{|G^{\vee}|^{N_d(M)}} \sum_{a_d^{\vee}} \mathcal{Z}_{\mathfrak{T}^{\vee}}[a_d^{\vee}] \exp\left\{ i \int_M a_d^{\vee} \cup A_1 \right\}, \tag{86}$$

where $N_p(M) = \sum_{j=1}^{p} (-1)^{j+1} b_{p-j}(M)$. Note that $N_1 = b_0(M)$, which recovers (85). Then inserting (85) into (86), one obtains

$$\mathcal{Z}_{\mathfrak{T}^{\vee\vee}}[A_1] = \exp\{-\chi(M)\ln(|G|)\} \times \mathcal{Z}_{\mathfrak{T}}[(-1)^d A_1], \tag{87}$$

where $\chi(M) = \sum_{j=0}^{d+1} (-1)^j b_j(M)$ is the Euler characteristic of the manifold $M$. Here we used the relation

$$\delta(A_{d+1-n}) = \frac{1}{|G|^{b_n(M)}} \sum_{a_n \in H^n(M, G)} \exp\left\{ i \int_M a_n \cup A_{d+1-n} \right\}. \tag{88}$$

Hence gauging twice acts as "charge conjugation" in odd spatial dimensions [122] upto a local curvature counterterm. In fact, the so-called Euler counterterm in (87) can be absorbed by redefining the normalization as $N_p = \sqrt{|G|^{b_p(M)}}$ in (85) and (86). We will however work with our initial choice of normalization as doing so simplifies the mapping of symmetry sectors between the original $\mathfrak{T}$ and the gauged theory $\mathfrak{T}^{\vee}$.

### 3.1.2 Mapping of symmetry sectors

Using (85) and (86), one can see how the symmetry sectors on the different sides of the gauging-related duality map into each other. Let us consider a $d+1$ dimensional spacetime manifold that decomposes as $M = S^1 \times M_d$, where $M_d$ is the spatial $d$-manifold and $S^1$ is

the circle in the time direction, such as to connect with the Hamiltonian description in the rest of the paper. We work with imaginary time and will therefore be considering thermal partition functions. A G background gauge field $A_1$ (upto gauge transformations) is valued in $H^1(M, \mathsf{G})$, i.e., it is labelled by the holonomies of the gauge field $A_1$ along the homology cycles of $M$. Using the Künneth theorem, the 1st homology group of $M = S^1 \times M_d$ decomposes as

$$H_1(S^1 \times M_d, \mathbb{Z}) = \mathbb{Z} \oplus H_1(M_d, \mathbb{Z}) = \mathrm{Span}_{\mathbb{Z}}\langle \widetilde{\gamma}, \gamma_1, \gamma_2, \dots, \gamma_{b_1(M_d)} \rangle. \tag{89}$$

Then the gauge field $A_1$ can be labelled by its holonomies around the homology cycles of $M$ as $A_1 = (\mathsf{g}_t, \vec{\mathsf{g}})$ where $\vec{\mathsf{g}} \equiv (\mathsf{g}_1, \mathsf{g}_2, \dots, \mathsf{g}_{b_1(M_d)}) \in H^1(M_d, \mathsf{G})$ and $\mathsf{g}_t \in \mathsf{G}$, i.e.,

$$\oint_{\gamma_j} A_1 = \mathsf{g}_j, \qquad \oint_{\widetilde{\gamma}} A_1 = \mathsf{g}_t. \tag{90}$$

The thermal partition function of a quantum system coupled to such a 0-form background gauge field is

$$\mathcal{Z}_{\mathfrak{T}}[A_1] \equiv \mathcal{Z}_{\mathfrak{T}}[\mathsf{g}_t, \vec{\mathsf{g}}] = \mathrm{Tr}\left[ \mathcal{U}_{\mathsf{g}_t} e^{-\beta H_{\vec{\mathsf{g}}}} \right], \tag{91}$$

where $\mathcal{U}_{\mathsf{g}_t}$ is the symmetry operator corresponding to $\mathsf{g}_t$ and $H_{\vec{\mathsf{g}}}$ is the Hamiltonian of interest with $\mathsf{g}_j$ twisted boundary conditions along the $j^{\text{th}}$ homology cycle of $M_d$. In Sec. 2, we could study the G symmetric bond-algebra in a definite symmetry sector labelled by $\alpha \in \mathrm{Rep}(\mathsf{G})$ and $\vec{\mathsf{g}} \in H^1(M_d, \mathsf{G})$ (see (22)). Physically the label $\alpha$ denoted the symmetry eigenspace we were restricting to and $\vec{\mathsf{g}}$ was a choice of symmetry twisted boundary conditions. In the spacetime partition function, this amounts to inserting a (codimension-1) projection operator $\mathcal{P}_\alpha$ at a fixed time, extending over all of $M_d$, which has the form

$$\mathcal{P}_\alpha = \frac{1}{|\mathsf{G}|} \sum_{\mathsf{g}_t \in \mathsf{G}} \mathsf{R}_\alpha(\mathsf{g}_t)^{-1} \mathcal{U}_{\mathsf{g}_t}, \tag{92}$$

where $\mathsf{R}_\alpha : \mathsf{G} \to \mathsf{U}(1)$ is the representation corresponding to the label $\alpha$. We define a symmetry character $\chi_{\mathfrak{T}}[\alpha, \vec{\mathsf{g}}]$ as the thermal trace in the sector transforming in the $\alpha$ representation and with $\vec{\mathsf{g}}$ twisted boundary conditions

$$\chi_{\mathfrak{T}}[\alpha, \vec{\mathsf{g}}] = \mathrm{Tr}_{\mathcal{V}_{M_d}}\left[ \mathcal{P}_\alpha e^{-\beta H_{\vec{\mathsf{g}}}} \right]. \tag{93}$$

After gauging, we obtain a dual theory $\mathfrak{T}^\vee$ which can be coupled to a $d$-form gauge field $A_d \in H^d(M, \mathsf{G}) = \mathrm{Hom}(H_d(M, \mathbb{Z}), \mathsf{G})$. Again, using the Künneth theorem,

$$\begin{aligned} H_d(S^1 \times M_d, \mathbb{Z}) &= H_{d-1}(M_d, \mathbb{Z}) \oplus H_d(M_d, \mathbb{Z}) \\ &\cong H_1(M_d, \mathbb{Z}) \oplus \mathbb{Z}. \end{aligned} \tag{94}$$

In the second line we have used the assumption that $M_d$ is closed and oriented which via the Universal coefficient theorem and Poincaré duality implies $H_{d-1}(M_d, \mathbb{Z}) = H_1(M_d, \mathbb{Z})$ and the fact that $M_d$ is path connected, which implies $H_d(M_d, \mathbb{Z}) \cong \mathbb{Z}$. More precisely we can canonically identify the generators $\Sigma_j^{(d-1)} \in H_{d-1}(M_d, \mathbb{Z})$ with the generators $\gamma_j \in H_1(M_d, \mathbb{Z})$. The $d$-form background gauge field $A_d$ can be labelled by its holonomies around the $d$-homology cycles of $M$ as $A_d = (\vec{\mathsf{g}}^\vee, \mathsf{g}_t^\vee)$ such that $\mathsf{g}_t^\vee \in \mathsf{G}^\vee$ and $\vec{\mathsf{g}}^\vee = (\mathsf{g}_1^\vee, \dots, \mathsf{g}_{b_1(M_d)}^\vee)$ such that

$$\oint_{M_d} A_d = \mathsf{g}_t^\vee, \qquad \oint_{\Sigma_j^{(d-1)} \times S^1} A_d = \mathsf{g}_j^\vee. \tag{95}$$

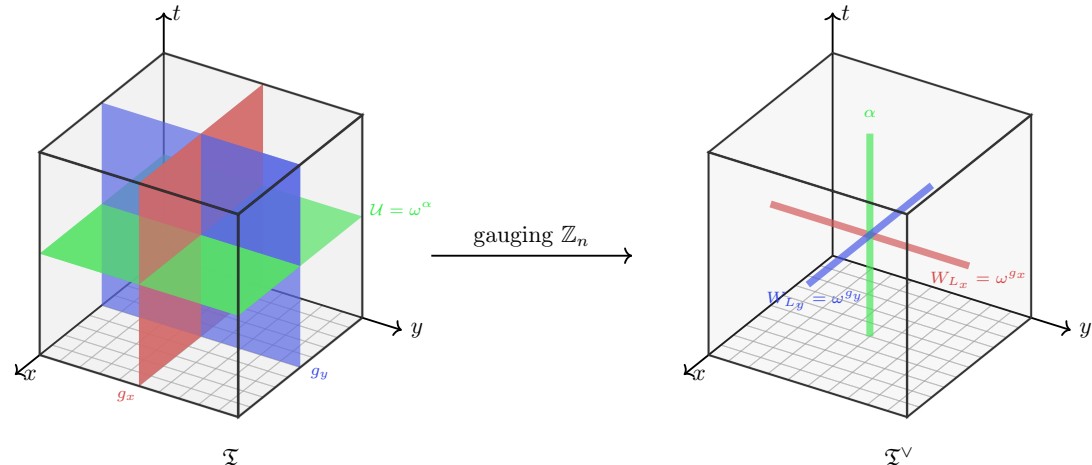

Figure 10: The figure illustrates a spacetime $M = S^1 \times T^2 = T^3$ where all space and time directions are periodic. The left figure depicts the insertion of 0-form symmetry operators (surfaces) in theory $\mathfrak{T}$. The red and blue surfaces along $x$-$t$ and $y$-$t$ planes correspond to twisting the boundary conditions along the $x$ and $y$ direction. The green surface on a time slice is the projection operator (92), projecting to the symmetry sector $\mathcal{U} = \omega^\alpha$. The insertion of these in the partition function gives rise to the character $\chi_{\mathfrak{T}}[\alpha, \vec{\mathsf{g}}]$ (see (93)). This contains the energy spectrum of the theory with twisted boundary conditions $\vec{\mathsf{g}}$ and in symmetry sector $\alpha$. After gauging the 0-form $\mathbb{Z}_n$ symmetry we obtain the theory $\mathfrak{T}^\vee$ and the relation (99) between the characters of the two theories. Here $\mathfrak{T}^\vee$ has a $(d-1)$-form symmetry (lines). The right-hand side illustrates the dual characters after gauging. The characters are equal, but the labels for twisted boundary conditions and symmetry sector has swapped.

The thermal partition function of the quantum system $\mathfrak{T}^\vee$ coupled to $A_d$ background has the form

$$\mathcal{Z}_{\mathfrak{T}^\vee}[A_d^\vee] = \text{Tr}\left[\prod_{j=1}^{b_1(M_d)} W_{\mathsf{g}_j^\vee}(\gamma_j) e^{-\beta H_{\mathsf{g}_t^\vee}^\vee}\right], \tag{96}$$

where $W_{\mathsf{g}_j^\vee}(\gamma_j)$ is the $(d-1)$-form symmetry operator (84) defined on the homology 1-cycle $\gamma_j$ and $H^\vee$ is a Hamiltonian of interest with $\mathsf{G}_{(d-1)}^\vee$-form global symmetry and $\mathsf{g}_t^\vee$ twisted boundary conditions. As before, we can project onto definite $\mathsf{G}^\vee$ eigenspaces of each of the line operators by using the projection operator

$$\mathcal{P}_{\mathsf{g}}(\gamma_j) = \frac{1}{|\mathsf{G}|} \sum_{\mathsf{g}_j^\vee \in \mathsf{G}^\vee} \mathsf{R}_{\mathsf{g}_j^\vee}(\mathsf{g})^{-1} W_{\mathsf{g}_j^\vee}(\gamma_j). \tag{97}$$

We define the symmetry character as a thermal trace in a definite symmetry sector with $\mathsf{g}_t^\vee \in \mathsf{G}^\vee$ twisted boundary conditions and $\mathsf{g}_j$ eigenvalues of the symmetry operator on the 1-cycle $\gamma_j$

$$\chi_{\mathfrak{T}^\vee}[\vec{\mathsf{g}}, \mathsf{g}_t^\vee] = \text{Tr}_{\mathcal{V}_{M_d}^\vee}\left[\prod_{j=1}^{b_1(M_d)} \mathcal{P}_{\mathsf{g}_j}(\gamma_j) q^{-\beta H_{\mathsf{g}_t^\vee}^\vee}\right]. \tag{98}$$

Using (85), (86), (93) and (98) it can be shown that

$$\chi_{\mathfrak{T}}[\alpha, \vec{\mathsf{g}}] = \chi_{\mathfrak{T}^\vee}[\vec{\mathsf{g}}, \alpha], \tag{99}$$

implying a duality between the corresponding sectors of the theory $\mathfrak{T}$ with $\mathsf{G}_{(0)}$ and theory $\mathfrak{T}^\vee$ with global symmetry $\mathsf{G}_{(d-1)}^\vee$. See figure 10 for more details.

### 3.2 Gauging finite Abelian sub-symmetry

#### 3.2.1 Dual symmetries and mixed anomaly

We now describe the more interesting case, where the full symmetry group $G$ of $\mathfrak{T}$, is a central extension of $K$ by $N$ [18,45,65]. More precisely, $G$ sits in the short exact sequence

$$1 \longrightarrow N \longrightarrow G \longrightarrow K \longrightarrow 1\,, \tag{100}$$

whose extension class is $\epsilon \in H^2(K, N)$. This implies that a $G$ background gauge field $A_1$ can be represented as a tuple $\left(A_1^{(N)}, A_1^{(K)}\right) \in C^1(M, N) \times C^1(M, K)$ which satisfy the modified cocyle conditions [65]

$$dA_1^{(N)} = \epsilon\left(A_1^{(K)}\right)\,, \qquad dA_1^{(K)} = 0\,. \tag{101}$$

Correspondingly, the partition function of $\mathfrak{T}$ coupled to a $G$ gauge field is denoted by $\mathcal{Z}_{\mathfrak{T}}\left[A_1^{(N)}, A_1^{(K)}\right]$. Recall the notation $G = N \times_\epsilon K$ from section 2.2. Instead of gauging the full group $G$, consider gauging $N \subset G$, using Eq. (85) to obtain the partially gauged theory $\mathfrak{T}^\vee$. The gauged theory has a residual 0-form symmetry $(G/N)_{(0)} \cong K_{(0)}$. Additionally, there is also a dual $(d-1)$-form symmetry $N_{(d-1)}^\vee$. Hence after the partial gauging, the resulting symmetry is a higher $d$-group $G_{(0,d-1)}^\epsilon$

$$G_{(0)} = N_{(0)} \times_\epsilon K_{(0)} \xrightarrow{\text{gauging } N_{(0)}} G_{(0,d-1)}^\epsilon = \left[K_{(0)},\ N_{(d-1)}^\vee\right]^\epsilon\,. \tag{102}$$

The partition function of the gauged theory can be coupled to background gauge fields of $G_{(0,d-1)}^\epsilon = \left[K_{(0)},\ N_{(d-1)}^\vee\right]^\epsilon$ as

$$\mathcal{Z}_{\mathfrak{T}^\vee}\left[A_d^{(N^\vee)}, A_1^{(K)}\right] = \frac{1}{|H|} \sum_{a_1^{(N)}} \mathcal{Z}_{\mathfrak{T}}\left[a_1^{(N)}, A_1^{(K)}\right] \exp\left\{i \int_M a_1^{(N)} \cup A_d^{(N^\vee)}\right\}\,. \tag{103}$$

Interestingly, the theory $\mathfrak{T}^\vee$ has a mixed 't Hooft anomaly between $K_{(0)}$ and $N_{(d-1)}^\vee$ which manifests in the lack of invariance of the partition function under background gauge transformations

$$A_d^{(N^\vee)} \longmapsto A_d^{(N^\vee)} + d\lambda_{d-1}^{(N^\vee)}\,, \tag{104}$$

where $\lambda_{d-1}^{(N^\vee)} \in C^{d-1}(M, N^\vee)$ is a gauge transformation parameter. Under such a background gauge transformation, the partition function (103) transforms as

$$\frac{\mathcal{Z}_{\mathfrak{T}^\vee}\left[A_d^{(N^\vee)} + d\lambda_{d-1}^{(N^\vee)}, A_1^{(K)}\right]}{\mathcal{Z}_{\mathfrak{T}^\vee}\left[A_d^{(N^\vee)}, A_1^{(K)}\right]} = \exp\left\{i \int_M \lambda_{d-1}^{(N^\vee)} \cup \epsilon\left(A_1^{(K)}\right)\right\}\,. \tag{105}$$

The lack of gauge invariance cannot be remedied by any choice of local counter-terms, however it can be absorbed by coupling $\mathfrak{T}^\vee$ to an invertible topological field theory [74,75] known as the anomaly theory with the action

$$S_{\text{anom}} = \int_{M_{d+2}} A_d^{(N^\vee)} \cup \epsilon\left(A_1^{(K)}\right)\,. \tag{106}$$

Such an invertible field theory describes the ground state physics of a symmetry protected topological phase of matter protected by $G_{(0,d-1)}^\epsilon = \left[K_{(0)},\ N_{(d-1)}^\vee\right]^\epsilon$. It is however crucial to emphasize that there is no physical need to associate $\mathfrak{T}^\vee$ to a 'bulk'. As described in Sec. 2, such a theory can very well be described on an $d$-dimensional lattice model. Instead the bulk or anomaly theory is a theoretical gadget to systematize our understanding of the anomaly, which

has significant non-perturbative implications for the infra-red phases/ground states realized in $\mathfrak{T}^\vee$. Specifically, as we will demonstrate, a systems with such anomalies cannot have a gapped and symmetric (disordered) ground state. Instead, any gapped ground state must break the symmetry down to a subgroup that trivializes the anomaly. This is a consequence of anomaly matching between the ultraviolet and infra-red physics.

For the remainder of this section, we specialize to the case $\mathsf{G}_{(0)} = \mathbb{Z}_4$ and $\mathsf{N}_{(0)} = \mathbb{Z}_2$, in which case the symmetry of $\mathfrak{T}^\vee$ is $\mathsf{G}_{(0,d-1)}^\epsilon = \left[\mathbb{Z}_{2,(0)},\, \mathbb{Z}_{2,(d-1)}\right]^\epsilon$ with the anomaly action having the explicit form

$$S_{\text{anom}} = i\pi \int_{M_{d+2}} A_d \cup \text{Bock}(A_1)\,, \tag{107}$$

where $A_p \in H^p(M, \mathbb{Z}_2)$ and Bock denotes the Bockstein homomorphism (see Appendix B of [3] for details) which is a map of cohomology classes

$$\text{Bock} : H^1(M, \mathbb{Z}_2) \longrightarrow H^2(M, \mathbb{Z}_2)\,, \qquad \text{Bock}(A_1) = \frac{1}{2}d\widetilde{A}_1\,, \tag{108}$$

where $\widetilde{A}_1$ is the lift of $A_1$ to a $\mathbb{Z}_4$ gauge field.

### 3.2.2 Mapping of symmetry sectors

Under gauging of a subgroup, the symmetry sectors of the pre-gauged and gauged theories $\mathfrak{T}$ and $\mathfrak{T}^\vee$ respectively, map into each other in a non-trivial way. In this section, we detail the map of sectors for the case of $\mathsf{N} = \mathbb{Z}_2$ and $\mathsf{G} = \mathbb{Z}_4$. However the analysis readily generalizes to any finite Abelian group $\mathsf{G}$ with subgroup $\mathsf{N}$. Since we want to gauge $\mathbb{Z}_2 \subset \mathbb{Z}_4$, it will be convenient to write a group element $\mathsf{g} \in \mathbb{Z}_4 \cong \{0, 1, 2, 3\}$ as a tuple $(\mathsf{n}, \mathsf{k})$ such that $\mathsf{n}, \mathsf{k} \in \mathbb{Z}_2 \cong \{0, 1\}$, with the identification $\mathsf{g} = 2\mathsf{n} + \mathsf{k}$ and the product rule in $\mathbb{Z}_4$ given by

$$(\mathsf{n}_1, \mathsf{k}_1) \cdot (\mathsf{n}_2, \mathsf{k}_2) = (\mathsf{n}_1 + \mathsf{n}_2 + \mathsf{k}_1 \cdot \mathsf{k}_2,\, \mathsf{k}_1 + \mathsf{k}_2)\,. \tag{109}$$

A $\mathbb{Z}_4$ gauge field $A_1^{(\mathsf{G})} \in H^1(M, \mathbb{Z}_4)$ for $M = M_d \times S^1$ is labelled by its holonomies on the homology 1-cycles of $M$ and can correspondingly be expressed as a tuple of $\mathbb{Z}_2$ gauge fields $\left(A_1^{(\mathsf{N})} A_1^{(\mathsf{K})}\right)$ with a modified cocycle condition (101) as

$$\begin{aligned} A_1^{(\mathsf{G})} &= (\mathsf{g}_t, \mathsf{g}_1, \dots, \mathsf{g}_N) = (\mathsf{g}_t, \vec{\mathsf{g}})\,, \\ A_1^{(\mathsf{N})} &= (\mathsf{n}_t, \mathsf{n}_1, \dots, \mathsf{n}_N) = (\mathsf{n}_t, \vec{\mathsf{n}})\,, \\ A_1^{(\mathsf{K})} &= (\mathsf{k}_t, \mathsf{k}_1, \dots, \mathsf{k}_N) = (\mathsf{k}_t, \vec{\mathsf{k}})\,, \end{aligned} \tag{110}$$

where $N = b_1(M_d)$, $\vec{\mathsf{g}} \in H^1(M_d, \mathbb{Z}_4)$ and $\vec{\mathsf{n}}, \vec{\mathsf{k}} \in H^1(M_d, \mathbb{Z}_2)$. The thermal partition function of $\mathfrak{T}$ coupled to the background $\mathbb{Z}_4$ gauge field $A_1^{(\mathsf{G})}$ has the form

$$\mathcal{Z}_{\mathfrak{T}}\left[A_1^{(\mathsf{G})}\right] \equiv \mathcal{Z}_{\mathfrak{T}}\left[\mathsf{g}_t, \vec{\mathsf{g}}\right] \equiv \mathcal{Z}_{\mathfrak{T}}\left[\mathsf{n}_t, \mathsf{k}_t, \vec{\mathsf{n}}, \vec{\mathsf{k}}\right] = \text{Tr}\left[\mathcal{U}^{2\mathsf{n}_t + \mathsf{k}_t}\, e^{-\beta H_{\vec{\mathsf{n}}, \vec{\mathsf{k}}}}\right]\,, \tag{111}$$

where $\mathcal{U}$ is the $\mathbb{Z}_4$ generator and $H_{\vec{\mathsf{n}}, \vec{\mathsf{k}}}$ is the $\mathbb{Z}_4$ symmetric Hamiltonian with $\vec{\mathsf{g}} = (\vec{\mathsf{n}}, \vec{\mathsf{k}})$ twisted boundary conditions. With the purpose, of tracking how symmetry sectors map under partial-gauging, we define $\mathcal{P}_{\alpha_{\mathsf{g}}}$ which is a projector onto the sub Hilbert space transforming in the $\alpha_{\mathsf{g}}$ representation of $\text{Rep}(\mathbb{Z}_4)$

$$\mathcal{P}_{\alpha_{\mathsf{g}}} = \frac{1}{4} \sum_{\mathsf{g}_t \in \mathbb{Z}_4} i^{-\alpha_{\mathsf{g}}\mathsf{g}_t} \times \mathcal{U}^{\mathsf{g}_t} = \frac{1}{4} \sum_{\mathsf{n}_t, \mathsf{k}_t \in \mathbb{Z}_2} (-1)^{\alpha_{\mathsf{n}}\mathsf{n}_t + \alpha_{\mathsf{k}}\mathsf{k}_t} i^{-\alpha_{\mathsf{n}}\mathsf{k}_t} \times \mathcal{U}^{2\mathsf{n}_t + \mathsf{k}_t}\,, \tag{112}$$

where we have used $\alpha_g = 2\alpha_k + \alpha_n$. Using (112) we may define a symmetry character $\chi[q, \alpha_g, g]$ as the thermal trace in a definite eigensector $\alpha_g \in \mathrm{Rep}(\mathbb{Z}_4)$ and with symmetry twisted boundary conditions $\vec{g} = 2\vec{n} + \vec{k}$ as

$$\chi_{\mathfrak{T}}[\alpha_g, g] = \mathrm{Tr}\left[\mathcal{P}_{\alpha_g} q^{-H_{\vec{n},\vec{k}}}\right] = \frac{1}{4}\sum_{n_t, k_t}(-1)^{\alpha_n n_t + \alpha_k k_t} i^{-\alpha_n k_t} \mathcal{Z}_{\mathfrak{T}}\left[n_t, k_t, \vec{n}, \vec{k}\right]. \tag{113}$$

The expression (113) can be readily inverted to express the partition function in terms of the symmetry characters as

$$\mathcal{Z}_{\mathfrak{T}}\left[n_t, k_t, \vec{n}, \vec{k}\right] = \sum_{\alpha_n, \alpha_k}(-1)^{\alpha_n n_t + \alpha_k k_t} i^{\alpha_n k_t} \chi_{\mathfrak{T}}\left[2\alpha_k + \alpha_n, 2\vec{n} + \vec{k}\right]. \tag{114}$$

Next, gauging the subgroup $\mathsf{N} \subset \mathsf{G}$ simply corresponds to summing over the symmetry background $A_1^{(\mathsf{N})}$. The partition function of the partially gauged theory $\mathfrak{T}^\vee$ has a $d$-group $\mathsf{G}^\epsilon_{(0,d-1)} = \left[\mathsf{K}_{(0)}, \mathsf{N}^\vee_{(d-1)}\right]^\epsilon$ global symmetry and can therefore be coupled to a symmetry background $\left(A_1^{(\mathsf{K})}, A_d^{(\mathsf{N}^\vee)}\right)$. In particular, the symmetry background to $A_d^{(\mathsf{N}^\vee)}$ are labelled by $\vec{n}^\vee$ and $n_t^\vee$, see (95). The partition function of $\mathfrak{T}^\vee$ coupled to $(A_{1,\mathsf{K}}, A_{d,\mathsf{N}^\vee})$ is

$$\mathcal{Z}_{\mathfrak{T}^\vee}\left[A_d^{(\mathsf{N}^\vee)}, A_1^{(\mathsf{K})}\right] \equiv \mathcal{Z}_{\mathfrak{T}^\vee}\left[\vec{n}^\vee, n_t^\vee, k_t, \vec{k}\right] = \frac{1}{2}\sum_{n_t, \vec{n}} \mathcal{Z}_{\mathfrak{T}}\left[n_t, k_t, \vec{n}, \vec{k}\right](-1)^{n_t n_t^\vee + \vec{n}\cdot\vec{n}^\vee}. \tag{115}$$

As a thermal partition function, this may be expressed as

$$\mathcal{Z}_{\mathfrak{T}^\vee}\left[\vec{n}^\vee, n_t^\vee, k_t, \vec{k}\right] = \mathrm{Tr}\left[\mathcal{U}^{k_t}\prod_{j=1}^{N} W_{\gamma_j}^{n_j^\vee} e^{-\beta H_{\vec{k}, n_t^\vee}^\vee}\right], \tag{116}$$

where $\mathcal{U}$ and $W_{\gamma_j}$ are the $\mathbb{Z}_2$ 0-form and $(d-1)$-form symmetry generators in the theory $\mathfrak{T}^\vee$ respectively. Since we are interested in relating the symmetry-resolved energy spectra of the two theories $\mathfrak{T}$ and $\mathfrak{T}^\vee$, we need to define the character of the dual theory $\mathfrak{T}^\vee$ as well. However, due to the mixed anomaly in this theory, see equation (82) and figure 7, some slight care is needed to define characters correctly. In particular, equation (82) implies that in twisted sectors the $\mathbb{Z}_2$ symmetry operators can square to $-1$ instead of $+1$ and thus have 'fractionalized' symmetry eigenvalues $\pm i$ instead of $\pm 1$. The appropriate symmetry projector for the $\mathsf{G}^\epsilon_{(0,d-1)} = \left[\mathsf{K}_{(0)}, \mathsf{N}^\vee_{(d-1)}\right]^\epsilon$ is thus[12]

$$\mathcal{P}[\vec{\alpha}_{n^\vee}, \alpha_k, n_t^\vee, \vec{k}] = \prod_{j=1}^{N}\mathcal{P}^{(W)}_{\alpha_{n_j^\vee}, k_j} P^{(\mathcal{U})}_{\alpha_k, n_t^\vee} \mathcal{P}^{(\mathcal{T}^{(d-1)})}_{n_t^\vee} \mathcal{P}^{(\mathcal{T}_{\gamma_j}^{(0)})}_{k_j}, \tag{117}$$

where the superscript of each projector denotes the operator whose eigenspace is being projected onto while the subscripts denote the eigenvalues. Explicitly, these projection operators have the form

$$
\begin{aligned}
\mathcal{P}^{(W)}_{\alpha_{n_j^\vee}, k_j} &= \frac{1 + (-1)^{\alpha_{n_j^\vee}} i^{-k_j} W_{\gamma_j}}{2}, & \mathcal{P}^{(\mathcal{T}_{\gamma_j}^{(0)})}_{k_j} &= \frac{1 + (-1)^{k_j}\mathcal{T}_{\gamma_{j'}}^{(0)}}{2}, \\
P^{(\mathcal{U})}_{\alpha_k, n_t^\vee} &= \frac{1 + (-1)^{\alpha_k} i^{-n_t^\vee}\mathcal{U}}{2}, & \mathcal{P}^{(\mathcal{T}^{(d-1)})}_{n_t^\vee} &= \frac{1 + (-1)^{n_t^\vee}\mathcal{T}^{(d-1)}}{2},
\end{aligned}
\tag{118}
$$

---

[12]Note that technically only $\mathcal{P}^{(W)}$ and $\mathcal{P}^{(\mathcal{U})}$ are projectors to symmetry eigenspaces (for the $(d-1)$ and 0-form symmetries, respectively). The 'twisted BC projectors' $\mathcal{P}^{(\mathcal{T}^{(d-1)})}$ and $\mathcal{P}^{(\mathcal{T}^{(0)})}$ are not real projectors, and somewhat trivial. But we include these here to easier keep track of how symmetry sectors and boundary conditions swap under gauging dualities.

where $\mathcal{U}$, $W_{\gamma_j}$, $\mathcal{T}^{(0)}_{\gamma_j}$ and $\mathcal{T}^{(d-1)}$ are defined in (80), (81) and (82). Here the symmetry characters of the gauged theory are labelled by a representation $\alpha_k$ and $\vec{\alpha}_{n^\vee} = (\alpha_{n_1^\vee}, \ldots, \alpha_{n_N^\vee})$, while twisted boundary conditions are labeled by $n_t^\vee$ and $\vec{k}$. Inserting this into the partition function, the characters of $\mathfrak{T}^\vee$ take the form (see also [123] for a similar discussion in $1+1$ dimensions)

$$\chi_{\mathfrak{T}^\vee}\left[\vec{\alpha}_{n^\vee}, \alpha_k, n_t^\vee, \vec{k}\right] = \frac{1}{2^{1+N}} \sum_{k_t, \vec{n}^\vee} \mathcal{Z}_{\mathfrak{T}^\vee}\left[\vec{n}^\vee, n_t^\vee, k_t, \vec{k}\right] (-1)^{\alpha_k k_t + \vec{\alpha}_{n^\vee} \cdot \vec{n}^\vee} i^{-n_t^\vee k_t - \vec{k} \cdot \vec{n}^\vee}. \tag{119}$$

The appearance of i in the characters above, is a consequence of symmetry fractionalization stemming from the mixed anomaly. The symmetry characters of $\mathfrak{T}^\vee$ can be written in terms of the characters of $\mathfrak{T}$ by using (115) and (114). However, let us instead derive this relation using the lattice realization of the symmetry structures of $\mathfrak{T}$ and $\mathfrak{T}^\vee$ described in Sec. 2. To extract the mapping of sectors, we note that a sector in the theory $\mathfrak{T}^\vee$ labelled as $(\vec{\alpha}_{n^\vee}, \alpha_k, n_t^\vee, \vec{k})$ is the sub-Hilbert space of $\mathcal{V}^\vee_{M_d}$ in the simultaneous image of the projection operators (117). The gauging map in Sec. 2.2 relates $\mathbb{Z}_{2,(0)} \times \mathbb{Z}_{2,(d-1)}$ symmetric operators on $\mathcal{V}^\vee_{M_d}$ to $\mathbb{Z}_4$ symmetric operators on $\mathcal{V}_{M_d}$. In particular there is the following mapping of operators

$$
\begin{aligned}
\mathcal{U}\Big|_{\mathcal{V}^\vee} &\longmapsto \mathcal{U}\Big|_{\mathcal{V}} = \prod_v X_v, & \mathcal{T}^{(d-1)}\Big|_{\mathcal{V}^\vee} &\longmapsto \mathcal{U}^2\Big|_{\mathcal{V}}, \\
W_\gamma\Big|_{\mathcal{V}^\vee} &\longmapsto \mathcal{T}^{(0)}\Big|_{\mathcal{V}} = \prod_{e \subset \gamma} Z_{s(e)} Z_{t(e)}^\dagger, & \mathcal{T}^{(0)}_{\gamma_j}\Big|_{\mathcal{V}^\vee} &\longmapsto \left[\mathcal{T}^{(0)}\right]^2\Big|_{\mathcal{V}}.
\end{aligned}
\tag{120}
$$

Using these operator isomorphisms, the product of projectors in (117) in the theory $\mathfrak{T}$ maps to a product of projectors in the theory $\mathfrak{T}^\vee$. More precisely, one obtains a projector onto the $2\alpha_k - n_t^\vee \in \mathrm{Rep}(\mathbb{Z}_4)$ eigensector of $\mathcal{U}$ and the sector with $2\alpha_{n_j^\vee} - k_j \in \mathbb{Z}_4$ symmetry twisted boundary conditions along the cycle $\gamma_j$. Which implies the following map of symmetry sectors

$$\chi_{\mathfrak{T}^\vee}[q, \vec{\alpha}_{n^\vee}, \alpha_k, n_t^\vee, \vec{k}] = \chi_{\mathfrak{T}}[q, 2\alpha_k + n_t^\vee, 2\vec{\alpha}_{n^\vee} + \vec{k}]. \tag{121}$$

# 4 Phase diagrams and dualities in $d = 2$

Having detailed the gauging of finite Abelian symmetries and their subgroups both in the lattice setting in Sec. 2 and more generally in a space-time approach in Sec. 3, we now turn our attention to the action of gauging-related dualities on phase diagrams in two-dimensional space. More precisely, let us again consider a theory $\mathfrak{T}$ with global symmetry $G$ by which we mean a parameter space of $G$-symmetric Hamiltonians. This space of local Hamiltonians is contained within the bond algebra $\mathfrak{B}_G(\mathcal{V})$. Upon gauging either the full group $G$ or a certain subgroup, we obtain a new theory $\mathfrak{T}^\vee$, i.e., a parameter space of models in the bond algebra $\mathfrak{B}_{G^\vee}(\mathcal{V}^\vee)$, where $G^\vee$ is typically a higher group, potentially with mixed anomalies. Since the gauging map is an isomorphism between the bond algebras, the physics before and after gauging is intimately related, or more precisely dual. This duality is evident in several aspects. For instance, the spectrum of a $G$ symmetric Hamiltonian $\mathcal{H}$ and its $G^\vee$-symmetric image $\mathcal{H}^\vee$ under (partial) gauging have the same spectrum in 'dual' symmetry sectors, in the sense detailed in Sec. 3. Another consequence is the equality of correlation functions

$$\langle \mathcal{O}_1(x_1, t_1) \cdots \mathcal{O}_n(x_n, t_n) \rangle_\Phi = \langle \mathcal{O}_1^\vee(x_1, t_1) \cdots \mathcal{O}_n^\vee(x_n, t_n) \rangle_{\Phi^\vee}, \tag{122}$$

where $\Phi$ collectively denotes the symmetry sector labels of theory $\mathfrak{T}$ and $\mathcal{O}_j$ are operators in the bond algebra $\mathfrak{B}_G$. $\Phi^\vee$ and $\mathcal{O}_j^\vee$ are the images of $\Phi$ and $\mathcal{O}_j$ under the gauging map.

## 4.1 Gauging finite Abelian symmetry

In this section, we describe how the phase diagrams of a theory with 0-form and 1-form $\mathbb{Z}_n$ symmetry are related. Our analysis generalizes to any finite Abelian group $G$ with a few caveats which we will elucidate as we go along. To organize the mapping of phase diagrams under (partial) gauging, we seek to enumerate $G$-symmetric gapped phases in two dimensions. Doing so, one immediately encounters a complication in the fact that there are infinitely many $G$-symmetric gapped phases of matter if one includes phases with long range entanglement. This should not come as a surprise as the set of gapped phases (which admit a lattice Hamiltonian description) without any symmetry enrichment is already infinitely large and contains all topological orders with gapped boundaries. This set is at least as rich as fusion categories since a topological order (with a gappable boundary) can be constructed from a given fusion category as input [112, 124, 125]. Here, we content ourselves by investigating the parameter space of $G_{(0)}$ symmetric short range entangled (SRE) systems.

SRE gapped phases are classified by tuples $[H, \nu]$ where $H \subseteq G$ and $\nu \in H^3(H, U(1))$. A phase thus labelled spontaneously breaks the global symmetry down to $H$. Therefore, such a phase possesses $|G/H|$ ground states that form an orbit under the action of $G$. Furthermore, each such ground state is a symmetry protected topological (SPT) phase labelled by $\nu \in H^3(H, U(1))$ of the unbroken $H$ subgroup [126]. We consider the following $G$-symmetric Hamiltonians which can access all SRE gapped phases

$$\mathcal{H}[\lambda] = \sum_{H, \nu} \lambda_{[H, \nu]} \mathcal{H}_{[H, \nu]}, \qquad (123)$$

where the parameters $\lambda_{[H, \nu]} \in \mathbb{R}$ and $\mathcal{H}_{[H, \nu]}$ is a fixed-point Hamiltonian for each gapped phase. Setting all $\lambda_{[H, \nu]} = 0$ except for one pair $[H', \nu']$ selects a fixed-point Hamiltonian $\mathcal{H}_{[H', \nu']}$ realizing the gapped phase labelled by $[H', \nu']$.

Upon gauging $G_{(0)}$, one realizes a model with a dual $G_{(1)}^\vee$ global symmetry. Correspondingly, each $G_{(0)}$ symmetric gapped phase $[H, \nu]$ maps to a $G_{(1)}^\vee$ symmetric gapped phase we label $[H, \nu]^\vee$. A natural question then is where does $[H, \nu]^\vee$ fit into the classification of $G_{(1)}^\vee$ symmetric gapped phases? We will see that the data $[H, \nu]$ correspond to information pertaining to (i) the 1-form symmetry that is preserved in the dual model and (ii) the topological properties of certain emergent 1-form symmetries that arise in phases with spontaneously broken 1-form symmetry. To dissect these statements let us first describe what is meant by 1-form symmetry breaking?

**1-form symmetry breaking:** Similar to conventional 0-form symmetry breaking, 1-form symmetry breaking is also signalled by long-range order of an operator which is charged under the relevant symmetry. In the case of 0-form symmetry, the charged local operator is the local order parameter. In contrast, 1-form symmetry breaking is diagnosed by a perimeter law expectation value of a closed line operator which is charged under the 1-form symmetry in question [56, 67, 127, 128]. In the parlance of gauge theory, the charged line operator is deconfined in the 1-form symmetry broken phase. In the infra-red/low-energy limit, such a charged line operator becomes topological, i.e., its expectation value is invariant under topological deformations. Furthermore, tautologically, there is a non-trivial linking between the charged line and 1-form symmetry generator (see fig 11). Therefore the low energy theory contains topological line operators that braid non-trivially and is topologically ordered [10]. These phases are paradigmatic examples of models with long-range entanglement and therefore one observes that gauging maps short range entangled phases to long range entangled phases in $2 + 1$ dimensions [69, 71, 129].



Figure 11: 1-form symmetry breaking is signalled by the deconfinement of a line operator which is charged under the 1-form symmetry generator. In the infra-red/low energy limit, such an operator becomes topological.

Let us consider the case where $G = \mathbb{Z}_n$. The gapped phases are labelled by the pair $[\mathbb{Z}_p, \ell]$ with $p$ a divisor of $n$ and $\ell \in H^3(\mathbb{Z}_p, U(1)) \cong \mathbb{Z}_p$. Before analyzing the dualities on the lattice, let us first get some intuition about how the gapped phases are mapped under gauging. To do so we turn to topological partition functions, which encode the low energy physics within a given gapped phase. For $G = \mathbb{Z}_n$, the topological partition function for the gapped phase labelled by $[\mathbb{Z}_p, \ell]$ has the form

$$\mathcal{Z}_{\mathfrak{I}}[M, A_1] = \frac{n}{p} \delta_{pA_1, 0} \exp\left\{ \frac{2\pi i \ell}{p} \int_M A_1 \cup \text{Bock}(A_1) \right\}, \tag{124}$$

where the Bockstein homomorphism $\text{Bock} : H^1(M, \mathbb{Z}_p) \to H^2(M, \mathbb{Z}_p)$ is implemented by

$$\text{Bock}(A_1) = \frac{1}{p} d\widetilde{A}_1, \tag{125}$$

and $\widetilde{A}_1$ is a lift of $A_1$ to $\mathbb{Z}_{p^2}$. The factor of $\delta_{pA_1, 0}$ implies that the theory cannot be coupled to a $\mathbb{Z}_{n/p} \subset \mathbb{Z}_p$ as it is broken in this phase. Meanwhile the topological action in the exponent of (124) is the SPT response theory/effective action of the $\mathbb{Z}_p$ global symmetry which remains unbroken. Then the low-energy or ground state physics of the dual gapped phase obtained after gauging, can be extracted using (85). The topological partition function corresponding to this dual gapped phase has the form

$$\begin{aligned}
\mathcal{Z}_{\mathfrak{I}^\vee}[M] &= \frac{1}{p} \sum_{a_1 \in H^1(M, \mathbb{Z}_p)} \exp\left\{ \frac{2\pi i \ell}{p} \int_M a_1 \cup \text{Bock}(a_1) \right\} \\
&= \frac{1}{p} \sum_{a_1, b_1 \in C^1(M, \mathbb{Z}_p)} \exp\left\{ \frac{2\pi i}{p} \int_M \left[ b_1 \cup da_1 + \ell a_1 \cup \frac{da_1}{p} \right] \right\}.
\end{aligned} \tag{126}$$

This theory is a $\mathbb{Z}_n$ Dijkgraaf-Witten theory [130] with the topological action labelled with $\ell \in H^3(\mathbb{Z}_p, U(1))$ [129, 131]. In the second line, we have simply re-formulated the Dijkgraaf-Witten theory in terms of 1-cochains $a_1, b_1 \in C^1(M, \mathbb{Z}_p)$, which is the quantum double formulation (see [129] and references therein). The additional field $b_1$ serves to parametrize/probe the aforementioned emergent 1-form symmetry and can be summed over to go back to the first line. By inspecting the equations of motion, the most general topological operator can be read off to be

$$W_{(q,m)}(L) = \exp\left\{ \frac{2\pi i}{p} \oint_L (q a_1 + m b_1) \right\}, \tag{127}$$

where $q, m \in \mathbb{Z}_p$. We note that $W_{(0,1)}(L)$ generates an emergent 1-form symmetry. Notice that due to the delta function $\delta_{pA_1,0}$ in (124), the following holds true

$$\left\langle \exp\left\{ ip \oint_L a_1 \right\} \right\rangle = 1 \, , \tag{128}$$

for any 1-cycle $L$. Since the operator $\exp\left\{ ip \oint_L a_1 \right\}$ generates a $\mathbb{Z}_{n/p,(1)} \subset \mathbb{Z}_{n,(1)}$ symmetry, (128) implies that all lines charged under $\mathbb{Z}_{n/p}$ are confined and do not appear in the infra-red fixed point. This is equivalent to the fact that $\mathbb{Z}_{n/p,(1)} \subset \mathbb{Z}_{n,(1)}$ remains unbroken.

Next let us inspect the fate of the remaining $\mathbb{Z}_{p,(1)}$ generated by $W_{(1,0)}$. To do so, we turn on sources for the $\mathbb{Z}_p$ 1-form symmetry as well as the emergent 1-form symmetry. This can be done by introducing two 2-form background gauge fields $A_2, B_2 \in Z^2(M, \mathbb{Z}_p)$ that enter the partition function as

$$\mathcal{Z}_{\mathfrak{T}^\vee}[M, A_2, B_2] = \frac{1}{p} \sum_{a_1, b_1} \exp\left\{ \frac{2\pi i}{p} \int_M \left[ b_1 \cup da_1 + \ell a_1 \cup \frac{da_1}{p} + A_2 \cup a_1 + B_2 \cup b_1 \right] \right\} . \tag{129}$$

The correlation functions of the topological line operators can be computed straightforwardly using standard methods [131, 132]. For instance consider two 1-cycles $\gamma_1$ and $\gamma_2$ embedded in $M = S^3$ such that they form a Hopf-link. The correlation function of two line operators with support on $L_1$ and $L_2$ is

$$\left\langle W_{(q_1,m_1)}(L_1) W_{(q_2,m_2)}(L_2) \right\rangle = \exp\left\{ \frac{2\pi i}{p}(q_1 m_2 + q_2 m_1) + \frac{4\pi i \ell}{p^2} m_1 m_2 \right\} . \tag{130}$$

Hence, the parameter $\ell \in H^3(\mathbb{Z}_p, U(1))$ changes the self-braiding and topological spin of the emergent 1-form symmetry generators.

### 4.1.1 SPT Hamiltonians on the lattice

We now turn to how these features manifest on the lattice. As before we consider a triangulated manifold $M_{2,\triangle}$ with each vertex endowed with a local Hilbert space $\mathcal{V}_v \cong \mathbb{C}^n$. We investigate how fixed-point Hamiltonians dualize under gauging. A fixed point Hamiltonian in the phase labelled as $[\mathbb{Z}_p, \ell]$ has the form

$$\mathcal{H}_{[\mathbb{Z}_p,\ell]} = -\sum_v X_v^{n/p} \exp\left\{ \frac{2\pi i \ell}{p^2} \sum_{e \subset \partial \text{Hex}_v} \mathcal{B}_e \right\} - \sum_e Z_{s(e)}^p Z_{t(e)}^{-p} + \text{H.c.} \, , \tag{131}$$

where $\text{Hex}_v$ denotes the smallest hexagon in the direct lattice enclosing vertex $v$ and $\partial \text{Hex}_v$ is the set of six edges at the boundary of this hexagon (see Fig. 12). $\mathcal{B}_e$ is defined as

$$\mathcal{B}_e = \sum_{\alpha=0}^{p-1} \alpha \mathcal{P}_e^{(\alpha)}, \qquad \mathcal{P}_e^{(\alpha)} = \frac{1}{p} \sum_{\tau=0}^{n-1} e^{-\frac{2\pi i \tau \alpha}{p}} (Z_{s(e)} Z_{t(e)}^\dagger)^\tau \, . \tag{132}$$

We are interested in the groundstate symmetry properties of (131). Since $X_v^{n/p}$ and $Z_v^p$ commute, (131) is a commuting projector Hamiltonian and the ground states lie in $Z_{s(e)}^p Z_{t(e)}^{-p} = 1$ subspace of the Hilbert space. It follows that in this subspace $Z_{s(e)} Z_{t(e)}^\dagger$ have eigenvalues that are the $p^{\text{th}}$ roots of unity and consequently $\mathcal{P}_e^{(\alpha)}$ is a projector onto the $\exp\{2\pi i \alpha/p\}$ eigenspace of this operator. The Hilbert space splits in $n/p$ super-selection sectors, that are dynamically disconnected in the thermodynamic (infinite size) limit as

$$\mathcal{V} = \bigoplus_{g=0}^{n/p-1} \mathcal{V}_g, \qquad \mathcal{V}_g = \text{Span}_\mathbb{C}\left\{ |\phi\rangle_g \,\middle|\, \phi \in C^0(M_\triangle, \mathbb{Z}_p) \right\} , \tag{133}$$

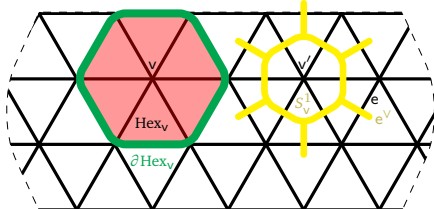 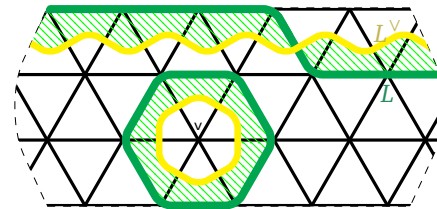

Figure 12: (a) $\text{Hex}_\text{v}$ denotes the smallest hexagon (in red) on the direct lattice enclosing vertex v and $\partial \text{Hex}_\text{v}$ is the set of six edges (in green) on the boundary of this hexagon. The triangular lattice is dual to the hexagonal lattice such that each edge e of the triangular lattice is associated to an edge $\text{e}^\vee$ of the dual lattice. We denote a unit cell of the dual lattice that links with the vertex v of the direct lattice as $S_\text{v}^1$. (b) The Hamiltonian (143) is sum of framed ribbon operators linking each vertex v of the triangular lattice. The figure (b) also depicts a twisted ribbon operator.

such that

$$Z_\text{v}|\phi\rangle_\text{g} = \exp\left\{\frac{2\pi\text{i}\phi_\text{v}}{p} + \frac{2\pi\text{i}g}{n}\right\}|\phi\rangle_\text{g}. \tag{134}$$

Note that the sub-Hilbert spaces $\mathcal{V}_g$ are distinct eigenspaces of $Z_\text{v}^p$, which is the order parameter of spontaneously breaking of $\mathbb{Z}_n$ symmetry to $\mathbb{Z}_p \subset \mathbb{Z}_n$. Since the different super-selection sectors are dynamically disconnected, we may look at the Hamiltonian (131) in a specific subspace $\mathcal{V}_g$, which we denote by

$$\mathcal{H}_{[\mathbb{Z}_p,\ell]}^{(g)} = \mathcal{H}_{[\mathbb{Z}_p,\ell]}\Big|_{\mathcal{V}_g}. \tag{135}$$

Since, $Z_\text{v} X_\text{v}^{n/p} = \exp\{2\pi\text{i}/p\} X_\text{v}^{n/p} Z_{\text{v}'}$, the operators $\left\{X_\text{v}^{n/p}, Z_\text{v}\right\}$ restricted to $\mathcal{V}_g$ generate a $\mathbb{Z}_p$ clock and shift algebra, i.e.,

$$Z_{\text{s(e)}} Z_{\text{t(e)}}^\dagger |\phi\rangle_\text{g} = \exp\left\{\frac{2\pi\text{i}}{p}(\text{d}\phi)_\text{e}\right\}|\phi\rangle_\text{g},$$
$$X_\text{v}|\phi\rangle_\text{g} = |\phi + \delta_\text{v}\rangle_\text{g}. \tag{136}$$

We define effective $\mathbb{Z}_p$ degrees of freedom in $\mathcal{V}_g$ as

$$X_\text{v}^{n/p}\Big|_{\mathcal{V}_g} = e^{\text{i}\Theta_\text{v}}, \qquad Z_\text{v}\Big|_{\mathcal{V}_g} = e^{\text{i}\Phi_\text{v}}, \tag{137}$$

which satisfy the commutation relations $[\Theta_\text{v}, \Phi_{\text{v}'}] = 2\pi\text{i}\delta_{\text{v},\text{v}'}/p$. In terms of these effective $\mathbb{Z}_p$ degrees of freedom, we have

$$\mathcal{H}_{[\mathbb{Z}_p,\ell]}^{(g)} = -\sum_\text{v} e^{\text{i}\Theta_\text{v}} \exp\left\{\frac{2\pi\text{i}\ell}{p^2} \oint_{\partial \text{Hex}_\text{v}} \text{d}\Phi\right\}. \tag{138}$$

This is nothing but a fixed point Hamiltonian for the $\mathbb{Z}_p$-SPT labelled by $\ell \in H^3(\mathbb{Z}_p, \text{U}(1))$ [69]. Therefore, we have demonstrated that the ground states of the Hamiltonian (131) break the global symmetry down to $\mathbb{Z}_p \subset \mathbb{Z}_n$ and furthermore, each symmetry broken ground state realizes an SPT of the unbroken symmetry.

### 4.1.2 Dual Hamiltonian with higher-form symmetry

Now we turn to gauging the global $\mathbb{Z}_n$ 0-form symmetry. The model after gauging is defined on $M_{2,\triangle}$ with the degrees of freedom defined on the edges instead of the vertices. The dual

Table 1: Summary of dualities between $\mathbb{Z}_{4,(0)}$-symmetric SRE gapped phases and $\mathbb{Z}_{4,(1)}$-symmetric models with gapped phases in $d = 2$ spatial dimensions. The SRE ground states of $\mathbb{Z}_{4,(0)}$-symmetric Hamiltonians are described by the pair $[\mathbb{Z}_{p,(0)}, \ell_p]$. Here, the subgroup $\mathbb{Z}_{p,(0)} \subset \mathbb{Z}_{4,(0)}$ with $p = 1, 2, 4$ denotes the symmetry preserved by the $4/p$-fold degenerate ground states. We refer to the case $p = 1$ as spontaneously symmetry broken (SSB), while the case of $p = 2$ is referred to as partial symmetry broken (PSB). The index $\ell_p \in \mathbb{Z}_p$ labels the SPT phase supported by each of the $4/p$-fold degenerate ground states, which we refer to as SPT($\ell_p$). The dual models are obtained by gauging the $\mathbb{Z}_{4,(0)}$ global symmetry. On the dual side, a phase preserving $\mathbb{Z}_{p,(0)}$ symmetry maps to a phase preserving $\mathbb{Z}_{4/p,(1)} \subset \mathbb{Z}_{4,(1)}$ subgroup. In this case, the ground state has an emergent $\mathbb{Z}_{p,(1)}$ symmetry whose generator carries topological spin $\theta = \pi \ell_p$. We denote such dual ground states by $[\mathbb{Z}_{p,(1)}, \ell_p]$. By "Triv." we mean the trivial symmetry group.

| $\mathbb{Z}_{4,(0)}$ SRE Phases | | Dual $\mathbb{Z}_{4,(1)}$ Gapped Phases | |
|---|---|---|---|
| Symmetry of GS | Description | Symmetry of GS | Description |
| $\left[\mathbb{Z}_{4,(0)}, \ell_4\right]$ | SPT($\ell_4$) | $\left[\text{Triv.}, \ell_4\right]$ | Emergent $\mathbb{Z}_{4,(1)}$, $\ell_4$ w/ $\theta = \pi\ell_4$ |
| $\left[\mathbb{Z}_{2,(0)}, \ell_2\right]$ | SPT($\ell_2$) × PSB | $\left[\mathbb{Z}_{2,(1)}, \ell_2\right]$ | Emergent $\mathbb{Z}_{2,(1)}$ w/ $\theta = \pi\ell_2$ |
| $[\text{Triv.}, 0]$ | SSB | $\left[\mathbb{Z}_{4,(1)}, 0\right]$ | Symmetry preserving |

Hamiltonians can be obtained by using the bond algebra isomorphsim between (20) and (40). The $\mathbb{Z}_n$ 0-form symmetric fixed point Hamiltonian (131) in the phase $[\mathbb{Z}_p, \ell]$ maps to the following dual Hamiltonian

$$\mathcal{H}^\vee_{[\mathbb{Z}_p, \ell]} = -\sum_v A_v^{n/p} \exp\left\{ \frac{2\pi i\ell}{p^2} \sum_{e \subset \partial\text{Hex}_v} \mathcal{B}_e^\vee \right\} - \sum_e Z_e^p + \text{H.c.}, \tag{139}$$

where

$$\mathcal{B}_e^\vee = \sum_{\alpha^\vee = 0}^{p-1} \alpha^\vee \mathcal{P}_e^{(\alpha^\vee)}, \qquad \mathcal{P}_e^{(\alpha^\vee)} = \frac{1}{p} \sum_{\tau=0}^{n-1} e^{-\frac{2\pi i\tau\alpha^\vee}{p}} Z_e^\tau. \tag{140}$$

Since the different terms in the Hamiltonian commute, it immediately follows that the ground-state is in the $Z_e^p = 1$ subspace of the Hilbert space. As a straightforward consequence, we have that

$$\prod_{e \in L} (Z_e^p)^{o(e,L)} = 1, \tag{141}$$

for any closed loop $L$. Since this line operator is the generator of $\mathbb{Z}_{n/p,(1)}$ 1-form symmetry, this implies that the ground state is invariant under this 1-form symmetry. Let us now investigate the fate of the remaining $\mathbb{Z}_p \subset \mathbb{Z}_n$. As before, we note that in a definite eigenspace of $Z_e^p$, the operators $X_e^{n/p}$ and $Z_e$ generate a clock and shift algebra. We use the representation

$$Z_e\Big|_{Z_e^p = 1} = e^{ia_e}, \qquad X_e^{n/p}\Big|_{Z_e^p = 1} = e^{ib_{e^\vee}}, \tag{142}$$

where we have used the convention that the $b_{e^\vee}$ operators are defined on the links of the dual lattice. They satisfy the algebra $[b_{e^\vee}, a_{e'}] = 2\pi i \, \text{Int}_{e^\vee, e'}/p$, where $\text{Int}_{e^\vee, e'}$ denotes the intersection number of edge $e^\vee$ and $e'$. Using this representation, we may express the Hamiltonian

(139) in the restricted $Z_e^p = 1$ subspace as

$$\mathcal{H}_{[\mathbb{Z}_p,\ell]}^{\vee}\bigg|_{Z_e=1} = -\sum_v \exp\left\{i\oint_{S_v^1} b + \frac{i\ell}{p}\oint_{\partial\text{Hex}_v} a\right\}. \tag{143}$$

This is nothing but the $\mathbb{Z}_p$ twisted quantum double Hamiltonian. The Hamiltonian is a sum over framed ribbon operators linking with the vertices of the direct lattice (see Fig. 12). More generally, the line

$$W(\Gamma) = \exp\left\{i\oint_{L^{\vee}} b + \frac{i\ell}{p}\oint_L a\right\}, \tag{144}$$

commutes with the Hamiltonian and is therefore topological, i.e., an emergent 1 form symmetry generator. Note that on the lattice one needs to provide two curves–$L$ and $L^{\vee}$ since $W(\Gamma)$ is a framed line operator. In terms of the Wilson operators defined in (127), this topological line operator is $W_{0,1}(\Gamma)$ with $\Gamma = (L, L^{\vee})$. Table 1 summarizes the dualities between Hamiltonians (131) and (143), when $n = 4$ and $p = 1, 2, 4$.

The discussion in this section generalizes to any finite Abelian group $\mathsf{G}$. Similar to the case of $\mathbb{Z}_n$, the gapped phases are labelled $[\mathsf{H}, \nu]$, labelling the symmetry $\mathsf{H}$ that is preserved by the ground state and the SPT class $\nu \in H^3(\mathsf{H}, \mathsf{U}(1))$ of each symmetry broken ground state. There are three possible types of cocycles for any finite Abelian group which are denoted as type-I, type-II and type-III cocycles [133]. The physics of type-I and type-II cocycles is a straightforward generalization of the case of $\mathbb{Z}_n$ presented in this section, however the physics of type-III is different. In particular after gauging an SPT labelled by a type-III cocycle $\nu$, one obtains a topological order with emergent non-invertible symmetries [129, 133, 134].

## 4.2 Gauging finite Abelian sub-symmetry

In this section, we study the dualities between two theories related by the partial gauging of $\mathbb{Z}_2 \subset \mathbb{Z}_4$ 0-form symmetry. Concretely, we gauge the $\mathbb{Z}_2$ subgroup of a $\mathbb{Z}_4$ symmetric system that can access all SRE gapped phases, i.e., combinations of SPT and symmetry broken phases. Such phases are labelled by $\mathsf{H} \subseteq \mathbb{Z}_4$ and a 3-cocycle (SPT action) $\nu \in H^3(\mathsf{H}, \mathsf{U}(1))$. There are a total of seven such gapped phases since there are three subgroups ($\mathbb{Z}_4, \mathbb{Z}_2$ and $\mathbb{Z}_1$) of $\mathbb{Z}_4$ and $H^3(\mathbb{Z}_n, \mathsf{U}(1)) = \mathbb{Z}_n$. We follow two complimentary strategies (i) partially gauge the topological (fixed-point) partition functions of each of the gapped phases in the $\mathbb{Z}_4$ model as in Sec. 3.2 and (ii) carry out a partial gauging of a specific lattice spin model using the bond algebra isomorphism described in Sec. 2.2. We summarize the results in Table 2.

### 4.2.1 Dualizing topological partition functions

As in the Sec. 3.2, it is useful to define a $\mathbb{Z}_4$ gauge field, $A_1 \in Z^1(M, \mathbb{Z}_4)$ as a tuple of $\mathbb{Z}_2$ fields $\left(A_1^{(\mathsf{N})}, A_1^{(\mathsf{K})}\right)$ that satisfy the following modified cocycle conditions

$$dA_1^{(\mathsf{N})} = \text{Bock}\left(A_1^{(\mathsf{K})}\right) = \frac{1}{2}d\widetilde{A}_1^{(\mathsf{K})}, \qquad dA_1^{(\mathsf{K})} = 0, \tag{145}$$

where Bock denotes the Bockstein homomorphism (see for example App. B of [3] for details) and $\widetilde{A}_1^{(\mathsf{K})}$ is the lift of $A_1^{(\mathsf{K})}$ to a $\mathbb{Z}_4$ gauge field. The expressions for the topological partition

Table 2: Summary of dualities between two dimensional $\mathbb{Z}_{4,(0)}$-symmetric SRE gapped phases and $\left[\mathbb{Z}_{2,(0)}, \mathbb{Z}_{2,(1)}\right]^\epsilon$-symmetric gapped phases. The $\mathbb{Z}_{4,(0)}$-phases are labelled by $[\mathbb{Z}_{p,(0)}, \ell_p]$, where $\mathbb{Z}_p \subset \mathbb{Z}_4$ is the symmetry preserved by the ground state(s). We refer to the case $p = 1$ and 2 as spontaneously symmetry broken (SSB) and partial symmetry broken (PSB) respectively. The index $\ell_p \in \mathbb{Z}_p$ labels the SPT phase realized by the $4/p$-fold degenerate ground states, which we refer to as SPT($\ell_p$). The dual models are obtained by gauging the $\mathbb{Z}_2 \subset \mathbb{Z}_4$ and have a $\left[\mathbb{Z}_{2,(0)}, \mathbb{Z}_{2,(1)}\right]^\epsilon$ global 2-group symmetry with a mixed anomaly. Due to the anomaly, there is no gapped phase that preserves the full $\left[\mathbb{Z}_{2,(0)}, \mathbb{Z}_{2,(1)}\right]^\epsilon$ symmetry. When $p = 4$, the dual $\mathbb{Z}_{2,(1)}$ symmetry is broken while $\mathbb{Z}_{2,(0)}$ symmetry is preserved. The emergent $\mathbb{Z}_{2,(1)}$ symmetry generator carries topological spin $\theta = \pi \ell_4$. The $\mathbb{Z}_{2,(0)}$ fractionalizes on the anyonic excitations of the $\mathbb{Z}_{2,(1)}$ broken phase and the corresponding symmetry fractionalization (SF) pattern is determined by $\ell_4$. When $p = 2$, full $\left[\mathbb{Z}_{2,(0)}, \mathbb{Z}_{2,(1)}\right]^\epsilon$ global symmetry is broken. Each degenerate ground state that break $\mathbb{Z}_{2,(0)}$ symmetry supports $\mathbb{Z}_2$ topological order with an emergent $\mathbb{Z}_{2,(1)}$ symmetry carrying topological spin $\theta = \pi \ell_2$. By "Triv.", we mean the trivial group.

| $\mathbb{Z}_{4,(0)}$ SRE Phases | | Dual $\left[\mathbb{Z}_{2,(0)}, \mathbb{Z}_{2,(1)}\right]^\epsilon$ Gapped Phases | |
|---|---|---|---|
| Symmetry of GS | Description | Symmetry of GS | Description |
| $\left[\mathbb{Z}_{4,(0)}, \ell_4\right]$ | SPT($\ell_4$) | $\left[\mathbb{Z}_{2,(0)}, \ell_4\right]$ | Emergent $\mathbb{Z}_{2,(1)}$ w/ $\theta = \pi \ell_4$, SF=$\ell_4$ |
| $\left[\mathbb{Z}_{2,(0)}, \ell_2\right]$ | SPT($\ell_2$) + PSB | [Triv., $\ell_2$] | Emergent $\left(\mathbb{Z}_{2,(1)}, \ell_2\right)$ + SSB |
| [Triv., 0] | SSB | $\left[\mathbb{Z}_{2,(1)}, 0\right]$ | PSB |

functions for the seven gapped phases labelled by $[\mathbb{Z}_p, \ell]$ with $p \in \{1, 2, 4\}$ and $\ell \in \mathbb{Z}_p$ are

$$
\begin{aligned}
\mathcal{Z}_{\mathfrak{T}}^{[\mathbb{Z}_1,0]}\left[A_1^{(\mathrm{N})}, A_1^{(\mathrm{K})}\right] &= 4\delta_{A_1^{(\mathrm{N})},0}\delta_{A_1^{(\mathrm{K})},0}, \\
\mathcal{Z}_{\mathfrak{T}}^{[\mathbb{Z}_2,\ell]}\left[A_1^{(\mathrm{N})}, A_1^{(\mathrm{K})}\right] &= 2\delta_{A_1^{(\mathrm{K})},0} \exp\left\{i\pi\ell \int_M A_1^{(\mathrm{N})} \cup \mathrm{Bock}\left(A_1^{(\mathrm{N})}\right)\right\}, \\
\mathcal{Z}_{\mathfrak{T}}^{[\mathbb{Z}_4,\ell]}\left[A_1^{(\mathrm{N})}, A_1^{(\mathrm{K})}\right] &= \exp\left\{\frac{i\pi\ell}{2} \int_M \left(2A_1^{(\mathrm{N})} + A_1^{(\mathrm{K})}\right) \cup \mathrm{Bock}\left(A_1^{(\mathrm{N})}\right)\right\}.
\end{aligned}
\tag{146}
$$

The gapped phases dual to each of these phases, i.e., related via gauging of $\mathbb{Z}_2 \subset \mathbb{Z}_4$ can be obtained by mapping the topological partition functions using

$$
\mathcal{Z}_{\mathfrak{T}^\vee}^{[\mathbb{Z}_p,\ell]^\vee}\left[A_2^{(\mathrm{N}^\vee)}, A_1^{(\mathrm{K})}\right] = \frac{1}{2}\sum_{a_1^{(\mathrm{N})}} \mathcal{Z}_{\mathfrak{T}}^{[\mathbb{Z}_p,\ell]}\left[a_1^{(\mathrm{N})}, A_1^{(\mathrm{K})}\right](-1)^{\int_M a_1^{(\mathrm{N})} \cup A_2^{(\mathrm{N}^\vee)}}.
\tag{147}
$$

The topological partition function dual to the fully symmetry broken phase $[\mathbb{Z}_1, 0]$ is

$$
\mathcal{Z}_{\mathfrak{T}^\vee}^{[\mathbb{Z}_1,0]^\vee}\left[A_2^{(\mathrm{N}^\vee)}, A_1^{(\mathrm{K})}\right] = 2\delta_{A_1^{(\mathrm{K})},0}.
\tag{148}
$$

The factor $\delta_{A_1^{(\mathrm{K})},0}$ signals that the theory $\mathfrak{T}^\vee$ cannot be coupled to the background $A_1^{(\mathrm{K})}$ since the 0-form $\mathsf{K}_{(0)} = \mathbb{Z}_2$ symmetry is spontaneously broken. The prefactor of 2 corresponds to the ground state degeneracy owing to this spontaneous symmetry breaking. Furthermore

since there is no constraint on $A_2^{(\mathsf{N}^\vee)}$, the dual phase trivially preserves the 1-form $\mathsf{N}_{(1)}^\vee = \mathbb{Z}_2$ symmetry.

The partition function of the gapped phase dual to the partial symmetry broken phases takes the form

$$\mathcal{Z}_{\mathsf{T}^\vee}^{[\mathbb{Z}_2,\ell]^\vee}\Big[A_2^{(\mathsf{N}^\vee)},A_1^{(\mathsf{K})}\Big] = \overbrace{2\delta_{A_1^{(\mathsf{K})},0}}^{\text{0-form SSB}} \times \overbrace{\frac{1}{2}\sum_{a_1^{(\mathsf{N})}}(-1)^{\int_M\left[\ell a_1^{(\mathsf{N})}\cup\mathrm{Bock}\left(a_1^{(\mathsf{N})}\right)+a_1^{(\mathsf{N})}\cup A_2^{(\mathsf{N}^\vee)}\right]}}^{\text{Dijkgraaf-Witten (1-form SSB)}}. \tag{149}$$

The expression in the first brace corresponds to the fact that the 0-form symmetry $\mathsf{K}$ is broken in the gapped phase $[\mathbb{Z}_2,\ell]^\vee$ in $\mathfrak{T}^\vee$. The expression in the second brace corresponds to the Dijkgraaf-Witten partition function for a topological $\mathbb{Z}_2$ gauge theory with the topological action given by

$$S_{\mathrm{DW}}^{(\ell)}[a_1^{(\mathsf{N})}] = \mathrm{i}\pi\ell\int_M a_1^{(\mathsf{N})}\cup\mathrm{Bock}\left(a_1^{(\mathsf{N})}\right). \tag{150}$$

Equivalently, this is the deconfined phase of the (twisted) $\mathbb{Z}_2$ gauge theory which has an emergent 1-form symmetry. The emergent 1-form symmetry is manifest in the quantum double presentation of the theory in terms of cochains $b_1^{(\mathsf{N})}$ and $a_1^{(\mathsf{N})}$. In this presentation, the theory is described by the action

$$S_{\mathfrak{T}^\vee}^{[\mathbb{Z}_2,\ell]} = \mathrm{i}\pi\int_M\left[b_1^{(\mathsf{N})}\cup da_1^{(\mathsf{N})}+\ell a_1^{(\mathsf{N})}\cup\frac{da_1^{(\mathsf{N})}}{2}+A_2^{(\mathsf{N}^\vee)}\cup a_1^{(\mathsf{N})}+B_2^{(\mathsf{N})}\cup b_1^{(\mathsf{N})}\right]. \tag{151}$$

The most general topological line operator has the form

$$W_{(\mathsf{q},\mathsf{m})}(L) = \exp\left\{\mathrm{i}\pi\oint_L\left(\mathsf{q}\,a_1^{(\mathsf{N})}+\mathsf{m}\,b_1^{(\mathsf{N})}\right)\right\}, \tag{152}$$

with the emergent $\mathbb{Z}_2$ 1-form symmetry generated by $W_{0,1}(L)$. There is a non-trivial braiding between lines which is captured by the correlation function

$$\left\langle W_{(\mathsf{q}_1,\mathsf{m}_1)}(L_1)W_{(\mathsf{q}_2,\mathsf{m}_2)}(L_2)\right\rangle = \exp\left\{\mathrm{i}\pi\,\mathrm{Link}(L_1,L_2)(\mathsf{q}_1\mathsf{m}_2+\mathsf{q}_2\mathsf{m}_1+\ell\mathsf{m}_1\mathsf{m}_2)\right\}, \tag{153}$$

where $\mathrm{Link}(L_1,L_2)$ is the linking number between the 1-cycles $L_1$ and $L_2$. By inspecting this topological correlation function we learn two important things. Firstly, the fact that the line $W_{(0,1)}$ which is charged under the $\mathsf{N}_{(1)}^\vee = \mathbb{Z}_2$ 1-form symmetry is topological signals the spontaneous breaking of this 1-form symmetry. Secondly, the self-braiding of the emergent 1-form symmetry depends on the choice of $\ell$ and therefore distinguishes the two gapped phases that are dual to the two $\mathbb{Z}_2$ SPTs labelled by $\ell$. To summarize, the gapped phase $[\mathbb{Z}_2,\ell]^\vee$ in $\mathfrak{T}^\vee$ spontaneously breaks the 0-form symmetry $\mathsf{K}_{(0)} = \mathbb{Z}_2$ as well as the 1-form symmetry $\mathsf{N}_{(0)}^\vee = \mathbb{Z}_2$. There are two ways of breaking the 1-form symmetry that are distinguished by the choice of $\ell$ and, equivalently, by the self-braiding of the emergent 1-form symmetry generated by $W_{(0,1)}$.

Next, let us move on to the phases that are dual to $[\mathbb{Z}_4,\ell]$ under $\mathbb{Z}_2$ gauging of the $\mathbb{Z}_4$ symmetry. Under the gauging of $\mathsf{N}_{(0)}$, a gapped phase which preserves $\mathsf{K}_{(0)}$ maps to a dual phase which also preserves $\mathsf{K}_{(0)}$. Conversely, a phase that preserves $\mathsf{N}_{(0)}$ maps to a dual phase which breaks the dual symmetry $\mathsf{N}_{(1)}^\vee$. 1-form symmetry breaking phases are topologically ordered and it can be shown using (146) that the phase $[\mathbb{Z}_4,\ell]^\vee$, has the following quantum double action

$$\begin{aligned}S_{\mathfrak{T}^\vee}^{[\mathbb{Z}_4,\ell]} = &\ \mathrm{i}\pi\int_M\left[b_1^{(\mathsf{N})}\cup da_1^{(\mathsf{N})}+\frac{\ell}{2}a_1^{(\mathsf{N})}\cup da_1^{(\mathsf{N})}\right]\\ &+\mathrm{i}\pi\int_M\left[a_1^{(\mathsf{N})}\cup A_2^{(\mathsf{N}^\vee)}+b_1^{(\mathsf{N})}\cup B_2^{(\mathsf{N})}\right]+\frac{\mathrm{i}\pi\ell}{4}\int_M A_1^{(\mathsf{K})}\cup da_1^{(\mathsf{N})}.\end{aligned} \tag{154}$$

We note that this is again simply a $\mathbb{Z}_2$ Dijkgraaf-Witten theory with a topological action labelled by $\ell \bmod 2 \in H^3(\mathbb{Z}_2, U(1))$, therefore the topological line operators have the form (152) and the topological correlations functions are given by (153). There is however an additional subtlety due to the 0-form symmetry. Summing over $b_1^{(N)}$ imposes that $da_1^{(N)} = B_2^{(N)}$. We obtain a new term in the response theory of the form

$$\frac{i\pi\ell}{4} \int_M A_1^{(K)} \cup B_2^{(N)} \subset S_{\mathfrak{T}^\vee, \text{resp.}}^{[\mathbb{Z}_2, \ell]} \left[ A_2^{(N^\vee)}, A_1^{(K)}, B_2^{(N)} \right]. \tag{155}$$

This term signals that the emergent 1-form symmetry carries a fractional charge under $K_{(0)}$ with the fractionalization labelled by $\ell \in \mathbb{Z}_4$. Concretely this means that the $\mathbb{Z}_2$ eigenvalue of the $K_{(0)}$ symmetry operator squares to $\exp\{2\pi i\ell/4\}$ on the emergent 1-form symmetry generator.

### 4.2.2 Dualizing fixed-point Hamiltonians

We now describe how fixed point Hamiltonians in each gapped phase transform under gauging $\mathbb{Z}_2 \subset \mathbb{Z}_4$. The fixed-point Hamiltonians have the form (131) with $n = 4$, $p \in \{1, 2, 4\}$ and $\ell \in \mathbb{Z}_p$. The dual Hamiltonians after gauging $\mathbb{Z}_2 \subset \mathbb{Z}_4$ can be directly obtained by noting that the bond algebra of $\mathbb{Z}_4$ symmetric quantum systems (57) transforms into an isomorphic bond algebra in (79). Using this bond algebra isomorphsim, any $\mathbb{Z}_4$ symmetric Hamiltonian can be mapped to its dual counterpart under the partial gauging.

Under the bond-algebra isomorphism, the fully symmetry breaking fixed point Hamiltonian

$$\mathcal{H}_{[\mathbb{Z}_1, 0]} = -\frac{1}{2} \sum_e \left[ Z_{s(e)} Z_{t(e)}^\dagger + \text{H.c.} \right], \tag{156}$$

maps into the dual Hamiltonian

$$\mathcal{H}_{[\mathbb{Z}_1, 0]^\vee}^\vee = -\frac{1}{2} \sum_e \left[ \frac{1 - i\sigma_{s(e)}^z}{\sqrt{2}} \sigma_e^z \frac{1 + i\sigma_{t(e)}^z}{\sqrt{2}} + \text{H.c} \right] = -\sum_e \sigma_e^z P_e^{(+)}, \tag{157}$$

where $P^{(+)} = (1 + \sigma_{s(e)}^z \sigma_{t(e)}^z)/2$. This Hamiltonian has two ground states

$$\begin{aligned}
|\text{GS}\rangle_\uparrow &= \bigotimes_{e,v} |\sigma_e^z = \uparrow\rangle \otimes |\sigma_v^z = \uparrow\rangle, \\
|\text{GS}\rangle_\downarrow &= \bigotimes_{e,v} |\sigma_e^z = \uparrow\rangle \otimes |\sigma_v^z = \downarrow\rangle,
\end{aligned} \tag{158}$$

which spontaneously break the $\mathbb{Z}_2$ 0-form symmetry (80) and preserve the $\mathbb{Z}_2$ 1-form symmetry (81). Similarly, the fixed point Hamiltonian describing partial symmetry breaking from $\mathbb{Z}_4$ to $\mathbb{Z}_2$

$$\mathcal{H}_{[\mathbb{Z}_2, \ell]} = -\frac{1}{2} \sum_v X_v^2 \exp\left\{ \frac{2\pi i\ell}{4} \sum_{e \subset \partial \text{Hex}_v} \mathcal{B}_e \right\} - \frac{1}{2} \sum_e Z_{s(e)}^2 Z_{t(e)}^2 + \text{H.c.}, \tag{159}$$

maps into the dual Hamiltonian

$$\mathcal{H}_{[\mathbb{Z}_2, \ell]^\vee}^\vee = -\sum_v A_v \exp\left\{ \frac{\pi i\ell}{4} \sum_{e \subset \partial \text{Hex}_v} \left[ 1 - \frac{1 + i\sigma_{s(e)}^z}{\sqrt{2}} \sigma_e^z \frac{1 - i\sigma_{t(e)}^z}{\sqrt{2}} \right] \right\} - \sum_e \sigma_{s(e)}^z \sigma_{t(e)}^z. \tag{160}$$

The ground state properties of this Hamiltonian can be obtained by first, minimizing the term $\sigma_{s(e)}^z \sigma_{t(e)}^z$ by setting $\sigma_v^z = \pm 1$. This amounts to studying the model in either of the two super-selection sectors that break the $\mathbb{Z}_2$ 0-form symmetry

$$\mathcal{H}_{[\mathbb{Z}_2, \ell]^\vee}^\vee \bigg|_{\sigma_v^z = \pm 1} = -\sum_v A_v \exp\left\{ \frac{\pi i\ell}{4} (1 - \sigma_e^z) \right\}. \tag{161}$$

We note that this projected Hamiltonian corresponds to the $\mathbb{Z}_2$ Toric code and double semion topological order for $\ell = 0$ and $\ell = 1$ respectively [69, 135]. We thus conclude that upon partial gauging of the phase labelled as $[\mathbb{Z}_2, \ell]$, the dual Hamiltonian realizes $2^{b_0(M_2)+b_1(M_2)}$ ground states such that the contributions of $2^{b_0(M_2)}$ and $2^{b_1(M_2)}$ are due to 0-form and 1-form symmetry breaking respectively. Lastly, as was demonstrated by Levin and Gu [69], $\ell$ can be diagnosed by the self-braiding of the emergent topological line operator in the ground state subspace. Finally, we describe the duality of the fixed-point Hamiltonian $[\mathbb{Z}_4, \ell]$. For simplicity, we restrict to the case $\ell = 0$, in which case the fixed-point Hamiltonian has the form

$$\mathcal{H}_{[\mathbb{Z}_4,0]} = -\frac{1}{2}\sum_v \left[X_v + X_v^\dagger\right].$$ (162)

Under the bond-algebra isomorphism, (162) maps into the dual Hamiltonian

$$\mathcal{H}_{[\mathbb{Z}_4,0]^\vee}^\vee = -\sum_v \sigma_v^x \left[\frac{1+A_v}{2}\right],$$ (163)

which realizes ground states which are disordered product state in the vertex degrees of freedom while the edge degrees of freedom realize the Toric code or $\mathbb{Z}_2$ topological order ground state. There are a total of $2^{b_1(M)}$ ground states labelled by elements in $a \in H^1(M, \mathbb{Z}_2)$. Explicitly, the ground states have the form

$$|GS[a]\rangle = \bigotimes_v |\sigma_v^x = \rightarrow\rangle \otimes \left[\frac{1+A_v}{2}\right]|a\rangle,$$ (164)

where $|a\rangle$ is a reference state in the $\sigma_e^z$ basis such that

$$W_\gamma|a\rangle = (-1)^{a(\gamma)}|a\rangle, \qquad W(\gamma) = \prod_{e \subset \gamma} \sigma_e^z.$$ (165)

Such ground states preserve $\mathsf{K}_{(0)} = \mathbb{Z}_2$ and spontaneously break $\mathsf{N}_{(1)}^\vee = \mathbb{Z}_2$. Therefore, the symmetry breaking transition between the $\mathbb{Z}_4$ symmetric and fully symmetry broken phases realized by the minimal model $\mathcal{H}_{[\mathbb{Z}_4,0]} + \mathcal{H}_{[\mathbb{Z}_1,0]}$ maps to the dual model $\mathcal{H}_{[\mathbb{Z}_4,0]^\vee}^\vee + \mathcal{H}_{[\mathbb{Z}_1,0]^\vee}^\vee$, which realizes a topological deconfined transition between a $\mathbb{Z}_2$ topological order (Toric code) and a $\mathbb{Z}_{2,(0)}$ symmetry broken phase. While the $\mathbb{Z}_2$ 0-form and 1-form symmetry groups appear independent, they are related via the mixed anomaly, which is responsible for the direct unconventional transition between these two phases. In fact, the phase diagram of the anomalous $[\mathbb{Z}_{2,(0)}, \mathbb{Z}_{2,(1)}]^\epsilon$ symmetric spin model after partial gauging realizes many interesting unconventional transitions that can be understood by studying the more conventional transitions realized in the phase diagram of the $\mathbb{Z}_4$ symmetric spin model.

# 5 Phase diagrams and dualities in $d = 3$

In this section, we extend our analysis in Sec. 4 to the case of three-dimensional space. For the group $\mathsf{G} = \mathbb{Z}_n$, we describe how gapped phases realized in $\mathsf{G}$-symmetric spin models are mapped under dualities related to gauging either the full group $\mathsf{G}$ or a subgroup thereof. As in Sec. 4, we restrict ourselves to short range entangled gapped phases. In three dimensions, such phases are enumerated by tuples $[\mathsf{H}, \nu]$, where $\mathsf{H}$ is a subgroup of $\mathsf{G}$ and $\nu \in H^4(\mathsf{H}, \mathsf{U}(1))$. In a gapped phase labelled by $[\mathsf{H}, \nu]$, the subgroup $\mathsf{H}$ remains unbroken in the $|\mathsf{G}/\mathsf{H}|$ dimensional ground-state manifold and each ground state realizes an $\mathsf{H}$ SPT labelled by $\nu$.

It suffices to look at the simplest non-trivial case to illustrate the general features of how phases map under such dualities. Therefore, in what follows, we consider models with $\mathsf{G}_{(0)} = \mathbb{Z}_{4,(0)}$ 0-form symmetry and relate them to models with 2-form dual symmetries. For $\mathsf{G}_{(0)} = \mathbb{Z}_{4,(0)}$, there are no non-trivial SPT phases in three dimensions since $H^4(\mathbb{Z}_4, \mathsf{U}(1))$ is trivial. This simplification allows us to limit the subsequent analysis to topologically trivial gapped phases labelled by their symmetry breaking pattern. Such phases can be described by Hamiltonians

$$\mathcal{H}_{[\mathbb{Z}_p]} = -\frac{1}{2}\sum_{\mathsf{v}} X_{\mathsf{v}}^{4/p} - \frac{1}{2}\sum_{\mathsf{e}} Z_{\mathsf{s(e)}}^p Z_{\mathsf{t(e)}}^{-p} + \text{H.c.}, \tag{166}$$

$p = 1, 2, 4$. For a given $p$, the Hamiltonian $\mathcal{H}_{[\mathbb{Z}_p]}$ has gapped ground states that are $4/p$-fold degenerate. The global 0-form $\mathbb{Z}_4$ symmetry is broken down to $\mathbb{Z}_p$ symmetry. This follows from the fact that operators $X_{\mathsf{v}}^{4/p}$ and $Z_{\mathsf{s(e)}}^p$ commute with each other and degenerate ground states are characterized by the expectation value $\langle Z_{\mathsf{s(e)}}^p Z_{\mathsf{t(e)}}^{-p} \rangle = 1$. More generally one consider a superposition of Hamiltonians

$$\mathcal{H}[\{\lambda\}] = \sum_{\mathbb{Z}_p} \lambda_p \, \mathcal{H}_{[\mathbb{Z}_p]}, \tag{167}$$

$\lambda_p \in \mathbb{R}$, which can access all three gapped phases and transitions between them.

## 5.1 Gauging finite Abelian symmetry

As described in Sec. 2 and Sec. 3, upon gauging a $\mathbb{Z}_{4,(0)}$ symmetry in $d = 3$, there is a $\mathbb{Z}_{4,(2)}$ symmetry in the dual gauged model. Here we describe the mapping of short range entangled phases under such a duality, which is summarized in Table 3. The ground state properties of Hamiltonians in a gapped phase labelled as $[\mathbb{Z}_p]$ are captured by the topological partition functions

$$\mathcal{Z}_{\mathfrak{T}}^{[\mathbb{Z}_p]}[A_1] = \frac{n}{p}\delta_{pA_1,0}. \tag{168}$$

The dual partition function is obtained by inserting (168) in (85), which delivers

$$\mathcal{Z}_{\mathfrak{T}^\vee}^{[\mathbb{Z}_p]^\vee} = \frac{1}{n}\sum_{a_1 \in H^1(M,\mathbb{Z}_n)} \delta_{p\,a_1,0} = \frac{1}{p}\sum_{a_1 \in H^1(M,\mathbb{Z}_p)} 1 = \frac{1}{p}\sum_{\substack{a_1 \in C^1(M,\mathbb{Z}_p) \\ b_2 \in C^2(M,\mathbb{Z}_p)}} \exp\left\{\frac{2\pi i}{p}\int_M b_2 \cup da_1\right\}. \tag{169}$$

The $\mathbb{Z}_4$ 2-form symmetry is generated by $\exp\left\{i\oint_\gamma a_1\right\}$. In the last equality in (169), we have used the quantum double formulation in terms of 1 and 2-cochains $a_1 \in C^1(M,\mathbb{Z}_p)$ and $b_2 \in C^2(M,\mathbb{Z}_p)$ respectively. Summing over $b_2$ gives back the second equality. The merit of the quantum double description is that it makes an emergent 1-form symmetry in $[\mathbb{Z}_p]^\vee$ for $p \neq 1$ manifest. More precisely, in this formulation there are topological line and surface operators

$$W_L^{\mathsf{q}} = \exp\left\{iq\oint_L a_1\right\}, \qquad T_{S^{(2)}}^{\mathsf{m}} = \exp\left\{im\oint_{S^{(2)}} b_2\right\}, \tag{170}$$

which have the correlation functions [131, 132, 136]

$$\langle W_L^{\mathsf{q}} T_{S^{(2)}}^{\mathsf{m}} \cdots \rangle = \exp\left\{\frac{2\pi i \text{Link}(L, S^{(2)})}{p}\right\} \langle \cdots \rangle, \tag{171}$$

Table 3: Summary of dualities between $\mathbb{Z}_{4,(0)}$-symmetric models with short range entangled (SRE) ground states (GS) and $\mathbb{Z}_{4,(2)}$-symmetric models with gapped ground states in $d = 3$ space dimensions. The SRE ground states of $\mathbb{Z}_{4,(0)}$-symmetric Hamiltonians are described by the subgroup $[\mathbb{Z}_{p,(0)}]$ with $p = 1, 2, 4$ which denotes the symmetry preserved by the $4/p$-fold degenerate ground states. We refer to the case $p = 1$ as spontaneously symmetry broken (SSB), while the case of $p = 2$ is referred to as partial symmetry broken (PSB). In space dimension $d = 3$, there are no SPT phases protected by $\mathbb{Z}_{4,(0)}$ symmetry. The dual models are obtained by gauging the $\mathbb{Z}_{4,(0)}$ global symmetry. On the dual side, a phase preserving $\mathbb{Z}_{p,(0)}$ symmetry maps to a phase preserving $\mathbb{Z}_{4/p,(2)} \subset \mathbb{Z}_{4,(2)}$ subgroup. In the topologically ordered ground states ($p = 2, 4$), there is an emergent $\mathbb{Z}_{p,(1)}$ symmetry. We refer by "Triv." to trivial symmetry group.

| $\mathbb{Z}_{4,(0)}$ SRE Phases | | Dual $\mathbb{Z}_{4,(2)}$ Gapped Phases | |
|---|---|---|---|
| Symmetry of GS | Description | Symmetry of GS | Description |
| $\left[\mathbb{Z}_{4,(0)}\right]$ | Symmetry preserving | Triv. | Emergent $\mathbb{Z}_{4,(1)}$ |
| $\left[\mathbb{Z}_{2,(0)}\right]$ | PSB | $\left[\mathbb{Z}_{2,(2)}\right]$ | Emergent $\mathbb{Z}_{2,(1)}$ |
| Triv. | SSB | $\left[\mathbb{Z}_{4,(2)}\right]$ | Symmetry preserving |

where $\cdots$ denotes any other operators in the correlation function and we have assumed $L$ and $S^{(2)}$ do not link with any other operators in $\cdots$. In other words, $T_{S^{(2)}}$ is charged under the 2-form symmetry. Additionally since $T_{S^{(2)}}^p = 1$ and $T_{S^{(2)}}$ is topological therefore $T_{S^{(2)}}$ generates an emergent 1-form symmetry. The existence of a topological charged operator signals that the $\mathbb{Z}_{4,(2)}$ symmetry is broken down to $\mathbb{Z}_{4/p,(2)}$.

The fixed point Hamiltonians (166) are mapped to the following dual Hamiltonians under the isomorphism between the bond algebras (31) and (40)

$$\mathcal{H}_{[\mathbb{Z}_p,0]^{\vee}}^{\vee} = -\sum_{\mathsf{v}} A_{\mathsf{v}}^{4/p} - \sum_{\mathsf{e}} Z_{\mathsf{e}}^p + \text{H.c.} \tag{172}$$

In the spin model the dual 2-form symmetry $\mathbb{Z}_{4,(2)}$ is generated by

$$W_L = \prod_{\mathsf{e} \subset L} Z_{\mathsf{e}}^{\mathsf{o}(\mathsf{e},L)}, \tag{173}$$

with $L$ being any 1-cycle [recall Eq. (42)] and $\mathsf{o}(\mathsf{e}, L) = \pm 1$ denotes the orientation of the edge $\mathsf{e}$ relative to $L$. In the ground state manifold, $A_{\mathsf{v}}^{4/p}$ have a unit expectation value. This defines a topological surface operators in the low energy description. In comparison with (170), an operator defined on 2-cycles $S^{(2)}$, a 2-cycle on the dual lattice may be defined as

$$T_{S^{(2)},\vee}^{\mathsf{m}} = \prod_{\mathsf{v}} A_{\mathsf{v}}^{4/p \times \text{Link}(S^{(2),\vee},\mathsf{v})}. \tag{174}$$

Such an operator is charged under the 2-form symmetry. This is to say that 2-form $\mathbb{Z}_{4,(2)}$ symmetry is spontaneously broken down to $\mathbb{Z}_{4/p,(2)}$.

When $p = 1$, the first term in (172) vanishes. The Hamiltonian then describes a "2-form paramagnet" with a non-degenerate and gapped ground state. This is nothing but the Higgs phase of the 1-form gauge theory [137]. This phase preserves the dual 2-form symmetry. As we shall see, when $p = 2, 4$, the dual 2-form symmetry is spontaneously broken because of which the ground state manifold supports topological order.

When $p = 2, 4$, the ground state of Hamiltonian (172) is obtained by simultaneously minimizing $Z_\mathsf{e}^p$ and $A_\mathsf{v}^{4/p}$ terms. Recall that the Hilbert space on which the Hamiltonian acts is constrained by the condition

$$\prod_{\mathsf{e} \subset L} Z_\mathsf{e}^{\mathsf{o}(\mathsf{e},L)} = 1 \,, \tag{175}$$

where $L$ is any contractible 1-cycle which is required such that the dimension of Hilbert space is $4^{|M_{3,\triangle}|}$, the same as the dimension of pre-gauged Hilbert space. Minimizing the second term restricts the ground state manifold to the subspace where $Z_\mathsf{e}^p = +1$ is satisfied. Let us denote this restricted subspace by $\mathcal{V}_\mathrm{rest.}^{4/p}$, i.e.,

$$\mathcal{V}_\mathrm{rest.}^{4/p} := \mathcal{V}_\mathrm{ext} \Big|_{Z_\mathsf{e}^p = 1} \,. \tag{176}$$

On the Hilbert space $\mathcal{V}_\mathrm{rest.}^{4/p}$, operators $A_\mathsf{v}^{4/p}$ and $\prod_{\mathsf{e} \subset L} Z_\mathsf{e}$ act as $\mathbb{Z}_p$-valued variables. The configurations in $\mathcal{V}_\mathrm{rest.}^{4/p}$, can be spanned by eigenstates $|a\rangle$ of $Z_\mathsf{e}$ such that

$$Z_\mathsf{e} |a\rangle = \exp\left\{ \frac{2\pi i a_\mathsf{e}}{p} \right\} |a\rangle \,, \qquad a \in C^1(M_{3,\triangle}, \mathbb{Z}_p) \,. \tag{177}$$

The constraint (175) imposes that, in fact $a \in Z^1(M_3, \mathbb{Z}_p)$, i.e., $\mathrm{d}a = 0$. In turn, the the operator $A_\mathsf{v}^{4/p}$ acts on a configuration $a$ as

$$A_\mathsf{v}^{4/p} |a\rangle = |a + \mathrm{d}\delta^{(\mathsf{v})}\rangle \,, \tag{178}$$

where $\delta^{(\mathsf{v})} \in C^0(M_3, \mathbb{Z}_p)$. Hence, the constraint $A_\mathsf{v}^{4/p} = +1$ together with Eq. (175) is satisfied on the states

$$|[a]\rangle = \frac{1}{\left| C^0(M_3, \mathbb{Z}_p) \right|} \sum_{\lambda \in C^0(M_3, \mathbb{Z}_p)} |a + d\lambda\rangle \,, \qquad [a] \in H^1(M_3, \mathbb{Z}_p) \,, \tag{179}$$

where the states $|[a]\rangle$ are labeled by the cohomology group $H^1(M_3, \mathbb{Z}_p)$, i.e., set of all $\mathbb{Z}_p$-valued 1-cycles $[a \in Z^1(M_3, \mathbb{Z}_p)]$ up to 1-coboundaries $[\mathrm{d}\lambda \in Z^1(M_3, \mathbb{Z}_p)]$. The topological ground state degeneracy is then given by the cardinality of $H^1(M_3, \mathbb{Z}_p)$, i.e.,

$$|H^1(M_3, \mathbb{Z}_p)| = p^{b_2(M_3)} \,, \tag{180}$$

where $b_2(M_3)$ is the second Betti number of 3-manifold $M_3$. The corresponding topological order supported by the ground state manifold is the deconfined phase of the $d = 3$ $\mathbb{Z}_p$ topological gauge theory or equivalently, the $\mathbb{Z}_p$ $d = 3$ Toric code. Table 3 gives a summary of phases described by fixed-point Hamiltonians in $d = 3$ and their respective duals under gauging $\mathbb{Z}_{4,(0)}$ symmetry.

## 5.2 Gauging finite Abelian sub-symmetry

We now describe the dualities related to gauging the $\mathbb{Z}_{2,(0)} \subset \mathbb{Z}_{4,(0)}$ symmetry. We will focus on Hamiltonians dual to the fixed-point Hamiltonians (166) for $p = 1, 2, 4$. The phases described by these Hamiltonians and their respective duals are summarized in Table 4. As described in

Table 4: Summary of dualities between $\mathbb{Z}_{4,(0)}$-symmetric models with SRE ground states and $\left[\mathbb{Z}_{2,(0)}, \mathbb{Z}_{2,(2)}\right]^{\epsilon}$-symmetric models with gapped ground states in $d = 3$ space dimensions. The SRE ground states of $\mathbb{Z}_{4,(0)}$-symmetric Hamiltonians are described by the subgroup $[\mathbb{Z}_{p,(0)}]$ with $p = 1, 2, 4$ which denotes the symmetry preserved by the $4/p$-fold degenerate ground states. We refer to the case $p = 1$ as spontaneously symmetry broken (SSB), while the case of $p = 2$ is referred to as partial symmetry broken (PSB). In space dimension $d = 3$, there are no SPT phases protected by $\mathbb{Z}_{4,(0)}$ symmetry. The dual models are obtained by gauging the $\mathbb{Z}_{2,(0)} \subset \mathbb{Z}_{4,(0)}$ subgroup of the global symmetry. On the dual side, a phase preserving the $\mathbb{Z}_{2,(0)}$ subgroup is mapped to a phase where dual $\mathbb{Z}_{2,(2)}$ symmetry is broken, and the converse also holds. The remaining $\mathbb{Z}_{2,(0)}$ global symmetry is either broken or preserved on both sides of the duality. There is a mixed anomaly between the $\mathbb{Z}_{2,(2)}$ dual symmetry and the remaining $\mathbb{Z}_{2,(0)}$ global symmetry. Therefore, on the dual side a gapped phase that is symmetric under $\left[\mathbb{Z}_{2,(0)}, \mathbb{Z}_{2,(2)}\right]^{\epsilon}$ cannot be realized. When $p = 2, 4$, $\mathbb{Z}_{2,(2)}$ dual symmetry is broken in the ground state manifold which supports an emergent $\mathbb{Z}_{2,(1)}$ symmetry. We refer by "Triv." to trivial symmetry group.

| $\mathbb{Z}_{4,(0)}$ SRE Phases | | Dual $\left[\mathbb{Z}_{2,(0)}, \mathbb{Z}_{2,(2)}\right]^{\epsilon}$ Gapped Phases | |
|---|---|---|---|
| Symmetry of GS | Description | Symmetry of GS | Description |
| $\left[\mathbb{Z}_{4,(0)}\right]$ | Symmetry preserving | $\left[\mathbb{Z}_{2,(0)}\right]$ | Emergent $\mathbb{Z}_{2,(1)}$ |
| $\left[\mathbb{Z}_{2,(0)}\right]$ | PSB | Triv. | Emergent $\mathbb{Z}_{2,(1)}$ + SSB |
| Triv. | SSB | $\left[\mathbb{Z}_{2,(2)}\right]$ | PSB |

Sec. 3.2, the corresponding dual models are symmetric under a 2-form dual symmetry $\mathbb{Z}_{2,(2)}$ and a residual 0-form symmetry $\mathbb{Z}_{2,(0)}$. Using the isomorphism between the bond algebras (31) and (79), we identify the generator of the remaining 0-form symmetry to be (recall Eq. (80))

$$\mathcal{U} = \prod_{\mathsf{v}} u_{\mathsf{v}}, \qquad u_{\mathsf{v}} = \sigma_{\mathsf{v}}^x \left[ P_{\mathsf{v}}^{(+)} + A_{\mathsf{v}} P_{\mathsf{v}}^{(-)} \right], \tag{181}$$

while the dual 2-form symmetry generator is (recall Eq. (81))

$$\mathcal{W}_L = \prod_{\mathsf{e} \subset L} w_{\mathsf{e}}^{\mathsf{o}(\mathsf{e}, L)}, \qquad w_e = \frac{1}{2} \left(1 - i\sigma_{\mathsf{s}(\mathsf{e})}^z\right) \sigma_{\mathsf{e}}^z \left(1 + i\sigma_{\mathsf{t}(\mathsf{e})}^z\right). \tag{182}$$

The fixed-point Hamiltonian (166) describes a phase where $\mathbb{Z}_{4,(0)}$ symmetry is spontaneously broken down to $\mathbb{Z}_{p,(0)}$ subgroup.

When $p = 1$, under the isomorphism between the bond algebras (31) and (79), the fixed point Hamiltonian (166) is mapped to

$$\mathcal{H}_{[\mathbb{Z}_1, 0]^{\vee}}^{\vee} = -\sum_{\mathsf{e}} \sigma_{\mathsf{e}}^z \frac{1 + \sigma_{\mathsf{s}(e)}^z \sigma_{\mathsf{t}(e)}^z}{2}. \tag{183}$$

There are two ground states of this Hamiltonian given by

$$|\mathrm{GS}\rangle_{\uparrow} = \bigotimes_{\mathsf{e},\mathsf{v}} |\sigma_{\mathsf{e}}^z = \uparrow\rangle \otimes |\sigma_{\mathsf{v}}^z = \uparrow\rangle,$$

$$|\mathrm{GS}\rangle_{\downarrow} = \bigotimes_{\mathsf{e},\mathsf{v}} |\sigma_{\mathsf{e}}^z = \uparrow\rangle \otimes |\sigma_{\mathsf{v}}^z = \downarrow\rangle. \tag{184}$$

The two-dimensional ground state manifold is separated from excited states by a finite gap. Since $\sigma_e^z = 1$ in the ground states, the $\mathbb{Z}_{2,(2)}$ dual symmetry is unbroken in the ground state manifold. In contrast, the twofold degeneracy of the ground states is due to spontaneous breaking of the $\mathbb{Z}_{2,(0)}$ symmetry that survives after gauging. One verifies that the two ground states are mapped to each other under the generator of $\mathbb{Z}_{2,(0)}$ symmetry, i.e.,

$$\mathcal{U}|\text{GS}\rangle_\uparrow = |\text{GS}\rangle_\downarrow, \qquad \mathcal{U}|\text{GS}\rangle_\downarrow = |\text{GS}\rangle_\uparrow. \tag{185}$$

When $p = 2$, under the isomorphism between the bond algebras (31) and (79), the fixed point Hamiltonian (166) is mapped to

$$\mathcal{H}_{[\mathbb{Z}_2, 0]^\vee}^\vee = -\sum_v A_v - \sum_e \sigma_{s(e)}^z \sigma_{t(e)}^z. \tag{186}$$

The degrees of freedom on vertices and on edges are decoupled. The ground state properties can be obtained by minimizing each term separately. The second term acts only on the vertices and imposes the constraint $\sigma_{s(e)}^z \sigma_{t(e)}^z = +1$ which has two solutions $\sigma_v^z = \pm 1$. Therefore, the vertex degrees of freedom are ferromagnetically ordered. The two ground states on the vertices are

$$
\begin{aligned}
|\text{GS}_{\text{vrt}}\rangle_\uparrow &= \bigotimes_v |\sigma_v^z = \uparrow\rangle, \\
|\text{GS}_{\text{vrt}}\rangle_\downarrow &= \bigotimes_v |\sigma_v^z = \downarrow\rangle.
\end{aligned}
\tag{187}
$$

The $\mathbb{Z}_{2,(0)}$ symmetry is clearly spontaneously broken by the ferromagnetically ordered ground states. The first term acts only on the edge degrees of freedom. The bond algebra (79) requires the condition

$$\prod_{e \subset L} \sigma_e^z = 1 \tag{188}$$

to hold on any contractible 1-cycle. Configurations of edge degrees of freedom that satisfy this condition are one-to-one with 1-cocycles $|b\rangle$ such that $b \in Z^1(M_3, \mathbb{Z}_2)$

$$\sigma_e^z|b\rangle = (-1)^{b_e}|b\rangle. \tag{189}$$

In turn, the configuration $|b\rangle$ is mapped to

$$A_v|b\rangle = |b + d\delta^{(v)}\rangle, \tag{190}$$

under the action of $A_v$, where $\delta^v \in C^0(M_3, \mathbb{Z}_2)$. Hence, the edge degrees of freedom support the ground states

$$|[b]\rangle = \frac{1}{|C^0(M_3, \mathbb{Z}_2)|} \sum_{\lambda \in C^0(M_3, \mathbb{Z}_2)} |b + d\lambda\rangle, \qquad [b] \in H^1(M_3, \mathbb{Z}_2), \tag{191}$$

where the states $|[b]\rangle$ are labeled by the cohomology group $H^1(M_3, \mathbb{Z}_2)$. The topological ground state degeneracy is then given by the cardinality of $H^1(M_3, \mathbb{Z}_2)$, i.e.,

$$|H^1(M_3, \mathbb{Z}_2)| = 2^{b_1(M_3)} = 2^{b_2(M_3)}, \tag{192}$$

where $b_p(M_3)$ is the $p^{\text{th}}$ Betti number of 3-manifold $M_3$. This is the ground state of three-dimensional Toric code, on which $\mathbb{Z}_{2,(2)}$ symmetry is spontaneously broken. The corresponding order parameter is any products of $A_v$ which is supported on 2-cycles and has a non-vanishing/topological ground state expectation value. Together the $2^{b_2(M_3)+1}$ dimensional total ground state manifold is spanned by the states

$$\left\{ |\text{GS}_{\text{vrt}}\rangle_\uparrow \otimes |[b]\rangle, \, |\text{GS}_{\text{vrt}}\rangle_\downarrow \otimes |[b]\rangle \right\}. \tag{193}$$

When $p = 4$, under the isomorphism between the bond algebras (31) and (79), the fixed point Hamiltonian (166) is mapped to

$$\mathcal{H}^\vee_{[\mathbb{Z}_4, 0]^\vee} = -\sum_{\mathsf{v}} \sigma^x_{\mathsf{v}} \frac{1 + A_{\mathsf{v}}}{2}. \tag{194}$$

The ground state of Hamiltonian (194) is similar to that of Hamiltonian (186). Using the same argument we observe that the condition $A_{\mathsf{v}} = +1$ must be satisfied and the ground state of three-dimensional Toric code is stabilized on the edge degrees of freedom. However, as opposed to Hamiltonian (186), there is no conventional order supported by the ground state and the vertex degrees of freedom realize a paramagnet. Therefore, the ground state manifold is spanned by the states

$$\left( \bigotimes_{\mathsf{v}} |\sigma^x_{\mathsf{v}} = \rightarrow\rangle \right) \otimes |[b]\rangle, \qquad [b] \in H^1(M_3, \mathbb{Z}_2), \tag{195}$$

with the total degeneracy $2^{b_2(M_3)}$ that is only due to the topological order supported on the edges. The dual $\mathbb{Z}_{2,(2)}$ symmetry is spontaneously broken while the remaining $\mathbb{Z}_{2,(0)}$ symmetry is preserved.

# 6 Gauging 1-form (sub) symmetry

In this section, we shift our attention to describing the gauging of 1-form finite Abelian (sub)-symmetries in quantum spin models. We will closely follow the approach in Sec. 2 and Sec. 3 adapted to higher-form symmetries. Higher-form symmetries have been useful in providing non-perturbative constraints that help solve the phase diagrams of quantum gauge theories [55, 56, 67, 137]. The goal of this section is to study how the phase diagrams of 1-form $\mathbb{Z}_n$ symmetric quantum spin models map under a duality related to gauging either the full or partial $\mathbb{Z}_n$ 1-form symmetry.

## 6.1 Gauging finite Abelian 1-form symmetry

Let us consider a quantum spin system defined on a $d = 2$ or 3 dimensional oriented lattice $M_{d,\triangle}$. For concreteness, we work with a square and cubic lattice in $d = 2$ and 3 respectively, with the orientation convention as in Fig. 13. However, the analysis in this section can be generalized straightforwardly to any other lattice and dimension. We are interested in describing spin systems with 1-form symmetries, i.e., those that are implemented by co-dimension-2 operators in spacetime and act on line operators [4], or more generally on operators defined on loci of dimension greater than one [47]. In the Hamiltonian presentation of a quantum spin model, 1-form symmetries are generated by co-dimension-1 operators in space that commute with the Hamiltonian.

Let us restrict to $\mathbb{Z}_n$ 1-form symmetries, for which we consider a Hilbert space $\mathcal{V}$ to be the tensor product of Hilbert spaces $\mathcal{V}_{\mathsf{e}}$ assigned to each edge of the square or cubic lattice

$$\mathcal{V} = \bigotimes_{\mathsf{e}} \mathcal{V}_{\mathsf{e}}, \qquad \mathcal{V}_{\mathsf{e}} \cong \mathbb{C}^n. \tag{196}$$

The algebra of operators acting on $\mathcal{V}_{\mathsf{e}}$ is generated by $X_{\mathsf{e}}$ and $Z_{\mathsf{e}}$ which satisfy the $\mathbb{Z}_n$ clock and shift algebra analogous to (6). The 1-form symmetry in $d$ dimensions is generated by the following operators defined on a closed and oriented $(d-1)$-dimensional sub-lattice $\mathsf{S}^{(d-1),\vee}$ of the dual lattice

$$\mathcal{U}_{\mathsf{g}}(\mathsf{S}^{(d-1),\vee}) = \prod_{\mathsf{e} \in \mathsf{S}^{(d-1),\vee}} X_{\mathsf{e}}^{\mathsf{g} \, \mathsf{o}(\mathsf{e}, \mathsf{S}^{(d-1),\vee})}, \tag{197}$$

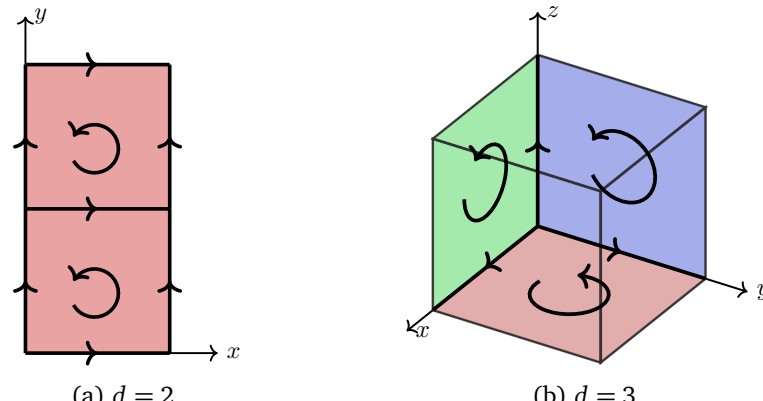

(a) $d = 2$.        (b) $d = 3$.

Figure 13: We pick the orientation convention in which an edge along the $\mu \in \{x, y, z\}$ direction is oriented in the positive $\mu$ direction. In 3 dimensions, the plaquettes in the $xy$, $yz$ and $zx$ planes are oriented in the positive $z$, $x$ and $y$ directions respectively.

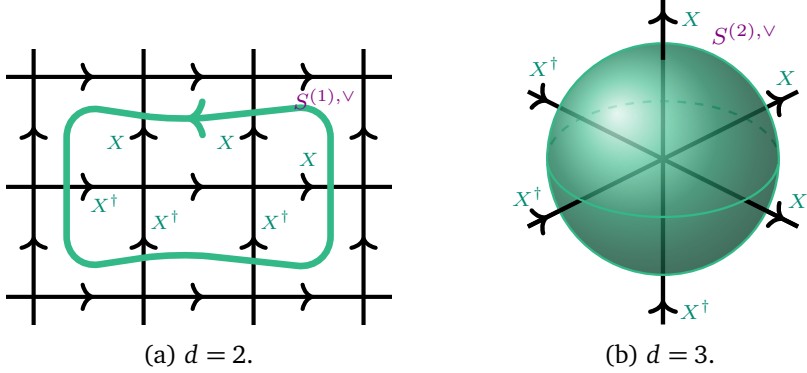

(a) $d = 2$.        (b) $d = 3$.

Figure 14: The figure depicts a 1-form symmetry generator in (197) in $d = 2$ and 3 dimensions defined on the line and surface $\mathsf{S}^{(1),\vee}$ and $\mathsf{S}^{(2),\vee}$ respectively.

where $\mathrm{o}(\mathsf{e}, \mathsf{S}^{(d-1),\vee})$ denotes the orientation of $\mathsf{e}$ with respect to the orientation (outward normal) of $\mathsf{S}^{(d-1),\vee}$ (see Fig. 14). The 1-form symmetry operators satisfy $\mathbb{Z}_n$ composition rules when defined on the same line or surface such that

$$\mathcal{U}_{\mathsf{g}_1}(\mathsf{S}^{(d-1),\vee}) \times \mathcal{U}_{\mathsf{g}_2}(\mathsf{S}^{(d-1),\vee}) = \mathcal{U}_{\mathsf{g}_1 + \mathsf{g}_2}(\mathsf{S}^{(d-1),\vee}), \qquad \mathsf{g}_1, \mathsf{g}_2 \in \mathbb{Z}_n, \tag{198}$$

while more generally two co-dimension-1 operators fuse at codimension-2 junctions according to the group composition in $\mathbb{Z}_n$ (see Fig. 15). The algebra of operators that commute with any such network of symmetry defects is the bond algebra

$$\mathsf{B}_{\mathbb{Z}_{n,(1)}}(\mathcal{V}) = \left\langle X_\mathsf{e}, B_\mathsf{p} \,\middle|\, \mathcal{U}(\mathsf{S}^{(d-1),\vee}) \overset{!}{=} 1, \, \forall \, \mathsf{e}, \mathsf{p} \right\rangle, \tag{199}$$

where $B_p$ are operators defined on plaquettes or 2-cells of the lattice and have the form

$$B_\mathsf{p} = \prod_{\mathsf{e} \subset \mathsf{p}} Z_\mathsf{e}^{\mathrm{o}(\mathsf{e}, \mathsf{p})}. \tag{200}$$

The product is over the edges on the boundary of $\mathsf{p}$ and $\mathrm{o}(\mathsf{e}, \mathsf{p})$ is the orientation of $\mathsf{e}$ with respect to the orientation of $\mathsf{p}$. This operator may be familiar from the Toric code Hamiltonian [135] for the case of $n = 2$. See Fig. 16 for the explicit form of the $B_\mathsf{p}$ operators in terms



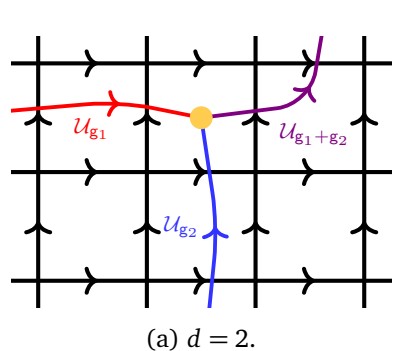
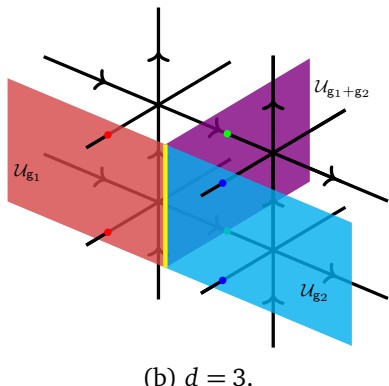

(a) $d = 2$.                     (b) $d = 3$.

Figure 15: The figure depicts the fusion of 1-form symmetry generators in $d = 2$ and 3 dimensions. Two codimension-1 surface operators corresponding to $g_1, g_2 \in G$ (in red an blue respectively) fuse to a codimension-1 surface operator corresponding to $g_1 + g_2$ mod n (in purple) via a co-dimension-2 junction operator (depicted in yellow).

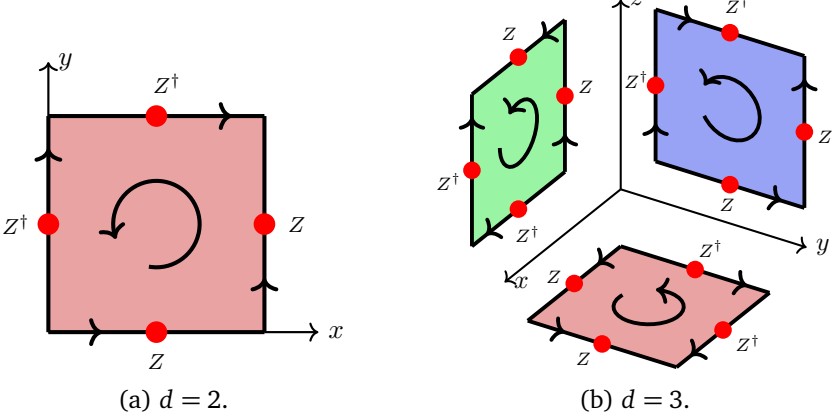

(a) $d = 2$.                     (b) $d = 3$.

Figure 16: The figure depicts the plaquette operator $B_p$ in (200) in (a) $d = 2$ dimensions and (b) $d = 3$ dimensions for the three types of plaquettes in the $xy$, $yz$ and $zx$ planes depicted in pink, gray and green respectively.

of the $\mathbb{Z}_n$ clock and shift operators. The bond algebra (199) is defined on a Hilbert space with constraints on contractible $(d-1)$-cycles on the dual lattice labelled as $S^{(d-1),\vee}$. Such constraints are important for two reasons: (i) they ensure that the bond algebra isomorphism related to gauging the $\mathbb{Z}_{n,(1)}$ symmetry is invertible and (ii) after gauging $\mathbb{Z}_{n,(1)}$, (199) maps to a bond algebra which has a trivial background of symmetry twist defects of the dual symmetry. Instead if one relaxes the constraints, and imposes some other fixed but non-trivial assignment of symmetry eigenvalues of $\mathcal{U}(S^{(d-1),\vee})$, after gauging $\mathbb{Z}_{n,(1)}$ this becomes a non-trivial background of symmetry twist defects of the dual symmetry.

Now, we gauge the $\mathbb{Z}_n$ 1-form symmetry by introducing $\mathbb{Z}_n$ gauge degrees of freedom on the plaquettes of the lattice. We thus obtain the extended Hilbert space

$$\mathcal{V}_{\text{ext}} = \bigotimes_e \mathcal{V}_e \bigotimes_p \mathcal{V}_p = \text{Span}_{\mathbb{C}} \left\{ |b, a\rangle \,\middle|\, b \in C^2(M_{d,\triangle}, \mathbb{Z}_n), \, a \in C^1(M_{d,\triangle}, \mathbb{Z}_n) \right\}, \quad (201)$$

such that the clock and shift operators act on the basis states as

$$
\begin{aligned}
Z_e|b, a\rangle &= \omega_n^{a_e}|b, a\rangle, & X_e|b, a\rangle &= |b, a + \delta^{(e)}\rangle, \\
Z_p|b, a\rangle &= \omega_n^{b_p}|b, a\rangle, & X_p|b, a\rangle &= |b + \delta^{(p)}, a\rangle.
\end{aligned}
\quad (202)
$$

Here, $\delta^{(\mathsf{p})}$ is a $\mathbb{Z}_n$-valued 2-cochain such that

$$\left[\delta^{(\mathsf{p})}\right]_{\mathsf{p}'} = \delta_{\mathsf{p},\mathsf{p}'}\,, \tag{203}$$

and $\delta^{(\mathsf{e})}$ was introduced in (25). Additionally, one needs to impose gauge invariance via the Gauss operators (see Fig. 17)

$$\mathcal{G}_{\mathsf{e}} = X_{\mathsf{e}} A_{\mathsf{e}}^{\dagger}\,, \qquad A_{\mathsf{e}}^{\dagger} := \prod_{\mathsf{p} \supset \mathsf{e}} X_{\mathsf{p}}^{\mathsf{o}(\mathsf{e},\mathsf{p})}\,, \tag{204}$$

where the product is over plaquettes that contain the edge $\mathsf{e}$ on their boundary and $\mathsf{o}(\mathsf{e},\mathsf{p}) = 1$ or $-1$ depending on whether the boundary of $\mathsf{p}$ is oriented along or against the edge $\mathsf{e}$. The most general Gauss operator can be parametrized by a 1-cochain $\lambda \in C^1(M_{d,\triangle},\mathbb{Z}_n)$ as $\mathcal{G}[\lambda] = \prod_{\mathsf{e}} \mathcal{G}_{\mathsf{e}}^{\lambda_{\mathsf{e}}}$. Such a Gauss operator implements the gauge transformation

$$\mathcal{G}[\lambda] : |b,a\rangle \longrightarrow |b + \mathrm{d}\lambda, a + \lambda\rangle\,, \tag{205}$$

where $(\mathrm{d}\lambda)_{\mathsf{p}} = \sum_{\mathsf{e} \subset \mathsf{p}} \mathsf{o}(\mathsf{e},\mathsf{p})\lambda_{\mathsf{e}}$. We note that the plaquette degrees of freedom $Z_{\mathsf{p}}$ and $X_{\mathsf{p}}$ embody a $\mathbb{Z}_n$ 2-form gauge field and "electric field" respectively, while the edge degrees of freedom embody the $\mathbb{Z}_n$ 1-form charged matter. The physical space of states and operators are invariant under the action of $\mathcal{G}[\lambda]$ for all $\lambda \in C^1(M_{d,\triangle},\mathbb{Z}_n)$.

In order to construct the gauged bond algebra, we need to consider operators that are gauge invariant. In particular, the operators $B_p$ in the bond algebra (199) are not gauge invariant and need to be minimally coupled to the 2-form gauge field $Z_{\mathsf{p}}$ as $B_{\mathsf{p}} \longmapsto B_{\mathsf{p}} Z_{\mathsf{p}}^{\dagger}$. The other generator, $X_{\mathsf{e}}$ of (199) is gauge invariant as is and therefore the bond algebra after gauging the $\mathbb{Z}_n$ 1-form symmetry is

$$\widetilde{\mathsf{B}}_{\mathbb{Z}_{n,(1)}}(\mathcal{V}_{\mathrm{ext}}) = \langle X_{\mathsf{e}}, B_{\mathsf{p}} Z_{\mathsf{p}}^{\dagger} \,\Big|\, \mathcal{U}(S^{(d-1),\vee}) \overset{!}{=} 1,\, \mathcal{G}_{\mathsf{e}} \overset{!}{=} 1,\, \prod_{p \subset S^{(2)}} \left[B_{\mathsf{p}} Z_{\mathsf{p}}^{\dagger}\right]^{\mathsf{o}(\mathsf{p},S^{(2)})} \overset{!}{=} 1,\, \forall\, \mathsf{e},\mathsf{p}\rangle\,, \tag{206}$$

where $S^{(2)}$ is any contractible 2-cycle and we impose the constraint

$$\prod_{p \subset S^{(2)}} \left[B_{\mathsf{p}} Z_{\mathsf{p}}^{\dagger}\right]^{\mathsf{o}(\mathsf{p},S^{(2)})} = 1\,, \tag{207}$$

since this operator is in the image of $\prod_{p \subset S^{(2)}} B_{\mathsf{p}}^{\mathsf{o}(\mathsf{p},S^{(2)})} = 1$. As before, the Gauss constraints can be removed via a unitary transformation that makes the Gauss operators local (on the edge degrees of freedom). More precisely, the unitary acts as follows

$$\begin{aligned} U X_{\mathsf{e}} U^{\dagger} &= X_{\mathsf{e}} A_{\mathsf{e}}\,, & U Z_{\mathsf{e}} U^{\dagger} &= Z_{\mathsf{e}}\,, \\ U Z_{\mathsf{p}} U^{\dagger} &= Z_{\mathsf{p}} B_{\mathsf{p}}\,, & U X_{\mathsf{p}} U^{\dagger} &= X_{\mathsf{p}}\,. \end{aligned} \tag{208}$$

Importantly, under the unitary action, the Gauss operator transforms as $U\mathcal{G}_{\mathsf{e}}U^{\dagger} = X_{\mathsf{e}}$ and therefore effectively freezes out the edge degrees of freedom. After the unitary transformation, the bond algebra is represented on the Hilbert space built up of only the plaquette degrees of freedom and has the form

$$\begin{aligned} \widetilde{\mathfrak{B}}'_{\mathbb{Z}_{n,(1)}}(\mathcal{V}_{\mathrm{plaq}}) &= U \widetilde{\mathfrak{B}}_{\mathbb{Z}_{n,(1)}}(\mathcal{V}_{\mathrm{ext}}) U^{\dagger} \\ &= \langle A_{\mathsf{e}}, Z_{\mathsf{p}}^{\dagger} \,\Big|\, \prod_{\mathsf{e} \in S^{(d-1),\vee}} A_{\mathsf{e}}^{\mathsf{o}(\mathsf{e},S^{(d-1),\vee})} \overset{!}{=} 1,\, \prod_{p \subset S^{(2)}} Z_{\mathsf{p}}^{\mathsf{o}(\mathsf{p},S^{(2)})} \overset{!}{=} 1,\, \forall\, \mathsf{p}\rangle\,, \end{aligned} \tag{209}$$

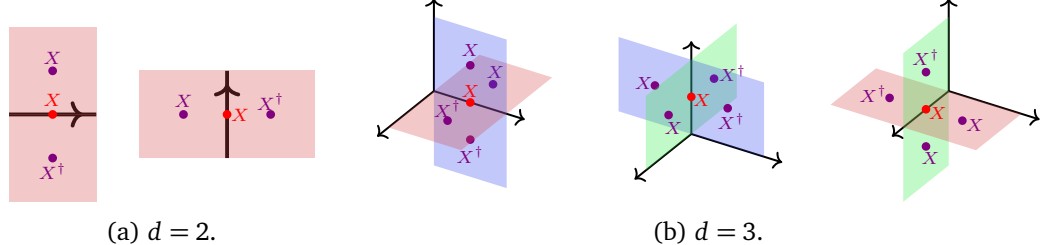

(a) $d = 2$.    (b) $d = 3$.

Figure 17: The figure depicts the Gauss operator $\mathcal{G}_{\mathsf{e}}$ associated to different kinds of edges for (a) $d = 2$ and (b) $d = 3$ respectively.

where $\mathcal{V}_{\mathrm{plaq}} = \otimes_{\mathsf{p}} \mathcal{V}_p \subset \mathcal{V}_{\mathrm{ext}}$. Note that the constraint

$$\prod_{\mathsf{e} \in S^{(d-1),\vee}} A_{\mathsf{e}}^{\mathsf{o}(\mathsf{e}, S^{(d-1),\vee})} \overset{!}{=} 1, \tag{210}$$

holds, unless we are working in a background of symmetry twist defects. We will henceforth leave this constraint implicit for brevity. Next, we implement a final isomorphism to bring (209) into a convenient form. This isomorphism comprises (i) the rotation $(Z_{\mathsf{p}}, X_{\mathsf{p}}) \to (X_{\mathsf{p}}^{-1}, Z_{\mathsf{p}})$ and (ii) the dualization of the square or cubic lattices. Recall that the plaquettes of a square or cubic lattices become the vertices or edges of the dual square or cubic lattices, respectively. Implementing this isomorphism we obtain the dual bond algebra, which in $d = 2$ dimensions is

$$\mathsf{B}_{\mathbb{Z}_{n,(0)}}^{\mathrm{dual}}(\mathcal{V}_{\mathrm{dual}}) = \langle Z_{\mathsf{s}(\mathsf{e})} Z_{\mathsf{t}(\mathsf{e})}^{\dagger}, X_{\mathsf{v}} \,\big|\, \forall\, \mathsf{e}, \mathsf{v} \rangle \qquad (d = 2). \tag{211}$$

This is nothing but the bond algebra of $\mathbb{Z}_n$ 0-form symmetric operators on the dual square lattice. Note that there is no constraint on contractible cycles $S^{(2)}$, since there are no contractible cycles on a closed two dimensional manifold. Furthermore, since we already discussed the mapping of sectors and the phases between 0-form symmetric and 1-form symmetric algebras/models in detail in Sec. 3 and Sec. 4, we will focus on the case of $d = 3$ in the remainder of this section.

For $d = 3$, the dual bond algebra has the form

$$\mathsf{B}_{\mathbb{Z}_{n,(1)}}^{\mathrm{dual}}(\mathcal{V}_{\mathrm{dual}}) = \langle B_{\mathsf{p}}, X_{\mathsf{e}} \,|\, \mathcal{U}(S^{(2),\vee}) \overset{!}{=} 1, \; \forall\, \mathsf{e}, \mathsf{p} \rangle \qquad (d = 3), \tag{212}$$

where edge operators in the $x$-direction in the original bond algebra dualize to plaquette operators in the $yz$ plane, and so on. The converse also holds as illustrated in Fig. 18. This is essentially a generalization of the Kramers-Wannier duality to 1-form $\mathbb{Z}_n$ symmetric models.

Just like the usual Kramers-Wannier duality, the symmetry sectors, i.e., symmetry eigenspaces and symmetry twisted boundary conditions map non-trivially under this automorphism of the bond algebra. Let us consider the following symmetry twisted partition function for a theory $\mathfrak{T}$ with $\mathbb{Z}_{n,(1)}$ symmetry on a manifold $M = M_3 \times S^1$ coupled to a background gauge field $A_2 \in H^2(M, \mathbb{Z}_n)$

$$\mathcal{Z}_{\mathfrak{T}}[A_2] \equiv \mathcal{Z}_{\mathfrak{T}}[\vec{\mathsf{g}}, \vec{\mathsf{h}}] = \mathrm{Tr}\left[ \prod_{j=1}^{b_2(M_3)} \mathcal{U}_{\mathsf{h}_j}(\Sigma_j^{(2)}) \exp\left\{ -\beta H_{\vec{\mathsf{g}}} \right\} \right]. \tag{213}$$

This equation needs some unpacking. Firstly, the gauge field $A_2$ can be labelled by its holonomies around non-contractible cycles in $H_2(M, \mathbb{Z})$, which, using the Künneth theorem,

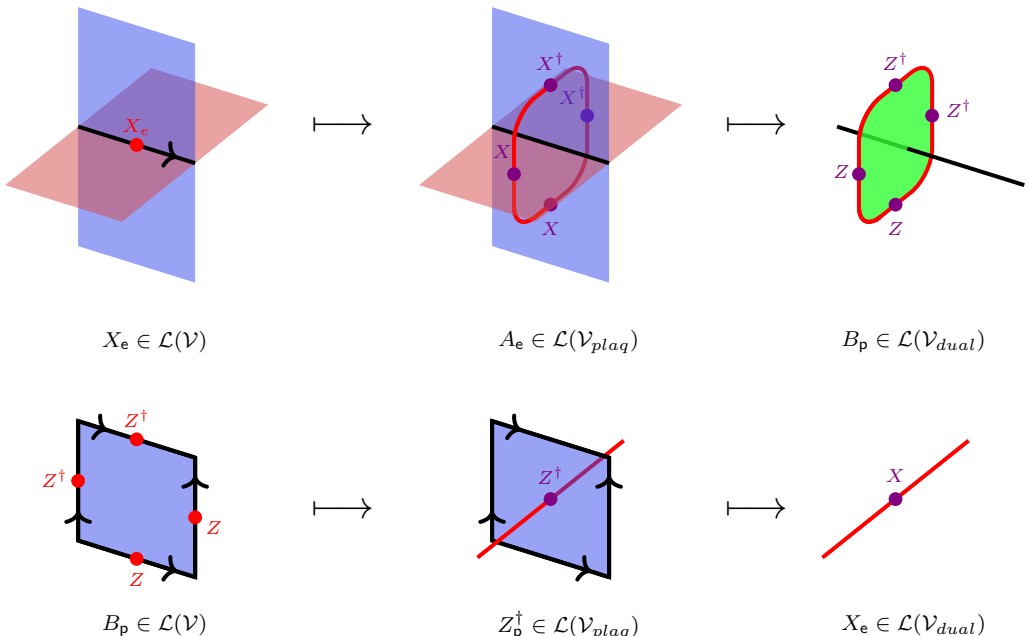

$$X_e \in \mathcal{L}(\mathcal{V}) \qquad A_e \in \mathcal{L}(\mathcal{V}_{plaq}) \qquad B_p \in \mathcal{L}(\mathcal{V}_{dual})$$

$$B_p \in \mathcal{L}(\mathcal{V}) \qquad Z_p^\dagger \in \mathcal{L}(\mathcal{V}_{plaq}) \qquad X_e \in \mathcal{L}(\mathcal{V}_{dual})$$

Figure 18: The figure depicts the Gauss operator $\mathcal{G}_e$ associated to different kinds of edges for (a) $d = 2$ and (b) $d = 3$ respectively.

$$
\begin{aligned}
H_2(S^1 \times M_3, \mathbb{Z}) &= H_2(M_3, \mathbb{Z}) \oplus H_1(M_3, \mathbb{Z}), \\
H_2(M_3, \mathbb{Z}) &= \operatorname{Span}_{\mathbb{Z}} \left\{ \Sigma_j^{(2)} \right\}, \\
H_1(M_3, \mathbb{Z}) &= \operatorname{Span}_{\mathbb{Z}} \left\{ \Sigma_j^{(1)} \right\}.
\end{aligned}
\tag{214}
$$

Therefore, we can label $A_2 = (\vec{g}, \vec{h})$ where $\vec{g} = (g_1, \dots, g_N)$ and $\vec{h} = (h_1, \dots, h_N)$ and $N = b_1(M_3) = b_2(M_3)$, such that

$$
\oint_{\Sigma_j^{(2)}} A_2 = g_j, \qquad \oint_{S_1 \times \Sigma_j^{(1)}} A_2 = h_j.
\tag{215}
$$

With the purpose of tracking how the symmetry sectors map under gauging $\mathbb{Z}_{n,(1)}$, we define a projector $\mathcal{P}_\alpha(\Sigma^{(2)})$, that projects onto the sub-Hilbert space that transforms in the $\alpha$ representation of the 1-form symmetry operator defined on the homology 2-cycle $\Sigma^{(2)}$.

$$
\mathcal{P}_\alpha(\Sigma^{(2)}) = \frac{1}{n} \sum_h \omega_n^{-\alpha h} \mathcal{U}_h(\Sigma^{(2)}).
\tag{216}
$$

Using (216), we may define a symmetry character $\chi[\vec{\alpha}, \vec{g}]$ as the thermal trace in a definite eigensector $\alpha_j$ of the 1-form symmetry and symmetry twisted boundary condition $g_j$ on $\Sigma_j^{(2)}$ as

$$
\chi_{\mathfrak{T}}[\vec{\alpha}, \vec{g}] = \operatorname{Tr} \left[ \prod_{j=1}^{b_2(M_3)} \mathcal{P}_{\alpha_j}(\Sigma_j^{(2)}) \exp\left\{ -\beta H_{\vec{g}} \right\} \right].
\tag{217}
$$

In the canonical approach, the following operator identities hold in this symmetry sector

$$
\mathcal{U}_h(\Sigma_j^{(2)}) = \omega_n^{h\alpha_j}, \qquad \mathcal{T}(\Sigma_j^{(2)}) := \prod_{p \subset \Sigma_j^{(2)}} B_p = \omega_n^{g_j}.
\tag{218}
$$

Table 5: Summary of Kramers-Wannier duality between gapped ground states of $\mathbb{Z}_{4,(1)}$-symmetric models in $d = 3$ space dimensions. We consider gapped ground states of $\mathbb{Z}_{4,(1)}$-symmetric Hamiltonians that preserve the subgroup $[\mathbb{Z}_{p,(1)}]$ with $p = 1, 2, 4$. Such phases have an emergent $\mathbb{Z}_{4/p,(2)}$ symmetry which is generated by closed surface loop operators. The dual models are obtained by gauging the $\mathbb{Z}_{4,(0)}$ global symmetry. On the dual side, a phase preserving $\mathbb{Z}_{p,(2)}$ symmetry maps to a phase preserving $\mathbb{Z}_{4/p,(1)} \subset \mathbb{Z}_{4,(1)}$ subgroup with an emergent $\mathbb{Z}_{p,(2)}$ symmetry. We refer by "Triv." to trivial symmetry group.

| $\mathbb{Z}_{4,(1)}$ Gapped Phases | | Dual $\mathbb{Z}_{4,(1)}$ Gapped Phases | |
|---|---|---|---|
| Symmetry of GS | Description | Symmetry of GS | Description |
| $\left[\mathbb{Z}_{4,(1)}\right]$ | Symmetry preserving | Triv. | Emergent $\mathbb{Z}_{4,(2)}$ |
| $\left[\mathbb{Z}_{2,(1)}\right]$ | Emergent $\mathbb{Z}_{2,(2)}$ | $\left[\mathbb{Z}_{2,(1)}\right]$ | Emergent $\mathbb{Z}_{2,(2)}$ |
| Triv. | Emergent $\mathbb{Z}_{4,(2)}$ | $\left[\mathbb{Z}_{4,(1)}\right]$ | Symmetry preserving |

Since gauging $\mathbb{Z}_{n,(1)}$ has the effect

$$\mathcal{U}_{\mathsf{h}}(\Sigma_j^{(2)}) \xleftrightarrow{\text{gauging } \mathbb{Z}_{n,(1)}} \mathcal{T}^{\mathsf{h}}(\Sigma_j^{(2)}), \tag{219}$$

the symmetry sectors in the original and gauged theory $\mathfrak{T}$ and $\mathfrak{T}^\vee$ map as

$$\chi_{\mathfrak{T}}[\vec{\alpha}, \vec{g}] = \chi_{\mathfrak{T}^\vee}[\vec{g}, \vec{\alpha}]. \tag{220}$$

This mapping of sectors can also be derived directly from topological gauging as described in Sec. 3. There, we recall that the partition function of the gauged theory coupled to a background $\mathbb{Z}_{n,(1)}$ gauge field $A_2^\vee$ has the form

$$\mathcal{Z}_{\mathfrak{T}^\vee}[A_2^\vee] = \frac{1}{n^{b_1(M) - b_0(M)}} \sum_{a_2} \mathcal{Z}_{\mathfrak{T}}[A_2] \exp\left\{ i \int_M a_2 \cup A_2^\vee \right\}. \tag{221}$$

### 6.1.1 Phase diagrams and the 1-form Kramers-Wannier duality in $d = 3$

The automorphism of the bond algebra of $\mathbb{Z}_{n,(1)}$ symmetric operators under gauging of the $\mathbb{Z}_{n,(1)}$ symmetry imposes strong constraints on the phase diagram. In particular, the spectrum of a Hamiltonian $\mathcal{H}$ in a symmetry sector $(\vec{\alpha}, \vec{g})$ is the same as the spectrum of a dual Hamiltonian $\mathcal{H}^\vee$ obtained by acting with the bond-algebra automorphism on $\mathcal{H}$, in the symmetry sector $(\vec{g}, \vec{\alpha})$. This has consequences for the both the $\mathbb{Z}_{n,(1)}$ symmetric gapped phases as well as the phase transitions.

Let us first focus on the gapped phase where the $\mathbb{Z}_n$ 1-form symmetry is spontaneously broken to a $\mathbb{Z}_p$ subgroup in the ground state. The fixed point Hamiltonian for such a gapped phase is

$$\mathcal{H}_{[\mathbb{Z}_p]} = -\frac{1}{2} \sum_{\mathsf{e}} X_{\mathsf{e}}^{n/p} - \frac{1}{2} \sum_{\mathsf{p}} B_{\mathsf{p}}^p + \text{H.c.} \tag{222}$$

Clearly all the operators in the Hamiltonian commute with one another and therefore the ground state(s) would be in the shared eigenvalue 1 subspace of all the operators appearing in (222). First, we may restrict to the subspace of $\mathcal{V}_{\text{rest.}}^{(n/p)} \subset \mathcal{V}$ on which $X_{\mathsf{e}}^{n/p} = 1$ is satisfied.

This effectively reduces the edge Hilbert space dimension to $n/p$. In this restricted Hilbert space, the Hamiltonian has the form

$$\mathcal{H}_{[\mathbb{Z}_p]}\Big|_{\mathcal{V}_{\text{rest.}}^{(n/p)}} = -\frac{1}{2}\sum_{\mathsf{p}} B_{\mathsf{p}}^p + \text{H.c.}, \tag{223}$$

which is an operator of order $n/p$, i.e.,

$$\left[B_{\mathsf{p}}^p\right]^{n/p}\Big|_{\mathcal{V}_{\text{rest.}}^{(n/p)}} = 1. \tag{224}$$

The Hamiltonian (223) in fact describes a $\mathbb{Z}_{n/p} \subset \mathbb{Z}_n$ topological gauge theory. One manifestation of this is that the Hamiltonian has $(n/p)^{b_2(M_3)}$ ground states, which are locally indistinguishable but mapped into each other under the action of topological line or surface operators. This can be seen by inspecting the ground state degeneracy on a spatial manifold $M_3$ with non-trivial topology.[13] First note that the states (or spin configurations) are one-to-one with 1-cochains

$$|a\rangle, \qquad a \in C^1\left(M_3, \mathbb{Z}_{n/p}\right), \tag{225}$$

where $a$ is nothing but an assignment of $\mathbb{Z}_{n/p}$ values on each edge (1-simplex). The action of the plaquette operators are

$$B_{\mathsf{p}}^p|a\rangle = \omega_{n/p}^{\mathrm{d}a_{\mathsf{p}}}|a\rangle. \tag{226}$$

Therefore the subspace of states that satisfy $B_{\mathsf{p}}^p|a\rangle = |a\rangle$ correspond to 1-cochains that satisfy $\mathrm{d}a = 0$. In other words, states such that $B_{\mathsf{p}}^p = 1$ are labeled by 1-cocycles $a \in Z^1\left(M_3, \mathbb{Z}_{n/p}\right)$. Furthermore, we are working in a restricted space (see (212)) where

$$\mathcal{U}(S_{\mathsf{v}}^{(2),\vee}) \overset{!}{=} 1, \tag{227}$$

where $\mathcal{U}(S_{\mathsf{v}}^{(2),\vee})$ is the 1-form symmetry generator defined on a minimal two sphere on the dual lattice that links with the vertex $\mathsf{v}$. The action of $\mathcal{U}(S_{\mathsf{v}}^{(2),\vee})$ on a state $|a\rangle$ without such a constraint is

$$\mathcal{U}\left(S_{\mathsf{v}}^{(2),\vee}\right)|a\rangle = |a + \mathrm{d}\delta^{(\mathsf{v})}\rangle, \tag{228}$$

where $\delta^{(\mathsf{v})} \in C^0\left(M_3, \mathbb{Z}_{n/p}\right)$. Hence a general product of $\mathcal{U}\left(S_{\mathsf{v}}^{(2),\vee}\right)$ implements a gauge transformation as

$$\prod_{\mathsf{v}}\mathcal{U}\left(S_{\mathsf{v}}^{(2),\vee}\right)^{\lambda_{\mathsf{v}}}|a\rangle = |a + \mathrm{d}\lambda\rangle. \tag{229}$$

States satisfying both $B_{\mathsf{p}}^p = 1$ and $\mathcal{U}\left(S_{\mathsf{v}}^{(2),\vee}\right) = 1$ are thus labeled by cohomology classes $H^1(M_3, \mathbb{Z}_{n/p}) = Z^1(M_3, \mathbb{Z}_{n/p})/B^1(M_3, \mathbb{Z}_{n/p})$. In particular

$$|[a]\rangle = \frac{1}{\left|C^0\left(M_3, \mathbb{Z}_{n/p}\right)\right|} \sum_{\lambda \in C^0\left(M_3, \mathbb{Z}_{n/p}\right)} |a + d\lambda\rangle, \qquad [a] \in H^1\left(M_3, \mathbb{Z}_{n/p}\right). \tag{230}$$

The ground-state degeneracy is thus

$$\left|H^1(M_3, \mathbb{Z}_{n/p})\right| = (n/p)^{b_1(M_3)} = (n/p)^{b_2(M_3)}. \tag{231}$$

The last equality comes from $H^1(M_3, \mathbb{Z}_{n/p}) \cong H^2(M_3, \mathbb{Z}_{n/p})$ for 3-manifolds. Note that $\mathrm{d}\lambda$ correspond to contractible loop configurations in the dual lattice with $\mathbb{Z}_{n/p}$ branching rules while $a + \mathrm{d}\lambda$ are loop configurations that wrap around non-contractible cycles according to the cohomology class of $a$. In other words, the states (230) are nothing but string-net condensates. This

---

[13]For simplicity, we assume $M_3$ has no torsion.

condensate of strings is another manifestation of spontaneous breaking of 1-form symmetry. The ground-state degeneracy is due to existence of line operators for each $\gamma \in H_1(M_3, \mathbb{Z}_{n/p})$ and surface operators for each $\Sigma^{(2)} \in H_1(M_3, \mathbb{Z}_{n/p})$ that commute with the Hamiltonian but not each other [138]. Concretely, these line operators are

$$W(\gamma) = \prod_{e \in \gamma} Z_e^{o(\gamma, e)}, \tag{232}$$

while the surface operators are the 1-form symmetry generators (197) defined on $\Sigma^{(2)}$. These operators generate an emergent 2-form symmetry and are charged under the $\mathbb{Z}_{n/p}$ 1-form symmetry. This signals a breaking of $\mathbb{Z}_{n/p}$ 1-form symmetry and the ground state of (222) only preserves $\mathbb{Z}_p \subset \mathbb{Z}_n$, spontaneously breaking the remaining group.

Under $\mathbb{Z}_n$ 1-form Kramers-Wannier duality, one obtains a dual Hamiltonian to (222) which is

$$\mathcal{H}^{\vee}_{[\mathbb{Z}_p]^{\vee}} = -\frac{1}{2} \sum_e X_e^p - \frac{1}{2} \sum_p B_p^{n/p} - \frac{1}{2} \sum_v \mathcal{U}(S_v^{(2),\vee}) + \text{H.c} \cong \mathcal{H}_{[\mathbb{Z}_{n/p}]}. \tag{233}$$

We therefore learn that under such a duality,

$$[\mathbb{Z}_{n/p,(1)} \text{ symmetry breaking}] \xleftrightarrow{\text{gauging } \mathbb{Z}_{n,(1)}} [\mathbb{Z}_{p,(1)} \text{ symmetry breaking}]. \tag{234}$$

The phases described by fixed-point Hamiltonians (222) and their duals (233) are summarized in Table 5 when $n = 4$ and $p = 1, 2, 4$.

Another interesting application of dualities is to the study of phase transitions. Dualities are particularly useful when there are multiple symmetry breaking phases. In such cases, dualities between different kinds of transitions can be used constrain the universality classes of transitions, assuming knowledge about a subset of transitions. For instance, consider a theory with $\mathbb{Z}_n = \mathbb{Z}_{p_1 p_2 p_3,(1)}$, 1-form symmetry with $p_1, p_2, p_3$ prime numbers. Then the 1-form Kramer's Wannier duality predicts:

1. The 1-form symmetry breaking transition between the fully symmetric phase $[\mathbb{Z}_{p_1 p_2 p_3,(1)}]$ and the partial symmetry broken phase $[\mathbb{Z}_{p_1 p_3,(1)}]$ is dual to the transition between the fully-symmetry broken phase $[\mathbb{Z}_{1,(1)}]$ and the partial symmetry broken phase $[\mathbb{Z}_{p_2,(1)}]$. There are two additional dualities obtained by cyclic permutations of $(1, 2, 3)$. Note that these are dualities between transitions involving 3+1 dimensional topological orders and are analogous to anyon condensation type transitions in $2 + 1$ dimensional topological orders [114, 115, 139].

2. The deconfined topological transitions between the partial symmetry broken phases $[\mathbb{Z}_{p_1 p_2,(1)}]$ and $[\mathbb{Z}_{p_2 p_3,(1)}]$ is dual to the deconfined topological transition between $[\mathbb{Z}_{p_3,(1)}]$ and $[\mathbb{Z}_{p_1,(1)}]$. There are two additional dualities obtained by cyclic permutations of $(1, 2, 3)$.

3. The 1-form partial symmetry breaking transition between the phases $[\mathbb{Z}_{p_1 p_2,(1)}]$ and $[\mathbb{Z}_{p_1,(1)}]$ is dual to a similar transition between $[\mathbb{Z}_{p_2 p_3,(1)}]$ and $[\mathbb{Z}_{p_3,(1)}]$. Again, there are two additional dualities obtained by cyclic permutations of $(1, 2, 3)$.

4. Additionally there are several transitions that are self-dual under gauging $\mathbb{Z}_{n,(1)}$, These are:

   (a) The symmetry breaking transitions between the fully symmetry broken and the fully symmetric phases $[\mathbb{Z}_{1,(1)}]$ and $[\mathbb{Z}_{p_1 p_2 p_3,(1)}]$ respectively.

(b) The deconfined topological transition between $[\mathbb{Z}_{p_1 p_2,(1)}]$ and $[\mathbb{Z}_{p_3,(1)}]$. Again, there are two additional dualities obtained by cyclic permutations of $(1,2,3)$. Since there are no 't Hooft anomalies constraining the phase diagram, one would expect such transitions to be accidental.

In principle there are several other dualities related to partial gauging of some subgroup of $\mathbb{Z}_n$. We will describe these in some detail in the next section. Such dualities are powerful as they constrain the spectra of transitions involving combinations of topologically ordered phases in $3 + 1$ dimensions, a subject about which little is known.

The self-dual transitions are particularly interesting from a symmetry point of view. It was recently appreciated [25], that theories that are self-dual under gauging a higher-form symmetry host non-invertible symmetries that are higher-dimensional higher-form generalizations of the Tambara-Yamagami fusion category [140]. The simplest such self-dual Hamiltonian is the transition between the symmetry breaking transition between the symmetric and fully symmetry broken phases, described by the minimal Hamiltonian

$$\mathcal{H} = -\frac{1}{2}\sum_e X_e - \frac{1}{2}\sum_p B_p + \text{H.c.} \tag{235}$$

Then it is expected that this Hamiltonian has an emergent non-invertible symmetry operator $\mathcal{D}$ which acts on all of space and has the fusion rules

$$\mathcal{D} \times \mathcal{D} = \prod_{j=1}^{b_2(M_3)} \sum_{\mathsf{g}_j=0}^{n-1} \left[ \mathcal{U}_{\mathsf{g}_j}(\Sigma^{(2)})_j \right]. \tag{236}$$

We leave a detailed study of non-invertible symmetry defects in 1-form Kramer's Wannier self-dual lattice models for future work.

## 6.2 Gauging finite Abelian 1-form sub-symmetry

In this section, we describe the gauging of a sub-symmetry of a finite Abelian 1-form symmetry. As in earlier sections, we focus on the simplest non-trivial case, which corresponds to gauging a $\mathbb{Z}_{2,(1)}$ subgroup of a $\mathbb{Z}_{4,(1)}$ 1-form symmetry. Our starting point is the bond algebra of $\mathbb{Z}_{4,(1)}$ symmetric operators defined in (199) with $n = 4$.

A $\mathbb{Z}_{4,(1)}$ symmetry can be understood via the short exact sequence

$$1 \longrightarrow \mathsf{N}_{(1)} = \mathbb{Z}_{2,(1)} \longrightarrow \mathbb{Z}_{4,(1)} \longrightarrow \mathsf{K}_{(1)} = \mathbb{Z}_{2,(1)} \longrightarrow 1. \tag{237}$$

More precisely, $\mathbb{Z}_{4,(1)}$ should be understood as the second Eilenberg-Maclane space and the short exact sequence as that between homotopy 2-types. However, we will suppress such technicalities in our presentation. It suffices to note that such a sequence is captured by the extension class $\epsilon_2 \in H^3(\mathbb{Z}_{2,(1)}, \mathbb{Z}_{2,(1)})$, where $\epsilon_2 = \text{Bock}$. Then a $\mathbb{Z}_{4,(1)}$ bundle can be expressed as a tuple of 2-form gauge fields $A_2^{(\mathsf{N})}$ and $A_2^{(\mathsf{K})}$ which satisfy

$$dA_2^{(\mathsf{N})} = \text{Bock}\left(A_2^{(\mathsf{K})}\right), \qquad \text{Bock}\left(A_2^{(\mathsf{K})}\right) = \frac{1}{2}d\widetilde{A}_2^{(\mathsf{K})}. \tag{238}$$

Starting from a $d + 1$ dimensional theory $\mathfrak{T}$ with $\mathbb{Z}_{4,(1)}$ symmetry, one may gauge to $\mathsf{N}_{(1)}$ to obtain a dual theory $\mathfrak{T}^\vee$ with a symmetry group $\mathsf{N}_{(d-2)}^\vee \times \mathsf{K}_{(1)}$ where $\mathsf{N}^\vee = \mathsf{K} = \mathbb{Z}_2$. Furthermore, there is a mixed anomaly between the two $\mathbb{Z}_2$ symmetries captured by the $d + 2$ dimensional invertible topological field theory

$$\int_{M_{d+2}} A_{d-1}^{(\mathsf{N}^\vee)} \cup \text{Bock}(A_2^{(\mathsf{K})}). \tag{239}$$

Such an invertible field theory describes the ground state physics of a symmetry protected topological phase of matter protected by

$$G_{(1,d-2)}^{\epsilon_2} = \left[\mathbb{Z}_{2,(1)}, \mathbb{Z}_{2,(d-2)}\right]^{\epsilon_2}. \tag{240}$$

It is however crucial to emphasize that there is no physical need to associate $\mathfrak{T}^\vee$ to a 'bulk'. As we will see, such a theory can well be described on an $d$-dimensional lattice model. Instead the bulk or anomaly theory is a theoretical gadget to systematize our understanding of the anomaly, which has significant non-perturbative implications for the infra-red phases/ground states realized in $\mathfrak{T}^\vee$.

Now, we describe the gauging of $\mathbb{Z}_{2,(1)} \subset \mathbb{Z}_{4,(1)}$ on the lattice. Our starting point is the bond algebra (199) with $n = 4$. In order to gauge the $\mathbb{Z}_{2,(1)}$ subgroup, we introduce $\mathbb{Z}_2$ degrees of freedom on each plaquette of the lattice. We thus obtain the extended Hilbert space

$$\mathcal{V}_{\text{ext}} = \bigotimes_e \mathcal{V}_e \bigotimes_p \mathcal{V}_p = \text{Span}_{\mathbb{C}} \left\{ |b,a\rangle \,\middle|\, b \in C^2(M_\triangle, \mathbb{Z}_2), \ a \in C^1(M_\triangle, \mathbb{Z}_4) \right\}, \tag{241}$$

such that

$$
\begin{aligned}
Z_e|b,a\rangle &= i^{a_e}|b,a\rangle, & X_e|b,a\rangle &= |b,a+\delta^{(e)}\rangle, \\
\sigma_p^z|b,a\rangle &= (-1)^{b_p}|b,a\rangle, & \sigma_p^x|b,a\rangle &= |b+\delta^{(p)},a\rangle,
\end{aligned}
\tag{242}
$$

where, $\delta^{(e)}$ and $\delta^{(p)}$ were introduced in (25) and (203), respectively. We impose the Gauss constraint through the operator

$$\mathcal{G}_e = X_e^2 A_e, \qquad A_e := \prod_{p \supset e} \sigma_p^x. \tag{243}$$

The gauge-invariant bond algebra is obtained by minimally coupling the bond algebra (199) as

$$\widetilde{B}'_{G_{(1,d-2)}^{\epsilon_2}}(\mathcal{V}_{\text{ext}}) = \langle X_e, B_p\,\sigma_p^z \,\middle|\, \mathcal{U}_{(1)}(S^{(d-1),\vee}) \overset{!}{=} 1, \mathcal{U}_{(d-2)}(S^{(2)}) \overset{!}{=} 1, \mathcal{G}_e \overset{!}{=} 1 \rangle, \tag{244}$$

where $\mathcal{U}_{(1)}\left(S^{(d-1),\vee}\right)$ and $\mathcal{U}_{(d-2)}(S^{(2)})$ are the 1-form and $d-2$-form symmetry generators defined on contractible cycles $S^{(d-1),\vee}$ and $S^{(2)}$ respectively. These take the following form

$$\mathcal{U}_{(1)}\left(S^{(d-1),\vee}\right) = \prod_{e \in S^{(d-1),\vee}} X_e^{o(e,S^{(d-1),\vee})}, \qquad \mathcal{U}_{(d-2)}(S^{(2)}) = \prod_{p \subset S^{(2)}} (B_p \sigma_p^z)^{o(p,S^{(2)})}. \tag{245}$$

Importantly, there is a mixed anomaly between the 1-form and $(d-2)$-form symmetry. This mixed anomaly manifests itself in the symmetry fractionalization patterns of the two symmetries. More precisely, although the 1-form symmetry corresponds to the group $\mathbb{Z}_{2,(1)}$, it fractionalizes into the group $\mathbb{Z}_{4,(1)}$ in the presence of a non-trivial background gauge field of the $(d-2)$-symmetry. Conversely, the $\mathbb{Z}_{2,(d-2)}$ symmetry fractionalizes into the group $\mathbb{Z}_{4,(d-2)}$ in the presence of a non-trivial background of the $\mathbb{Z}_{2,(1)}$ symmetry. To see this, note that the way background fields $A_2$ and $A_{d-1}$ for the $\mathbb{Z}_{2,(1)}$ and $\mathbb{Z}_{2,(d-2)}$ symmetries appear in the bond algebra is via the minimal coupling

$$B_p \longmapsto B_p e^{i\pi A_{2,p}}, \qquad A_e \longmapsto A_e e^{i\pi A_{d-1,e}}. \tag{246}$$

It is worth emphasizing, that $A_{d-1,e}$ should be understood as the integral/evaluation of $A_{d-1}$ on the $(d-1)$-cell on the dual lattice, which is dual to e. Then one obtains the important identities

$$
\begin{aligned}
\mathcal{U}_{(1)}\left(\Sigma^{(d-1),\vee}\right)^2 &= \exp\left\{i\pi \oint_{\Sigma^{(d-1),\vee}} A_{d-1}\right\}, \\
\mathcal{U}_{(d-2)}(\Sigma^{(2)}) &= \exp\left\{i\pi \oint_{\Sigma^{(2)}} A_2\right\},
\end{aligned}
\tag{247}
$$

where $\Sigma^{(d-1),\vee}$ and $\Sigma^{(2)}$ are non-contractible cycles. Having a non-trivial holonomy for $A_{d-1}$ or $A_2$ on a given $(d-2)$- or 2-cycle simply corresponds to imposing symmetry twisted boundary conditions corresponding to $\mathbb{Z}_{2,(d-2)}$ or $\mathbb{Z}_{2,(1)}$ on that cycle, respectively. This is to say that the mixed anomaly manifest itself as the generators for $\mathbb{Z}_{2,(d-2)}$ or $\mathbb{Z}_{2,(1)}$ squaring to the operators that detect the twisted boundary conditions for $\mathbb{Z}_{2,(1)}$ or $\mathbb{Z}_{2,(d-2)}$ symmetries, respectively.

Next, as in Sec. 2.2, we solve the Gauss constraint by implementing a unitary transformation that localizes the Gauss operators onto the edges

$$U\,\mathcal{G}_{\mathsf{e}}\,U^{\dagger} = X_{\mathsf{e}}^{2}\,, \tag{248}$$

such that the unitary transformed Gauss constraint $X_{\mathsf{e}}^{2}=1$ can be readily solved. In the basis (242), this unitary operator has the form

$$U = \sum_{b,a}|b+\lfloor \mathrm{d}a/2\rfloor\rangle\langle b,a|\,, \tag{249}$$

where $\lfloor\cdot\rfloor$ is the floor function. Note that this operator is different than the unitary operator (74) since the coboundary operator $\mathrm{d}$ is inside the floor function $\lfloor\cdot\rfloor$. The remaining operators in the bond algebra transform as

$$
\begin{aligned}
UX_{\mathsf{e}}U^{\dagger} &= \sum_{b,a}|b+\lfloor \mathrm{d}a/2\rfloor+\lfloor \mathrm{d}(a+\delta^{(\mathsf{e})})/2\rfloor,a\rangle\langle b,a| \\
&= X_{\mathsf{e}}\left[P_{\mathsf{e}}^{+}+P_{\mathsf{e}}^{-}A_{\mathsf{e}}\right], \\
U\sigma_{\mathsf{p}}^{z}U^{\dagger} &= \sum_{b,a}(-1)^{b_{\mathsf{p}}+\lfloor \mathrm{d}a/2\rfloor_{\mathsf{p}}}|b,a\rangle\langle b,a| \\
&= \frac{1}{2}\sigma_{\mathsf{p}}^{z}\left[(1-\mathrm{i})B_{\mathsf{p}}+(1+\mathrm{i})B_{\mathsf{p}}^{\dagger}\right], \\
U\sigma_{\mathsf{p}}^{z}B_{\mathsf{p}}U^{\dagger} &= \sum_{b,a}(-1)^{b_{\mathsf{p}}+\lfloor \mathrm{d}a/2\rfloor_{\mathsf{p}}}\mathrm{i}^{(\mathrm{d}a)_{\mathsf{p}}}|b,a\rangle\langle b,a| \\
&= \frac{1}{2}\sigma_{\mathsf{p}}^{z}\left[(1-\mathrm{i})B_{\mathsf{p}}^{2}+(1+\mathrm{i})\right],
\end{aligned}
\tag{250}
$$

where $P_{\mathsf{e}}^{(\pm)}=(1\pm Z_{\mathsf{e}}^{2})/2$ and $A_{\mathsf{e}}=\prod_{\mathsf{p}\supset\mathsf{e}}\sigma_{\mathsf{p}}^{x}$. The unitarily transformed version of the bond algebra (244) therefore has the form

$$\widetilde{\mathfrak{B}}'_{\mathsf{G}^{\epsilon_{2}}_{(1,d-2)}}(\mathcal{V}_{\text{ext}}) = U\widetilde{\mathfrak{B}}_{\mathsf{G}^{\epsilon_{2}}_{(1,d-2)}}(\mathcal{V}_{\text{ext}})U^{\dagger} \tag{251}$$

$$= \left\langle X_{\mathsf{e}}\left[P_{\mathsf{e}}^{(+)}+P_{\mathsf{e}}^{(-)}A_{\mathsf{e}}\right], \frac{1}{2}\sigma_{\mathsf{p}}^{z}\left[(1-\mathrm{i})B_{\mathsf{p}}^{2}+(1+\mathrm{i})\right]\,\Big|\,\mathcal{U}_{(1)}(S^{(d-1),\vee})\stackrel{!}{=}1,\,\mathcal{U}_{(d-2)}(S^{(2)})\stackrel{!}{=}1,\,X_{\mathsf{e}}^{2}\stackrel{!}{=}1\right\rangle,$$

where the constraints are on the unitary transformed versions of the contractible symmetry operators defined in (245). The constraint $X_{\mathsf{e}}^{2}=1$ can be solved by projecting to the effective two dimensional Hilbert space on each edge on which $X_{\mathsf{e}}^{2}=1$. The operators $X_{\mathsf{e}}$ and $Z_{\mathsf{e}}^{2}$ commute with $X_{\mathsf{e}}^{2}$ and therefore act within this restricted subspace $\mathcal{V}_{\text{rest.}}$. We work in a basis where $X_{\mathsf{e}}\sim\sigma_{\mathsf{e}}^{x}$, $Z_{\mathsf{e}}^{2}\sim\sigma_{\mathsf{e}}^{z}$ and $B_{p}^{2}=\prod_{\mathsf{e}\in\mathsf{p}}\sigma_{\mathsf{e}}^{z}$ (see (78)) in $\mathcal{V}_{\text{rest.}}$. Therefore the bond algebra in its final form is

$$\mathfrak{B}_{\mathsf{G}^{\epsilon_{2}}_{(1,d-2)}}(\mathcal{V}_{\text{rest.}}) \tag{252}$$

$$= \left\langle \sigma_{\mathsf{e}}^{x}\left[\bar{P}_{\mathsf{e}}^{(+)}+\bar{P}_{\mathsf{e}}^{(-)}A_{\mathsf{e}}\right], \frac{1}{2}\sigma_{\mathsf{p}}^{z}\left[(1-\mathrm{i})\bar{B}_{\mathsf{p}}+(1+\mathrm{i})\right]\,\Big|\,\mathcal{U}_{(1)}(S^{(d-1),\vee})\stackrel{!}{=}1,\,\mathcal{U}_{(d-2)}(S^{(2)})\stackrel{!}{=}1\right\rangle,$$

where

$$\bar{P}_{\mathsf{e}}^{(\pm)} = \frac{1}{2}(1\pm\sigma_{\mathsf{e}}^{z}),\qquad \bar{B}_{\mathsf{p}} = \prod_{\mathsf{e}\in\mathsf{p}}\sigma_{\mathsf{e}}^{z}\,. \tag{253}$$

We are now ready to use the isomorphism of bond algebras (199) and (253) to study dualities between phase diagrams of quantum systems with $\mathbb{Z}_{4,(1)}$ symmetry and $\mathsf{G}^{\epsilon_{2}}_{(1,d-2)}$ symmetry.

Table 6: Summary of dualities between $\mathbb{Z}_{4,(1)}$-symmetric models with gapped ground states and $\left[\mathbb{Z}_{2,(0)}, \mathbb{Z}_{2,(d-2=1)}\right]^{\epsilon_2}$-symmetric models with gapped ground states in $d = 3$ space dimensions. We consider gapped ground states of $\mathbb{Z}_{4,(1)}$-symmetric Hamiltonians that preserve the subgroup $[\mathbb{Z}_{p,(1)}]$ with $p = 1, 2, 4$. Such phases have an emergent $\mathbb{Z}_{4/p,(2)}$ symmetry which is generated by closed surface loop operators. The dual models are obtained by gauging the $\mathbb{Z}_{2,(1)} \subset \mathbb{Z}_{4,(1)}$ subgroup of the global symmetry. On the dual side, a phase preserving the $\mathbb{Z}_{2,(1)}$ subgroup is mapped to a phase where dual $\mathbb{Z}_{2,(d-2=1)}$ symmetry is broken, and the converse also holds. The remaining $\mathbb{Z}_{2,(1)}$ global symmetry is either broken or preserved on both sides of the duality. There is a mixed anomaly between the $\mathbb{Z}_{2,(1)}$ dual symmetry and the remaining $\mathbb{Z}_{2,(1)}$ global symmetry. Therefore, on the dual side a gapped phase that is symmetric under $\left[\mathbb{Z}_{2,(0)}, \mathbb{Z}_{2,(d-2=1)}\right]^{\epsilon_2}$ cannot be realized. When $p = 2, 4$, the topologically ordered ground state manifold supports emergent $\mathbb{Z}_{2,(2)}$ and $\mathbb{Z}_{2,(2)} \times \mathbb{Z}_{2,(2)}$ symmetries, respectively. We refer by "Triv." to trivial symmetry group.

| $\mathbb{Z}_{4,(1)}$ Gapped Phases | | Dual $\left[\mathbb{Z}_{2,(0)}, \mathbb{Z}_{2,(d-2=1)}\right]^{\epsilon_2}$ Gapped Phases | |
|---|---|---|---|
| Symmetry of GS | Description | Symmetry of GS | Description |
| $\left[\mathbb{Z}_{4,(1)}\right]$ | Symmetry preserving | $\left[\mathbb{Z}_{2,(1)}\right]$ | Emergent $\mathbb{Z}_{2,(2)}$ |
| $\left[\mathbb{Z}_{2,(1)}\right]$ | Emergent $\mathbb{Z}_{2,(2)}$ | Triv. | Emergent $\mathbb{Z}_{2,(2)} \times \mathbb{Z}_{2,(2)}$ |
| Triv. | Emergent $\mathbb{Z}_{4,(2)}$ | $\left[\mathbb{Z}_{2,(d-2=1)}\right]$ | Emergent $\mathbb{Z}_{2,(2)}$ |

### 6.2.1 Phase diagrams and dualities

Let us study the duality in $d = 2$ or 3 dimensional spin models arising from partial gauging of $\mathbb{Z}_{2,(1)} \subset \mathbb{Z}_{4,(1)}$. After such a gauging, one obtains a spin model with a $\mathsf{G}^{\epsilon_2}_{(1,d-2)}$ symmetry, i.e., a $\mathbb{Z}_{2,(1)} \times \mathbb{Z}_{2,(d-2)}$ global symmetry with a mixed anomaly. For brevity, we only consider the simplest gapped phases in the spin system with $\mathbb{Z}_{4,(1)}$ symmetry, i.e., those corresponding to symmetry breaking of the 1-form symmetry to $\mathbb{Z}_{p,(1)} \subseteq \mathbb{Z}_{4,(1)}$. We simply consider the fixed-point Hamiltonians in each gapped phase and obtain the dual partially gauged Hamiltonians by isomorphism of bond algebras between (199) and (253). Such fixed-point Hamiltonians are given in (222) for $n = 4$ and $p = 1, 2, 4$. The results are summarized in Table 6 for $d = 3$ space dimensions. The Hamiltonian corresponding to no symmetry breaking and its dual under partial gauging are

$$\mathcal{H}_{[\mathbb{Z}_4]} = -\frac{1}{2}\sum_{\mathsf{e}} X_{\mathsf{e}} + \text{H.c} \longmapsto \mathcal{H}^{\vee}_{[\mathbb{Z}_4]^{\vee}} = -\sum_{\mathsf{e}} \sigma^x_{\mathsf{e}} \frac{1 + A_{\mathsf{e}}}{2}. \tag{254}$$

The ground states of $\mathcal{H}^{\vee}_{[\mathbb{Z}_4]^{\vee}}$ are eigenvalue $+1$ states of $A_{\mathsf{e}}$ and $\sigma^x_{\mathsf{e}}$ for all $\mathsf{e}$. This implies that the 1-form symmetry

$$\mathcal{U}_{(1)}(\Sigma^{(d-1),\vee}) = \prod_{\mathsf{e} \subset \Sigma^{(d-1),\vee}} \sigma^x_{\mathsf{e}} \left[ \bar{P}^{(+)}_{\mathsf{e}} + \bar{P}^{(-)}_{\mathsf{e}} A_{\mathsf{e}} \right], \tag{255}$$

acts as the identity on the ground states and therefore it is preserved. Let us now consider a product of $A_{\mathsf{e}}$ operators taken along an open line $L$ in $d = 2$ or along an open disc $D_2$ in $d = 3$ on the dual lattice. In $d = 2$, such a product delivers a bi-local operator $\sigma^x_{\mathsf{i}(L)} \sigma^x_{\mathsf{f}(L)}$ with $\mathsf{i}(L)$ and $\mathsf{f}(L)$ being the plaquettes at the two ends of $L$. Since $\sigma^x_{\mathsf{p}}$ is charged under the 0-form

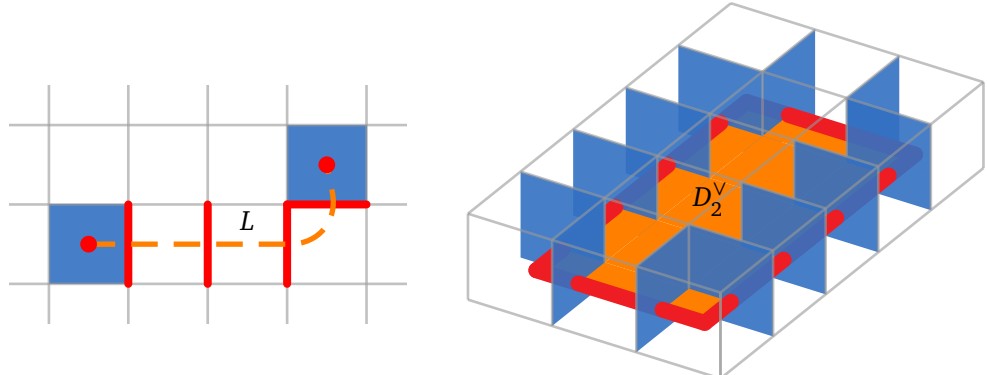

Figure 19: In $d = 2$, a product of $A_e$ operators along a line on the dual lattice furnishes a bi-local operator with support on the two endpoints of the line. Similarly, in $d = 3$, a product of $A_e$ operators on an open disc $D_2^\vee$ in the dual lattice furnishes a line operator $\mathcal{W}$ on the boundary of the disc.

dual symmetry $\mathcal{U}_{(d-2)}$, this signals the spontaneous breaking of the 0-form dual symmetry. Similarly, in $d = 3$, one obtains a line operator

$$\mathcal{W}(\partial D_2^\vee) = \prod_{e \in D_2^\vee} A_e = \prod_{p \in \partial D_2^\vee} \sigma_p^x. \tag{256}$$

The line $\mathcal{W}$ is topological in the low energy subspace in the sense that it commutes with the Hamiltonian and therefore does not cost any energy to deform and is charged under the $(d - 2 = 1)$-form symmetry. Therefore, $\mathcal{H}_{[\mathbb{Z}_4]^\vee}^\vee$ preserves $\mathbb{Z}_{2,(1)}$ while it breaks $\mathbb{Z}_{2,(1)}$ dual symmetry.

Next, we consider the partial symmetry breaking Hamiltonian $\mathcal{H}_{[\mathbb{Z}_2]}$ which dualizes as

$$\mathcal{H}_{[\mathbb{Z}_2]} = -\sum_e X_e^2 - \frac{1}{2}\sum_p B_p^2 \longmapsto \mathcal{H}_{[\mathbb{Z}_2]^\vee}^\vee = -\sum_e A_e - \sum_p \bar{B}_p. \tag{257}$$

Since the ground states of $\mathcal{H}_{[\mathbb{Z}_2]^\vee}^\vee$ are eigenvalue $+1$ states of $A_e$, it follows due to the above reasoning that $\mathbb{Z}_{2,(d-2)}$ dual symmetry is spontaneously broken. Similarly, the fact the $\bar{B}_p$ has eigenvalue $+1$ on the ground state subspace implies that one may consider an open disc on the direct lattice, which furnishes a line of $\sigma_e^z$ operators on the boundary of this disc. This line operator has a unit expectation value in the ground state of the fixed-point Hamiltonian but more generally a non-vanishing expectation value anywhere in the gapped phase labelled as $[\mathbb{Z}_2]^\vee$. Finally noting that this line operator is charged under $\mathbb{Z}_{2,(1)}$, implies that the 1-form symmetry is spontaneously broken. To summarize, we find that in the gapped phase $[\mathbb{Z}_2]^\vee$ the full symmetry $[\mathbb{Z}_{2,(d-2)}, \mathbb{Z}_{2,(1)}]$ is broken.

Finally, we move onto the $\mathbb{Z}_{4,(1)}$ symmetric gapped phase labelled as $\mathbb{Z}_{1,(1)}$ where the 1-form symmetry is completely broken. The fixed-point Hamiltonian and its dual under partial-gauging have the form

$$\mathcal{H}_{[\mathbb{Z}_1]} = -\frac{1}{2}\sum_p B_p + \text{H.c} \longmapsto \mathcal{H}_{[\mathbb{Z}_1]^\vee}^\vee = -\sum_p \sigma_p^z \frac{1 + \bar{B}_p}{2}. \tag{258}$$

This Hamiltonian breaks $\mathbb{Z}_{2,(1)}$ since its ground states are in the $\bar{B}_p = 1$ eigenspace. It however preserves the $\mathbb{Z}_{2,(d-2)}$ dual symmetry as can be readily confirmed.

# 7 Conclusion

In this paper, we explored various symmetry aspects of quantum spin models models with global higher-form finite Abelian symmetries on arbitrary $d$-dimensional lattices. Given a $p$-form symmetry corresponding to a finite Abelian group $\mathsf{G}$, we described (i) a systematic gauging of the group $\mathsf{G}$ or any subgroup $\mathsf{H} \subset \mathsf{G}$ for $p = 0, 1$, (ii) the gauging related duality maps between models with a $\mathsf{G}$ $p$-form symmetry and $\mathsf{G}_{(p,d-p-1)}$ higher group symmetry which has a $\mathsf{G}/\mathsf{H}$ $p$-form symmetry, a $\mathsf{H}^\vee$ $(d-p-1)$-form symmetry and a mixed 't-Hooft anomaly between these two higher-form symmetries and (iii) dualities between phase diagrams of spin models with the corresponding symmetries.

In Sec. 2, we detailed how the gauging of finite Abelian 0-form (sub)-symmetries can be understood as an isomorphism between a bond algebra symmetric with respect to $\mathsf{G}_{(0)}$ and a dual bond algebra symmetric under the dual $\mathsf{G}_{(0,d-1)}$ symmetry. In particular, we described how the symmetry sectors, i.e., twisted-boundary conditions and symmetry eigenvalue sectors, map under such an isomorphism of bond algebras. For the case of gauging a subgroup, we clarified how the mixed anomaly manifests in the symmetry structure of the dual bond algebra. In doing so, we clarified some anomaly-related subtle symmetry fractionalization patterns of higher-form symmetries in lattice spin models. In Sec. 3, we discussed these gauging-related dualities from a quantum field theory perspective.

In Secs. 4 and 5, we explored consequences of such gauging related dualities to phase diagrams of two and three dimensional spin models respectively. We specialized to $\mathbb{Z}_n$ clock models with $\mathbb{Z}_{n,(0)}$ symmetry and studied a Hamiltonian built as a linear combination of fixed-point Hamiltonians, one for each $\mathbb{Z}_{n,(0)}$-symmetric short-range entangled gapped phase. By dualizing such a Hamiltonian, using the bond algebra isomorphism obtained in Sec. 2, we could study various aspects of the phase diagram. In particular, we could pin-point how all these gapped phases dualize and how certain unconventional (beyond Landau) transitions are dual to Landau transitions under such gaugings. These studies have potential applications in understanding aspects of such exotic transitions in quantum spin models with global categorical symmetries, a subject about which little is understood. In Sec. 6, we studied the gauging of $\mathbb{Z}_{n,(1)}$ sub-symmetries in two and three spatial dimensions and applied the corresponding gauging related isomorphisms to spin models with such symmetries. Among other findings, we showed that in $d = 3$, a gauging of $\mathbb{Z}_{n,(1)}$ symmetry is realized as an automorphism on the symmetric bond algebra. This automorphism implied dualities between three-dimensional $\mathbb{Z}_k$ and $\mathbb{Z}_{n/k}$ topological orders and Hamiltonians self-dual under such automorphisms host emergent non-invertible symmetry structures [25].

# Acknowledgments

We thank Lakshya Bhardwaj, Lea Bottini, Clement Delcamp, Julia D. Hannukainen, Faroogh Moosavian, Christopher Mudry and Sakura Schafer Nameki for discussions.

**Funding information**    HM is supported by Engineering and Physical Sciences Research Council under New Horizon grant award no. EP/V048678/1 and the Leverhulme Trust Early Career Fellowship. ÖMA is supported by the Swiss National Science Foundation (SNSF) under Grant No. 200021 184637. AT and JHB received funding from the European Research Council (ERC) under the European Union's Horizon 2020 research and innovation program (Grant Agreement No. 101001902), the Swedish Research Council (VR) through grants number 2019-04736 and 2020-00214 and the Knut and Alice Wallenberg Foundation (KAW) via the project Dynamic Quantum Matter (2019.0068). The work of JHB was performed in part at the Aspen Center for Physics, which is supported by National Science Foundation grant PHY- 2210452.

# A   BF type description of bond algebra

In this Appendix, we present an alternative description of the gauging procedure presented in Sec. 2 as a BF-like theory of compact scalar fields. We may represent $\mathbb{Z}_n$ clock operators as

$$Z_v \sim e^{i\Phi_v}, \qquad X_v \sim e^{i\widetilde{\Phi}_v}, \tag{A.1}$$

which satisfy the commutation relations $[\widetilde{\Phi}_v, \Phi_{v'}] = 2\pi i \delta_{vv'}/n \mod 2\pi$. Here we can think of $\Phi_v$ as a compact scalar field and $\widetilde{\Phi}_v$ as its canonical momentum operator. Comparing with (7) and (9), we may write the $\mathbb{Z}_n$ symmetry operator as

$$\mathcal{U} = \prod_v e^{i\widetilde{\Phi}_v}, \tag{A.2}$$

and the bond algebra takes the form

$$B_{\mathbb{Z}_{n,(0)}} = \left\langle e^{i\widetilde{\Phi}_v}, e^{-i\int_e d\Phi} \,\middle|\, \forall\, v, e \right\rangle, \tag{A.3}$$

where $e^{-i\int_e d\Phi} = e^{i\Phi_{s(e)}} e^{-i\Phi_{t(e)}} = Z_{(e)} Z^\dagger_{t(e)}$. The local Hilbert space on each vertex is $n$-dimensional such that tensor product Hilbert space is spanned by states labelled by $\phi \in C^0(M_{d,\triangle}, \mathbb{Z}_n)$, where $\phi = \{\phi_v\}_v$ with $\phi_v = 0, 1, \ldots, n-1$. We choose to work in the eigenbasis of $e^{i\Phi_v}$ such that

$$\begin{aligned} e^{i\Phi_v}|\phi\rangle &= \omega_n^\phi |\phi\rangle, \\ e^{i\widetilde{\Phi}_v}|\phi\rangle &= |\phi + \delta^{(v)}\rangle, \end{aligned} \tag{A.4}$$

where $+$ denotes addition modulo $n$ and $\delta^{(v)}$ is a 0-cochain which evaluates to 1 on the vertex $v$ and 0 elsewhere, i.e., $\delta^{(v)}_{v'} = \delta_{v,v'}$.

In order to gauge the $\mathbb{Z}_n$ global symmetry, we similarly introduce a $\mathbb{Z}_n$ degree of freedom on each edge of the lattice. We denote the operators acting on the edges as $e^{iA_e}$ and $e^{iB_{e^\vee}}$, via the identification

$$Z_e \sim e^{iA_e}, \qquad X_e \sim e^{iB_{e^\vee}}. \tag{A.5}$$

Note that the $B$ operators are defined on $e^\vee$ which are $(d-1)$-cells of the dual lattice. Since, there is a canonical bijection between these $(d-1)$-cells of the dual lattice and the 1-cells of the direct lattice (see, for instance, Fig. 5) it is always possible to do so. These operators satisfy the commutation relations

$$[B_{e^\vee}, A_{e'}] = \frac{2\pi i\,\delta_{e,e'}}{n} = \frac{2\pi i\,\mathrm{Int}_{e^\vee,e'}}{n}, \tag{A.6}$$

where in the final expression, $\mathrm{Int}_{e^\vee,e'}$ is the intersection number of the $(d-1)$-cell $e^\vee$ with the edge $e'$. Upon introducing edge degrees of freedom, we span the Hilbert space by basis states $|a, \phi\rangle$, labelled by $a \in C^1(M_{d,\triangle}, \mathbb{Z}_n)$ and $\phi \in C^0(M_{d,\triangle}, \mathbb{Z}_n)$. The vertex operators act on the basis states as (A.4), while the edge operators act similarly as

$$\begin{aligned} e^{iA_e}|a, \phi\rangle &= \omega_n^{a_e}|a, \phi\rangle, \\ e^{iB_{e^\vee}}|a, \phi\rangle &= |a + \delta^{(e)}, \phi\rangle, \end{aligned} \tag{A.7}$$

where $\delta^{(e)}$ is a 1-cochain which evaluates to 1 on the edge $e$ and 0 elsewhere, i.e., $\delta^{(e)}_{e'} = \delta_{e,e'}$. After gauging, the physical Hilbert space is the gauge invariant subspace of the full Hilbert

space spanned by $\{|a, \phi\rangle\}$. The gauge invariant subspace is obtained as the identity eigensector of the Gauss operator

$$\mathcal{G}_{\mathsf{v}} = \exp\left\{ i\widetilde{\Phi}_{\mathsf{v}} + i\oint_{S_{\mathsf{v}}^{(d-1),\vee}} B \right\}, \tag{A.8}$$

where $S_{\mathsf{v}}^{(d-1),\vee}$ is a minimal $(d-1)$-sphere on the dual lattice that links with the vertex $\mathsf{v}$ (see Fig. 14). This a field theory notation version of equation (26), which was written in a more spin-model language.

A general gauge transformation is implemented by the operator $\mathcal{G}[\lambda] = \prod_{\mathsf{v}} \mathcal{G}_{\mathsf{v}}^{\lambda_{\mathsf{v}}}$ parametrized by a 0-cochain $\lambda \in C^0(M_{d,\triangle}, \mathbb{Z}_n)$ which acts on the basis states as

$$\mathcal{G}[\lambda]|a, \phi\rangle = |a + d\lambda, \phi + \lambda\rangle. \tag{A.9}$$

The bond algebra needs to be suitably modified post gauging such that all the operators are gauge invariant. We can do so by minimal coupling $d\Phi \longrightarrow d\Phi + A$, such that bond algebra becomes

$$\mathsf{B}_{\mathbb{Z}_{n,(d-1)}} = \left\langle e^{i\widetilde{\Phi}_{\mathsf{v}}}, e^{-i\int_{\mathsf{e}}(d\Phi + A)} \,\middle|\, e^{i\oint_L A} \overset{!}{=} 1, \, \mathcal{G}_{\mathsf{v}} \overset{!}{=} 1 \, \forall \, \mathsf{v}, \mathsf{e}, L \right\rangle. \tag{A.10}$$

There is an additional constraint $\exp\left\{i\oint_L A\right\} \overset{!}{=} 1$ for each loop $L$ on the lattice. This follows from the fact that this operator is the image of the operator $\exp\left\{i\oint_L d\Phi\right\} = 1$ in the pre-gauged algbera. Since gauging is a bond algebra isomorphism, it must map the identity operator to the identity operator. Compare with (31). A consequence of this is the fact that $da = 0 \bmod n$ and therefore $a \in Z^1(M_{d,\triangle}, \mathbb{Z}_n)$ and corresponds to a $\mathbb{Z}_n$-valued field.

Next, we seek a unitary transformation that disentangles the edges from the Gauss constraint, i.e.,

$$\mathcal{U}\mathcal{G}_{\mathsf{v}}\mathcal{U}^\dagger = e^{i\widetilde{\Phi}_{\mathsf{v}}}. \tag{A.11}$$

Such a transformation is achieved by the unitary

$$\mathcal{U} = \sum_{a,\phi} |a + d\phi, \phi\rangle\langle a, \phi| = \prod_{\mathsf{v}} \exp\left\{ i\Phi_{\mathsf{v}} \oint_{S_{\mathsf{v}}^{(d-1),\vee}} B \right\}, \tag{A.12}$$

which acts on the remaining operators as

$$\begin{aligned}
\mathcal{U}\exp\{iA_{\mathsf{e}}\}\mathcal{U}^\dagger &= \exp\{i(A + d\Phi)_{\mathsf{e}}\}, \\
\mathcal{U}\exp\left\{i\oint_{S_{\mathsf{v}}^{(d-1),\vee}} B\right\}\mathcal{U}^\dagger &= \exp\left\{i\oint_{S_{\mathsf{v}}^{(d-1),\vee}} B\right\}, \\
\mathcal{U}\exp\{i\Phi_{\mathsf{v}}\}\mathcal{U}^\dagger &= \exp\{i\Phi_{\mathsf{v}}\}, \\
\mathcal{U}\exp\{i\widetilde{\Phi}_{\mathsf{v}}\}\mathcal{U}^\dagger &= \exp\left\{i\widetilde{\Phi}_{\mathsf{v}} + i\oint_{S_{\mathsf{v}}^{(d-1),\vee}} B\right\}.
\end{aligned} \tag{A.13}$$

The bond algebra (A.10) becomes the following after the action of the unitary $\mathcal{U}$

$$\begin{aligned}
\widetilde{\mathfrak{B}}_{\mathbb{Z}_{n,(d-1)}} &= \left\langle e^{-iA_{\mathsf{e}}}, \exp\left\{i\widetilde{\Phi}_{\mathsf{v}} - i\oint_{S_{\mathsf{v}}^{(d-1),\vee}} B\right\} \,\middle|\, e^{i\widetilde{\Phi}_{\mathsf{v}}} \overset{!}{=} 1, \, e^{i\oint_L A} \overset{!}{=} 1, \, \forall \, \mathsf{v}, \mathsf{e}, L \right\rangle \\
&= \left\langle \exp\{-iA_{\mathsf{e}}\}, \exp\left\{-i\oint_{S_{\mathsf{v}}^{(d-1),\vee}} B\right\} \,\middle|\, e^{i\oint_L A} \overset{!}{=} 1, \, \forall \, \mathsf{v}, \mathsf{e}, L \right\rangle.
\end{aligned} \tag{A.14}$$

In the second line we have solved the Gauss constraint and frozen out all scalar field d.o.f., the dual bond algebra is organized by conjugate fields $A$ and $B$ reminiscent of BF-theories. This is the bond algebra of a $(d-1)$-form $\mathbb{Z}_{n,(d-1)}$ symmetry generated by closed loops operators

$$W_L = \exp\left(-\mathrm{i}\int_L A\right),\tag{A.15}$$

compare this to equation (42). Similar to the discussion in section 2.1.3, let us drop the constraint on contractible loops and define the bond algebra

$$\hat{\mathsf{B}}_{\mathbb{Z}_{n,(d-1)}} = \left\langle \exp\left\{-\mathrm{i}A_\mathsf{e}\right\},\ \exp\left\{-\mathrm{i}\oint_{S_\mathsf{v}^{(d-1),\vee}} B\right\}\ \Big|\ \forall\,\mathsf{v},\mathsf{e},L\right\rangle.\tag{A.16}$$

Similar to the discussion in section 2.1.3, we can extend the duality to this larger algebra but the other side will contain algebras with twist defects. Consider the commutative subalgebra

$$\hat{\mathfrak{B}}_{\left[\mathbb{Z}_{n,(d-1)},\mathbb{Z}_{n,(1)}\right]} = \left\langle \exp\left\{-\mathrm{i}\oint_L A\right\},\ \exp\left\{-\mathrm{i}\oint_{S_\mathfrak{v}^{(d-1),\vee}} B\right\}\ \Big|\ \forall\,\mathfrak{v},\mathfrak{e},L\right\rangle \subset \hat{\mathfrak{B}}_{\mathbb{Z}_{n,(d-1)}}.\tag{A.17}$$

This subalgebra has an extra emergent 1-form symmetry, generated by $(d-1)$-dimensional submanifolds

$$\Gamma(S^{(d-1)^\vee}) = \exp\left(-\mathrm{i}\oint_{S^{(d-1)^\vee}} B\right).\tag{A.18}$$

Compare these to (44) and (46). Similar to (47), we can write the $\mathbb{Z}_n$ toric code Hamiltonian as

$$\begin{aligned}
H &= -\sum_\mathsf{v} e^{-\mathrm{i}\oint_{S_\mathsf{v}^{d-1\vee}} B} - \sum_\mathsf{p} e^{-\mathrm{i}\oint_{L_\mathsf{p}} A} + \text{H.c.}\\
&= -\sum_\mathsf{v} e^{-\mathrm{i}\int_{X_\mathsf{v}^{d\vee}} H} - \sum_\mathsf{p} e^{-\mathrm{i}\int_{D_\mathsf{p}} F} + \text{H.c.},
\end{aligned}\tag{A.19}$$

where $S_\mathsf{v}^{d-1\vee}$ is a $d-1$ dimensional sphere in the dual lattice wrapping around the vertex $\mathsf{v}$, $L_\mathsf{p}$ is the curve around a plaquette $\mathsf{p}$, and $X_\mathsf{v}^{d\vee}$ is solid $d$-dim ball such that $\partial X_\mathsf{v}^{d\vee} = S_\mathsf{v}^{d-1\vee}$ and $D_\mathsf{p}$ is a disk such that $\partial D_\mathsf{p} = L_\mathsf{p}$. In the second line we used Stokes theorem and defined the 2- and $d$-form Field strength operators

$$F = \mathrm{d}A,\qquad H = \mathrm{d}B.\tag{A.20}$$

The ground state subspace of toric code requires

$$F = 0,\qquad \text{and}\qquad B = 0,\tag{A.21}$$

in other words that we have flat $A$ and $B$ connections. This is exactly the Hilbert space of the underlying TQFT, a $BF$-theory with the action

$$S = \frac{n}{2\pi}\int_M B\wedge dA.\tag{A.22}$$

# B  Parallel transport operator for $\mathbb{Z}_n$ spin-chains

A $\mathbb{Z}_n$ symmetric spin chain can be coupled to a background gauge field, also called a $\mathbb{Z}_n$ connection. This allows us to define parallel transport on the space of operators. In this appendix we construct the parallel transport operator explicitly in terms of spin operators. This construction works in any dimension and on any triangulation, no need for translation symmetry.

Consider a triangulation $M_{d,\triangle}$ of an oriented $d$ dimensional manifold $M_d$. There is a Hilbert space $\mathcal{V}_v \simeq \mathbb{C}^n$ associated to each vertex v of $M_{d,\triangle}$, with the total Hilbert space $\mathcal{V} = \bigotimes_v \mathcal{V}_v$. All linear operators on $\mathcal{V}$ are generated by $X_v$ and $Z_v$ with the algebra

$$Z_v^\alpha X_{\overline{v}}^g = \omega_n^{\alpha g\, \delta_{v\overline{v}}} X_{\overline{v}}^g Z_v^\alpha\,. \tag{B.1}$$

We want to construct a permutation operator $P_{v\overline{v}}$ with the property

$$P_{v\overline{v}} \mathcal{O}_v P_{v\overline{v}}^{-1} = \mathcal{O}_{\overline{v}}\,, \tag{B.2}$$

written in terms of the operators $X_v$ and $Z_v$. Here $P_{v\overline{v}}$ permutes local operators acting only on the vertices v and $\overline{v}$, while commuting with any other local operators. The space of linear invertible operators at each vertex

$$GL(\mathcal{V}_v) \simeq \mathrm{span}_{\mathbb{C}} \left\{ X_v^g Z_v^\alpha \,\middle|\, g,\alpha = 0,\dots,n-1 \right\}, \tag{B.3}$$

can be endowed with the inner product

$$\langle A,B\rangle = \frac{1}{n}\,\mathrm{tr}_{\mathcal{V}_v}\left[A^\dagger B\right], \qquad A,B \in GL(\mathcal{V}_v)\,. \tag{B.4}$$

A short calculation shows that with this inner product, the above chosen basis is orthonormal

$$\langle X_v^g Z_v^\alpha, X_v^{\overline{g}} Z_v^{\overline{\alpha}} \rangle = \delta_{g,\overline{g}}\delta_{\alpha,\overline{\alpha}}\,. \tag{B.5}$$

We can extend this basis to all of $GL(\mathcal{V}) \equiv \mathcal{L}(\mathcal{V})$, by using the standard isomorphism

$$GL\left(\bigotimes_v \mathcal{V}_v\right) \simeq \bigotimes_v GL(\mathcal{V}_v)\,, \tag{B.6}$$

where on $\mathcal{V}$ we normalize the inner product as

$$\langle A,B\rangle = \frac{1}{n^{\#\text{vertices}}}\,\mathrm{tr}_{\mathcal{V}}\left[A^\dagger B\right], \qquad A,B \in \mathcal{L}(\mathcal{V})\,. \tag{B.7}$$

Using the eigenbasis of the $Z_v$ operators

$$Z_v|\phi\rangle = \omega_n^{\phi_v}|\phi\rangle\,, \tag{B.8}$$

we can explicitly construct the permutation operator as

$$P_{v\overline{v}} = \sum_\phi |S_{v\overline{v}}\cdot\phi\rangle\langle\phi|\,, \tag{B.9}$$

where $S_{v\overline{v}}$ acting on the labels $\phi = \{\phi_v\}$, swaps $\phi_v$ and $\phi\overline{v}$. One can easily see that following properties

$$P_{v\overline{v}}|\dots,\phi_v,\dots,\phi_{\overline{v}},\dots\rangle = |\dots,\phi_{\overline{v}},\dots,\phi_v,\dots\rangle\,, \qquad P_{v\overline{v}}^2 = 1\,, \tag{B.10}$$

as wanted. In order to write this operator in terms of the aforementioned basis of $\mathcal{L}(\mathcal{V})$, let us consider a general superposition

$$P_{v\bar{v}} = \sum_{g,\alpha=0}^{n-1} \sum_{\bar{g},\bar{\alpha}=0}^{n-1} \mathcal{M}_{g,\alpha}^{\bar{g},\bar{\alpha}} X_v^g Z_v^\alpha X_{\bar{v}}^{\bar{g}} Z_{\bar{v}}^{\bar{\alpha}}. \tag{B.11}$$

We are only expanding this in the subspace of $\mathcal{L}(\mathcal{V})$ corresponding to operators that only act on the vertices $v$ and $\bar{v}$, since it must commute with any operator $\mathcal{O}_{v'}$ such that $v' \neq v$ or $\bar{v}$. The coefficients can be computed using the inner product (B.7) and the orthonormality of our basis

$$\left\langle \prod_v X_{v'}^{g_{v'}} Z_{v'}^{\alpha_{v'}}, P_{v\bar{v}} \right\rangle = \mathcal{M}_{g_{\bar{v}},\alpha_{\bar{v}}}^{g_v,\alpha_v} \prod_{v' \neq v,\bar{v}} \delta_{g_{v'},\bar{g}_{v'}} \delta_{\alpha_{v'},\bar{\alpha}_{v'}}. \tag{B.12}$$

In particular, using (B.9) and only the basis vectors of the relevant two-body subspace, one can show by a calculation that

$$\mathcal{M}_{g_{\bar{v}},\alpha_{\bar{v}}}^{g_v,\alpha_v} = \langle X_v^{g_v} Z_v^{\alpha_v} X_{\bar{v}}^{g_{\bar{v}}} Z_{\bar{v}}^{\alpha_{\bar{v}}}, P_{v\bar{v}} \rangle \tag{B.13}$$

$$= \frac{1}{n} \omega^{g_v \alpha_v} \delta_{\alpha_v + \alpha_{\bar{v}},0} \delta_{g_v + g_{\bar{v}},0}. \tag{B.14}$$

We can therefore write $P_{v\bar{v}}$ in the form we wanted

$$P_{v\bar{v}} = \frac{1}{n} \sum_{g,\alpha=0}^{n-1} \omega^{g\alpha} X_v^g Z_v^\alpha X_{\bar{v}}^{-g} Z_{\bar{v}}^{-\alpha}. \tag{B.15}$$

One can readily check that this satisfies all the needed properties such as (B.2). For a standard spin-model with $n = 2$ we get

$$P_{v\bar{v}} = \frac{1}{2} \Big[ \mathbb{1} + X_v \otimes X_{\bar{v}} - (XZ)_v \otimes (XZ)_{\bar{v}} + Z_v \otimes Z_{\bar{v}} \Big] \tag{B.16}$$

$$= \frac{1}{2} \Big[ \mathbb{1} + \sigma_v^x \otimes \sigma_{\bar{v}}^x + \sigma_v^y \otimes \sigma_{\bar{v}}^y + \sigma_v^z \otimes \sigma_{\bar{v}}^z \Big]. \tag{B.17}$$

For each edge e, we can define

$$T_e = P_{s(e),t(e)}, \tag{B.18}$$

where $s(e)$ and $t(e)$ is the source and target of e, respectively.

For regular lattices, this can be used to construct translation operators in terms of spin operators. For example for a one-dimensional lattice of $L$ sites, we have

$$T = \prod_{i=1}^{L-1} P_{i,i+1} = P_{1,2} P_{2,3} \cdots P_{L-1,L}, \tag{B.19}$$

which satisfies

$$T|\phi_1 \cdots \phi_L\rangle = |\phi_L \phi_1 \phi_2 \cdots \phi_{L-1}\rangle. \tag{B.20}$$

On general triangulations we can construct Parallel transport operators, when coupling the global symmetry to a background connection (gauge field). See equation (14) and surrounding discussion.

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
