# Peer review of "Symmetry fractionalization, mixed-anomalies and dualities in quantum spin models with generalized symmetries"

_SciPost Physics, doi:SciPost Phys. 18, 097 (2025)_

## Round 2 · Referee Report · Anonymous (Referee 1) · 2024-5-21

Report

The manuscript discusses (1) gauging a finite Abelian subgroup higher-form symmetry in lattice models, exam the dual higher-form symmetry and demonstrate the anomalies from group extension as discussed in e.g. [18],[3] from field theory perspective.

(2) Using the gauging procedure, the authors models that are self-dual under gauging and thus enjoy Kramers-Wannier type non-invertible duality symmetry as in [24],[25].

The discussion is systematic and the I recommend publication provided the following comments are addressed:

  • For finite Abelian groups $G= prod Z_{N_i}$ with integers $N_i$, maybe add that the dual is isomorphic to itself $G^vee\cong G$

  • The manuscript used a terminology that equates "gauging" with "duality". However, gauging a symmetry in general leads to a different theory, e.g. SPT v.s. topological order. The former is short range entangled while the later has long range entanglement, but they can be related by gauging a symmetry. In general, gauging a symmetry corresponds to a topological interface between different theories, and only when the theory is "self-dual" under gauging the topological interface becomes a topological domain wall within the same theory such as the Kramers-Wannier type duality symmetry.

  • In (6.27), the local Z_e^p on each edge e and the usual vertex term A_v= prod XX^dagX X^dag X X^dag both commute with the Hamiltonian, but they are not Hamiltonian terms and also do not commute. Therefore, the model has local logical degrees of freedom. For instance, the states |0> and Z_e^p|0> for each edge e must be different due to different eigenvalues of the vertex term A_v. So I don't think the GSD is just (n/p)^b3 and the ground state subspace is not just Z_{n/p} topological gauge theory.

Recommendation

Ask for minor revision

---

## Round 2 · Referee Report · Anonymous (Referee 2) · 2024-12-29

Report

In this paper, the authors studied the gauging of 0-form and 1-form finite abelian symmetries, both on the lattice and in field theory. The discussion highlighted the relation between the symmetry before gauging and the emergent dual symmetry after gauging. Moreover, the authors discussed explicitly how the symmetric operator space is mapped under such gauging and correspondingly how the phase diagram is mapped before and after. While the results are not too surprising, the presentation is systematic and easy to follow. It is a good addition to the literature, especially for people who are familiar with the lattice formulation and want to learn the field theory formulation and vice versa.

I have only minor suggestions regarding the writing of this paper:

  1. on page 9, please check for typos in the "organization of the paper" paragraph.

  2. the paper focused on abelian symmetries. I wonder how easy the conclusions can be generalized to non-abelian symmetries and what new features can emerge. I hope the authors can comment on this point.

Recommendation

Publish (meets expectations and criteria for this Journal)

---

## Round 3 · Author Response

Referee 1:
-- Statements emphasizing that the Pontryagin group of an Abelian group is isomorphic to the group itself appear at several places in the draft. See for example above eq. 3.2. However we would like to emphasize that the symmetry category upon gauging an Abelian group in d+1 dimensions is not isomorphic to the Abelian group, but instead is the higher representation category dRep(G). This statement can be found in the last paragraph in Section 2.1.
-- That is correct. Gauging a symmetry can indeed relate short-range and long range entangled phases as the referee points out. These are dualities in the sense that they are spectrum and correlation function preserving isomorphisms between two distinct quantum systems. In certain special cases, the quantum system before and after gauging are isomorphic. In such cases, the codimension-1 gauging interface can be treated as a (non-invertible) 0-form symmetry operator.
-- We remind the referee that the operator Z_e^{p} is not within the Z_n 1-form symmetric bond algebra, i.e., it takes us out of the physical constrained state space with 1-form symmetry. If the referee insists on working with an unconstrained state space, they may add the 1-form symmetry generators associated to each vertex of the lattice and take the corresponding coupling constant to infinity, then one reproduces the claimed Z_{n/p} topological order.
Referee 2:
-- We thank the referee for pointing these typos out. We now have corrected them.
-- This is indeed a very interesting question. Gauging non-Abelian (sub) symmetries lead to more interesting symmetry categories which are necessarily non-invertible. In general, in d+1 spatial dimensions, gauging a non-Abelian G 0-form symmetry, produces a dual quantum system with a dRep(G) symmetry. This contains a non-invertible 1-form subsymmetry Rep(G) corresponding to topological Wilson lines obtained after gauging G. All other symmetry generators in dRep(G) are obtainable as condensation defects of the Rep(G) lines. We have now added these comments as well as corresponding references at the end of Sec. 2.1.

---

## Editorial Decision

published